# Freshwater genome-reduced bacteria exhibit pervasive episodes of adaptive stasis

Lucas Serra Moncadas[1], Cyrill Hofer [1], Paul-Adrian Bulzu [2], Jakob Pernthaler[1] & Adrian-Stefan Andrei [1] ✉

The emergence of bacterial species is rooted in their inherent potential for continuous evolution and adaptation to an ever-changing ecological landscape. The adaptive capacity of most species frequently resides within the repertoire of genes encoding the secreted proteome (SP), as it serves as a primary interface used to regulate survival/reproduction strategies. Here, by applying evolutionary genomics approaches to metagenomics data, we show that abundant freshwater bacteria exhibit biphasic adaptation states linked to the eco-evolutionary processes governing their genome sizes. While species with average to large genomes adhere to the dominant paradigm of evolution through niche adaptation by reducing the evolutionary pressure on their SPs (via the augmentation of functionally redundant genes that buffer mutational fitness loss) and increasing the phylogenetic distance of recombination events, most of the genome-reduced species exhibit a nonconforming state. In contrast, their SPs reflect a combination of low functional redundancy and high selection pressure, resulting in significantly higher levels of conservation and invariance. Our findings indicate that although niche adaptation is the principal mechanism driving speciation, freshwater genome-reduced bacteria often experience extended periods of adaptive stasis. Understanding the adaptive state of microbial species will lead to a better comprehension of their spatiotemporal dynamics, biogeography, and resilience to global change.

Bacteria display remarkable adaptability, thriving in a wide spectrum of environments ranging from the mundane to the extreme[1]. They can survive anywhere, from icy poles and scorching geysers to the depths of the ocean or the heights of the atmosphere[2–5]. This ability to flourish across large physicochemical gradients largely hinges not on the diversity of their metabolic machinery[6] but on their capacity to use a wide range of substrates to power a conserved array of biochemical reactions[7]. In this context, extant chemical diversity fosters the coexistence of metabolically concurrent bacteria and significantly contributes to their constant diversification. Consequently, genetically identical populations gradually diverge and undergo speciation as they adapt to distinct niche facets by accumulating fitness-enhancing variations[8].

While the diversity within species is fine-tuned by ongoing processes of variation generation, natural selection, and genetic drift, variability is introduced into populations through mutation and gene flow. Genetic drift operates randomly, leading to the elimination of genetic variation within a population, while natural selection selectively preserves or eliminates variations based on their fitness advantages or disadvantages[9].

Several evolutionary frameworks have been developed to elucidate the origin of bacterial species and their intrapopulation diversity, with adaptive models being the most accepted and widespread[8,10,11]. Although these models may vary in their proposed equilibrium between mutation/recombination and selection forces, they are united by their shared aim of explaining bacterial diversity through niche

[1]Limnological Station, Department of Plant and Microbial Biology, University of Zurich, Kilchberg, Switzerland. [2]Department of Aquatic Microbial Ecology, Institute of Hydrobiology, Biology Centre of the Czech Academy of Sciences, České Budějovice, Czech Republic. ✉e-mail: stefan.andrei@limnol.uzh.ch

adaptation (i.e., periodic selection or phage predation dynamics)[8,11]. Though certain predictions of these models have been confirmed in environmental microbial communities (genome-wide selective sweeps[12,13] and genomic islands[11]), there has been limited exploration of the process underlying bacterial adaptation to their natural niche.

Bacteria sense and respond to the surrounding physiochemical environment via proteins that are secreted into the extracellular milieu, confined to periplasmic space, or tethered to their plasma membrane/cell wall[14]. These proteins collectively referred to as the secreted proteome, play essential roles in bacterial niche adaptation by enabling recognition and uptake of nutrients, communication with other bacteria, surface attachment, and signal transduction[15]. According to the adaptive models of bacterial speciation, the SP is expected to evolve faster[16] than the corresponding cytoplasmic proteome (CP) as it represents the main interface used by bacteria to fine-tune survival/reproduction strategies (i.e., phage evasion and nutrient acquisition). Henceforth, we will employ the term secreted proteome (SP) to encompass all proteins that undergo translocation across the membrane. This includes proteins localized in the periplasmic space, associated with the membrane/cell wall, and those ultimately released into the external environment. In this study, proteins containing a signal peptide were designated as belonging to the SP, while those lacking this feature were assigned to the CP. Consequently, we will use the term secreted proteome to denote proteins computationally identified to possess a signal peptide, and cytoplasmic proteome for the remainder. It is important to note that within the scope of this study, the term proteomes refers to the entirety of proteins encoded by a specific bacterial species cluster. It is crucial to emphasize that these designations do not reflect a physiological state, as additional mass spectrometry-based experimental techniques were not employed.

In this work, we utilize metagenomic time-series to characterize the diversity of dominant freshwater bacterial species and elucidate the eco-evolutionary factors driving their diversification. By employing ecogenomics approaches on genome-resolved metagenomic data, we reveal the evolutionary strategies and forces shaping bacterial lifestyles. Our analyses underscore niche adaptation as the principal driver of speciation, while also revealing the widespread occurrence of extended periods of adaptive stasis among abundant freshwater species with small genome sizes.

## Results and discussion
### pdCEL-prokaryotic diversity in central European lakes
An extensive database of ~5500 prokaryotic metagenome-assembled genomes (MAGs) was constructed by applying genome-resolved metagenomics techniques. This involved analysing 52 independent shotgun-sequenced samples, collectively comprising around 11 billion reads and 3.31 Tb (Fig. 1). The datasets were obtained from time-series samples collected from five freshwater lakes (Fig. 2) spanning a range of trophic states, from oligotrophic to dystrophic (see "Methods"). We will henceforth refer to this database as pdCEL (prokaryotic diversity in Central European Lakes). pdCEL was further divided into small and large genome species based on previous observations that bacteria with reduced genome sizes often exhibit specific lifestyle strategies[17]. Species with estimated genome sizes ≤2.1 Mbp were classified as small, consistent with previous size limits utilised to refer to genome-reduced bacteria[18]. This threshold represents the lower quartile of the genome size distribution in the GTDB R05-RS95 database[19] (Supplementary Fig. S1). For simplicity of data presentation, all species with predicted genome sizes >2.1 Mbp were included in the large category. The redundancy of pdCEL was utilised to identify MAGs that belonged to the same species. Thus, the MAGs were grouped using the established criterion for prokaryotic species definition, which relies on ensuring that the average nucleotide identity values between genomes exceed 95%[20]. In order to have a representative sample of

intrapopulation diversity, our downstream analyses focused on species that contained nine or more MAGs ($n = 30$ species clusters; median completeness = 77.03%).

### Contrasting intraspecific amino acid diversity patterns
The analysis of amino acid similarity values among proteomes belonging to the same species cluster led to intriguing findings. While some species aligned with the predictions of the adaptive models, displaying greater dissimilarity in their SPs than in CPs, others exhibited unexpectedly conserved SPs (Fig. 3a). This unanticipated discrepancy challenges the assumptions of ecological speciation models and was observed across multiple species from a variety of taxonomic groups (Fig. 3a). Further investigation revealed that coding density, which was used as a surrogate for genome size (Supplementary Fig. S2), was the primary explanatory variable of the observed dichotomy ($\rho = 0.77$, $P$ value = 2.1e-06; Fig. 3b). As a result, a conspicuous pattern emerged regarding the level of evolutionary conservation observed between SPs and CPs: the larger genome species exhibited SPs characterised by comparatively lower evolutionary conservation, whereas the smaller genome species displayed a divergent pattern, showcasing highly conserved SPs (Fig. 3a, b). After normalising the intraspecific amino acid identity values with the within-species genetic distance, it became evident that the level of dissimilarity in CPs is comparable between species with large and small genomes, whereas that of SPs is not (Fig. 3c). One possible interpretation of this observation is that the SPs, which differentiate the two categories, are involved in lifestyle strategies and undergo selection at the niche level, while the similarity in CPs across categories reflects the selection for function. Thus, it becomes apparent that while genetic drift acts uniformly across populations (purging both CP and SP diversity simultaneously), selection can operate at different levels and may be driven by different factors: niche adaptation in the case of SPs and the conservation of core metabolic functions in the case of CPs.

### Dynamics of intraspecific variation generation
Subsequent analyses revealed that within-species variation is intricately influenced by the interplay between mutation and recombination (Fig. 4). The results indicate that, although recombination occurs less frequently than mutation, its impact is more significant than previously thought[21]. Particularly noteworthy is the striking contrast in the ν values, representing the average length of imports (i.e., DNA fragments introduced through recombination) for the SPs of large-genome bacteria (Fig. 4). This observation not only suggests that large-genome bacteria engage in recombination with more phylogenetically distant groups but also underscores that, despite its infrequency, recombination introduces sequence novelty. Remarkably, this novel sequence diversity is preferentially preserved in the SP of large-genome species where it likely contributes to proteome variability.

Gene selection force analyses revealed that species with large genomes experience comparable levels of negative selection pressure in both their SPs and CPs (~61% of protein-coding genes; Fig. 5a), with most of their adaptive potential located in the former (~1.7% of protein-coding genes under positive selection) (Fig. 5a). In contrast, the SPs of species with small genomes are characterised by the near absence of positive selection pressure and a reduction in the number of genes and sites evolving under negative selection when compared with the CPs (Pearson's Chi-squared test: $\chi^2 = 44.71$, df = 2, $P < 0.001$; Fig. 5a). Given that genes coding for SPs and CPs typically evolve under similar mutational rates (as they are located on the same bacterial chromosome) and are therefore subjected to similar recombination effects, a reduction in the number of genes under negative selection compensated by the increase in the invariable genes indicates enhanced selection pressure[22]. This suggests that even mutations that do not impact the amino acid identity within the SPs incur a fitness cost. This

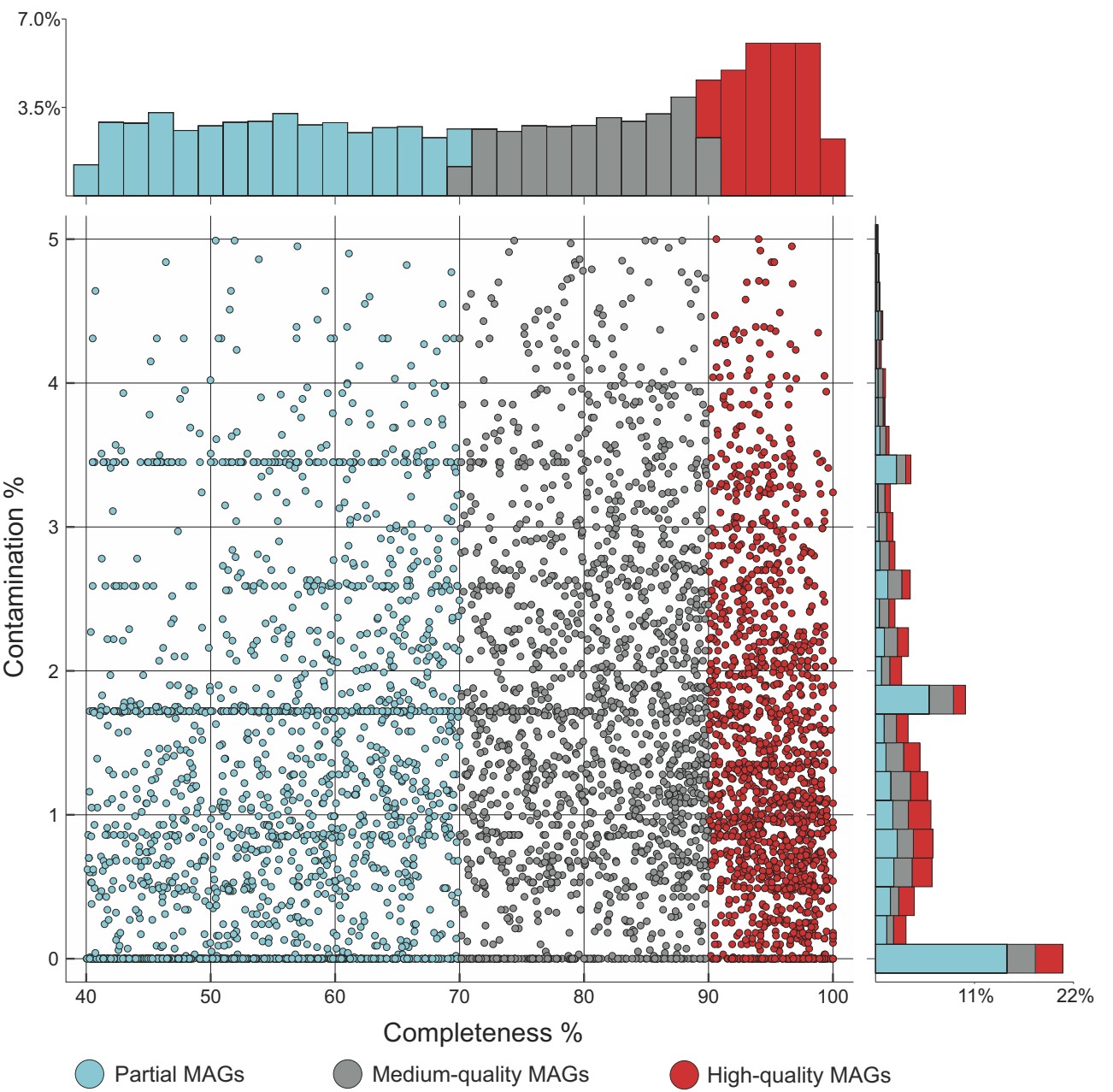

**Fig. 1 | Estimated completeness and contamination for 5519 MAGs present in the pdCEL database.** Only MAGs with contamination ≤5% and completeness ≥40% were retained. Three categories of MAGs were defined based on CheckM estimated completeness levels: high-quality MAGs (completeness ≥90%) representing 27.7% (1530) of the total, medium-quality MAGs (completeness 70–89%) amounting to 30.1% (1,663), and partial MAGs (completeness <70%) the remaining 42.2% (2332). The histogram parallel to the *X* axis shows the percentage of MAGs at different levels of completeness while the one parallel to the *Y* axis indicates the percentage of MAGs at varying levels of contamination. Raw data is provided as a Source Data file.

interpretation aligns with the recent observations of Shen and colab. who showed that even synonymous mutations decrease fitness by altering the transcription levels of the mutated genes in yeasts[23]. If this also holds true for small genome species, it would indicate that the expression levels of genes encoding SPs are of greater significance for their survival than those encoding CPs. Nonetheless, it is crucial to acknowledge that in some bacterial species, the mutation rate exhibits symmetry around the origin of replication[24,25].

### Protein length disparities and functional redundancy

Protein length comparisons showed that the CPs of both large and small genome species were of similar size, but that their SPs showed notable length disparities (Fig. 5c). The subsequent analysis of protein segment subcellular localisation revealed that the non-cytoplasmic and transmembrane regions of SPs in small genome species were significantly shorter, accounting for most of the observed differences (Fig. 5d). While identifying the exact cause of length reduction in SPs compared to CPs remains elusive, it is plausible that small genome species undergo more frequent SP turnover, involving the removal and degradation of existing proteins, along with a dynamic renewal process, where new proteins are synthesized to replace those removed. Consequently, the reduction in length could potentially alleviate some of the energetic expenses associated with the translation and transcription processes.

To understand the emergence of contrasting conservation patterns between the SPs and CPs of species with different genome sizes, we further looked at the proportion of functionally redundant proteins (Fig. 6b). Interestingly, we found that larger genome species tend to

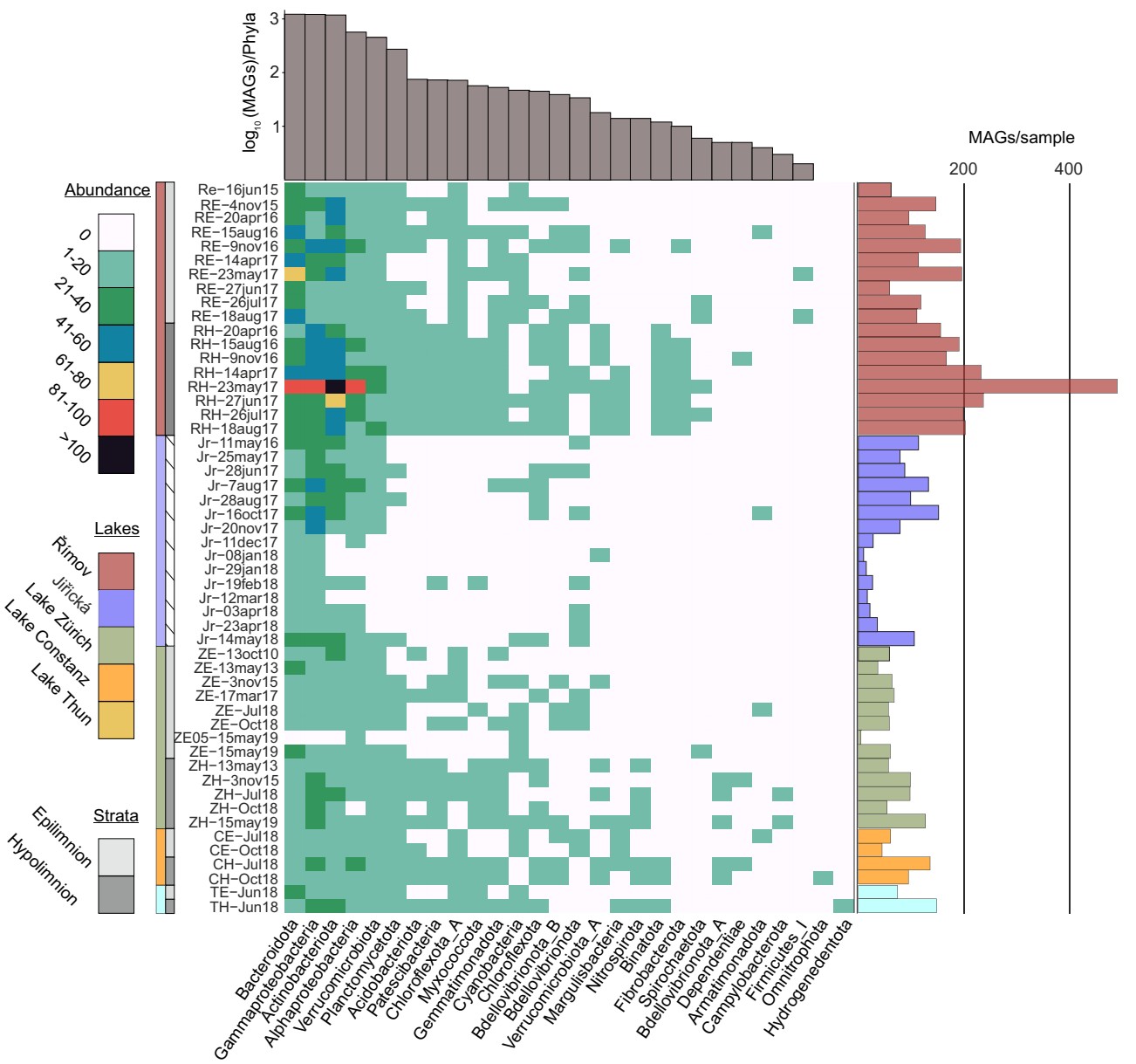

**Fig. 2 | The taxonomic distribution of the pdCEL database, with the GTDB R05-RS95 database taxonomic classification of 5519 MAGs presented in a sample-based manner.** The colored panels highlight the abundance of taxonomic categories (number of MAG species per phylum), sample provenance, and lake strata (as indicated in the left legend). The *X* axis displays the phylum-level taxonomy, while the *Y* axis shows sample identifiers. Raw data is provided as a Source Data file.

have a higher proportion of functionally redundant proteins in their SPs rather than in their CPs (Fig. 6b). In contrast, small genome species exhibited an inverse pattern with a higher proportion of functionally unique proteins present in their SPs (Fig. 6b). Thus, larger genome species enhance their niche adaptability through the maintenance of duplicated SPs coding genes/functions, proving an evolutionary playground for their SPs to adapt and diversify without the inconveniences caused by loss of fitness mutations[26]. Conversely, the limited functional redundancy within the SPs of small genome species poses a potential obstacle to evolutionary innovation. Any variation (caused by mutation or recombination) occurring in a single-copy gene has the capacity to impact the organism's fitness and is likely to be eliminated by selection. This mutational cost may be further exacerbated in genome-reduced species bearing minimal physiological redundancies. Thus, while the secreted proteomes of large genome species typically harboured a diverse set of proteins that were functionally identical yet divergent in sequence, the SPs of small genome species showed the opposite (Fig. 6a, c).

## Similar interspecific amino acid diversity patterns

Our finding that small genome species display adaptive stasis implies that their populations might progressively differentiate at the level of CPs, eventually giving rise to novel species featuring an SP that is more conserved than the corresponding CP. However, intra-genus (*n* = 7 genera) amino acid similarity analyses showed that both small and large genome species diverge through SP alterations (Fig. 7). This indicates that adaptive stasis as observed here is not an evolutionary lobster trap and that small genome species only transiently halt the evolution of their secreted proteomes. This conclusion is supported by the small genome species *Planktophila sp.6* which does not seem to experience adaptive stasis and displays a more conserved CP (Fig. 1a). It is worth noting that this species demonstrated the widest

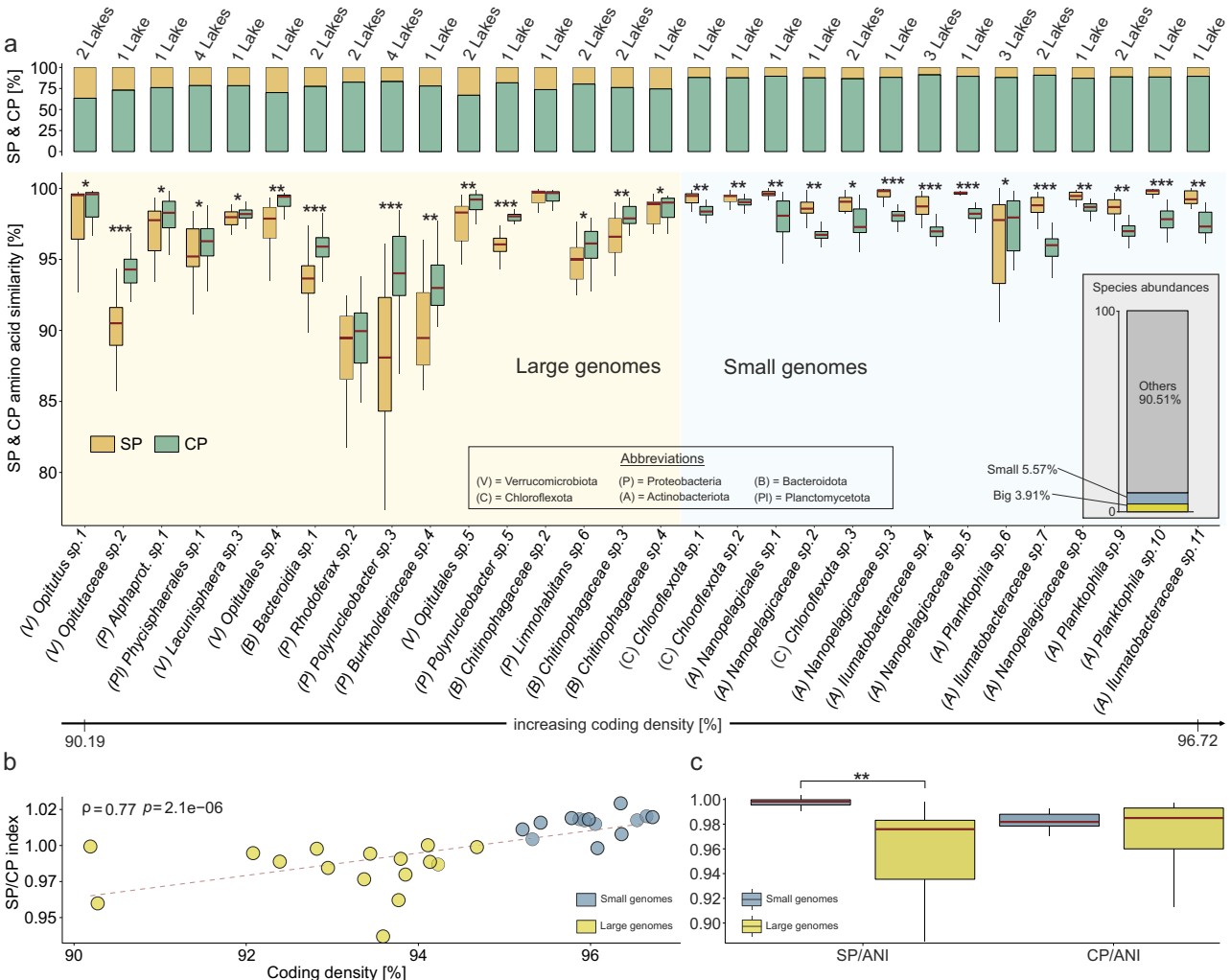

**Fig. 3 | Proteome similarity within species-level boundaries. a** The upper panel depicts the relative proportions of secreted (SP) and cytoplasmic (CP) proteomes within the analyzed species clusters. The bottom panel indicates the average amino acid identity (AAI) within species-level boundaries between the SPs and CPs ((A) *Illumatobacteraceae.11*: CP $n = 36$, SP $n = 36$; (A) *Illumatobacteraceae sp.4*: CP $n = 45$, SP $n = 45$; (A) *Illumatobacteraceae sp.7*: CP $n = 55$, SP $n = 55$; (A) *Nanopelagicaceae.2*: CP $n = 45$, SP $n = 45$; (A) *Nanopelagicaceae.3*: CP $n = 55$, SP $n = 55$; (A) *Nanopelagicaceae.5*: CP $n = 66$, SP $n = 66$; (A) *Nanopelagicaceae.8*: CP $n = 36$, SP $n = 36$; (A) *Nanopelagicales.1*: CP $n = 36$, SP n = 36; (A) *Planktophila sp.10*: CP $n = 55$, SP $n = 55$; (A) *Planktophila sp.6*: CP $n = 91$, SP $n = 91$; (A) *Planktophila sp.9*: CP $n = 36$, SP $n = 36$; (B) *Bacteroidia sp.1*: CP $n = 136$, SP $n = 136$; (B) *Chitinophagaceae sp.2*: CP $n = 78$, SP $n = 78$; (B) *Chitinophagaceae sp.3*: CP $n = 78$, SP $n = 78$; (B) *Chitinophagaceae sp.4*: CP $n = 55$, SP $n = 55$; (C) *Chloroflexota sp.1*: CP $n = 55$, SP $n = 55$; (C) *Chloroflexota sp.2*: CP $n = 91$, SP $n = 91$; (C) *Chloroflexota sp.3*: CP $n = 45$, SP $n = 45$; (P) *Alphaprot. sp.1*: CP $n = 55$, SP $n = 55$; (P) *Burkholderiaceae sp.4*: CP $n = 36$, SP $n = 36$; (P) *Limnohabitans sp.6*: CP $n = 45$, SP $n = 45$; (P) *Polynucleobacter sp.3*: CP $n = 171$, SP $n = 171$; (P) *Polynucleobacter sp.5*: CP $n = 36$, SP $n = 36$; (P) *Rhodoferax sp.2*: CP $n = 45$, SP $n = 45$; (Pl) *Phycisphaerales sp.1*: CP $n = 136$, SP $n = 136$; (V) *Lacunisphaera sp.3*: CP $n = 55$, SP $n = 55$; (V) *Opituaceae sp.2*: CP $n = 66$, SP $n = 66$; (V) *Opitutales sp.4*: CP $n = 36$, SP $n = 36$; (V) *Opitutales sp.5*: CP $n = 66$, SP $n = 66$; (V) *Opitutus sp.1*: CP $n = 36$, SP $n = 36$). Species clusters are ordered based on their coding density (bottom black arrow). Stars indicate statistical differences as determined by two-sided Pairwise Wilcoxon rank-sum tests ((A) *Nanopelagicaceae sp.5*: P value = 3.089104e-22, (A) *Planktophila sp.9*: P value = 1.220085e-10, (A) *Planktophila sp.6*: P value = 2.372142e-03, (A) *Planktophila sp.10*: P value = 3.843260e-19, (P) *Polynucleobacter sp.3*: P value = 1.379306e-30, (P) *Polynucleobacter sp.5*: P value = 3.037072e-13, (P) *Rhodoferax sp.2*: P value = 1.700911e-01, (P) *Burkholderiaceae sp.4*: P value = 3.754924e-05, (V) *Opituaceae sp.2*: P value = 2.539766e-21, (C) *Chloroflexota sp.2*: P value = 1.046075e-08, (C) *Chloroflexota sp.3*: P value = 1.006413e-03, (A) *Nanopelagicaceae*

*sp.3*: P value = 2.787598e-18, (C) *Chloroflexota sp.1*: P value = 6.095921e-13, (B) *Bacteroidia sp.1*: P value = 2.917558e-27, (A) *Illumatobacteraceae sp.7*: P value = 2.133974e-16, (A) Illumatobacteraceae sp.11: P value = 2.060721e-11, (A) *Nanopelagicales sp.1*: P value = 6.234337e-10, (P) *Alphaprot. sp.1*: P value = 5.805572e-04, (B) *Chitinophagaceae sp.3*: P value = 9.695927e-10, (B) *Chitinophagaceae sp.2*: P value = 9.251429e-01, (B) *Chitinophagaceae sp.4*: P value = 5.272749e-02, (V) *Opitutales sp.4*: P value = 1.833335e-07, (A) *Nanopelagicaceae sp.2*: P value = 2.962116e-15, (V) *Opitutales sp.5*: P value = 1.328782e-08, (Pl) *Phycisphaerales sp.1*: P value = 2.016058e-03, (A) *Illumatobacteraceae sp.4*: P value = 8.537497e-15, (V) *Lacunisphaera sp.3*: P value = 1.173421e-02, (P) *Limnohabitans sp.6*: P value = 8.110538e-04, (V) *Opitutus sp.1*: P value = 2.284776e-02, (A) *Nanopelagicaceae sp.8*: P value = 1.028353e-09). The bottom label displays species identifiers. A stacked barplot depicts species abundances. Small genomes are depicted in blue color, while the large ones are in yellow. **b** The relationship between the SP/CP similarity index (intraspecific index obtained by dividing median amino acid similarity values of secreted and cytoplasmatic proteomes) and genome coding density assessed by Spearman's correlation test ($\rho = 0.7730$; S = 1020; P value = 2.086e-06). Small MAGs are depicted in blue, while large ones are shown in yellow (Large genome species $n = 16$, Small genome species $n = 14$). **c** SPs and CPs normalized to species average nucleotide identity (ANI) values (Large genome species SP/ANI $n = 16$, CP/ANI $n = 16$; Small genome species SP/ANI $n = 14$, CP/ANI $n = 14$). Statistical difference between categories was determined through two-sided Pairwise Wilcoxon rank-sum tests (SP/ANI P value = 6.987e-06; CP/ANI P value = 0.6374). The central line across the boxplots identifies the median, marking the dataset's midpoint. The box itself demarcates the interquartile range, extending from the first quartile to the third quartile, encapsulating the central 50% of the data. The whiskers project from the box to the furthest data points not categorized as outliers and show the spread of the main body of the dataset. Raw data is provided as a Source Data file.

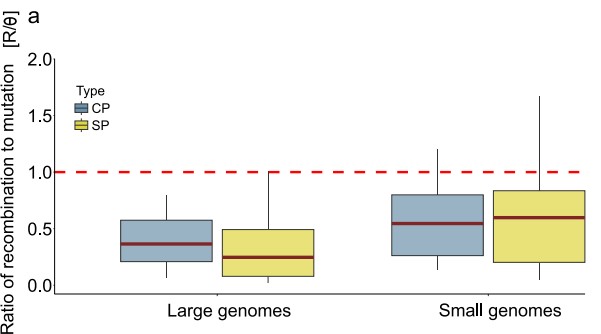

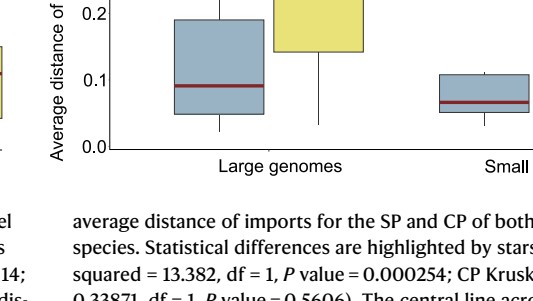

**Fig. 4 | Recombination to mutation rates measurements. a** The right panel depicts recombination to mutation rates for both large and small genomes (Large genomes SP $n = 16$, Large genomes CP $n = 16$; Small genomes SP $n = 14$; Small genomes CP $n = 14$). Two boxplots are presented for each category, distinguishing the secreted proteome (SP) in yellow and the cytoplasmatic proteome (CP) in blue. The red line serves as an indicator representing a ratio where recombination equals mutation. No statistical differences were found (SP Kruskal–Wallis chi-squared = 3.3456, df = 1, $P$ value = 0.06738; CP Kruskal–Wallis chi-squared = 1.6607, df = 1, $P$ value = 0.1975). **b** The left panel showcases the average distance of imports for the SP and CP of both large and small genomes species. Statistical differences are highlighted by stars (SP Kruskal–Wallis chi-squared = 13.382, df = 1, $P$ value = 0.000254; CP Kruskal–Wallis chi-squared = 0.33871, df = 1, $P$ value = 0.5606). The central line across the boxplots identifies the median, marking the dataset's midpoint. The box itself demarcates the interquartile range, extending from the first quartile to the third quartile, encapsulating the central 50% of the data. The whiskers project from the box to the furthest data points not categorized as outliers and show the spread of the main body of the dataset. Raw data is provided as a Source Data file.

geographic distribution among the smaller genome species (Fig. 3), showcasing increased niche adaptability. Therefore, the evolution of SPs emerges as a pivotal factor driving freshwater bacteria differentiation in both the analysed small and large genome species.

## Bacterial lifestyle strategies and ecophysiological traits

To evaluate the alignment of the observed SP conservation pattern with specific bacterial lifestyle strategies, we inferred ecophysiological characteristics by leveraging processes such as replication and genomic features, including codon usage[27–29]. Additional technical details about these approaches are provided in "Methods". The findings suggest that, during the sampling period, all species with large genomes were actively undergoing replication, as evidenced by elevated Growth Rate Index values (GRiD >2), indicative of rapid growth facilitated by multi-fork replication[27] (Fig. 8). It is important to note that in a population where the majority of bacteria are replicating, the Growth Rate Index would be equal to 2. In contrast, small-genome species generally exhibited slower replication rates, with exceptions like *Nanopelagicaceae sp.1*, which achieved a median value of 1.97, challenging the observed trend in other small-genome species. Additionally, the analysis of codon usage bias suggested that the majority of species have the potential for high growth rates (CUBHE > 0.6), even though these rates were not realized in situ, at least during our sampling collection times (Fig. 8).

Considering species genome sizes, in situ replication rates, and aligning taxonomy with known ecophysiological traits (when possible)[17,30], it appears that larger genome species tend towards a copiotrophic lifestyle, while their small genome counterparts exhibit characteristics indicative of an oligotrophic one. Copiotrophic bacteria, whether free-living or associated with lake snow particles, display substantial cell and genome sizes, providing metabolic and regulatory flexibility. This increased physiological adaptability allows nuanced responses to rapid nutrient changes through motility and chemotaxis (Supplementary Fig. 7), leading to peak growth rates during favourable conditions. In contrast, genome-reduced oligotrophic bacteria, with smaller cell and genome sizes, thrive in stable, nutrient-scarce, oligotrophic waters. Their growth strategy prioritizes efficiency over speed, maintaining a slow yet consistent pace[17]. Viewing the within-species diversity from an eco-evolutionary perspective it is likely that the upsurge in positively selected genes and heightened sequence diversity in the SPs of large genome species contributes to the copiotrophic lifestyle by broadening the potential range of utilizable substrates. In contrast, the restricted SP diversity in small genome species implies enhanced substrate fidelity. Diversifying the substrate range may not confer an advantage for oligotrophic bacteria, which rely on a reduced set of core metabolic pathways[30].

## The eco-evolutionary landscape of bacterial diversity

Our findings indicate that genes experiencing positive selection within the SPs of small genome species are remarkably scarce, and even the proportion of sites under negative selection is significantly diminished (Fig. 5b). These observations suggest that these bacteria likely achieved transient fitness peaks by attaining optimal protein topologies and preserving unaltered expression levels. The ecological success of genome-reduced bacterial species[31–33] is thus likely attributed to their exceptional efficiency in assimilating a limited number of substrates through a rather undiversified metabolic circuitry. The observed lack of significant evolutionary changes in the secreted proteomes of these organisms can be likely attributed to the notion that these proteomes have already reached an optimal state through the course of evolution, where further major alterations are neither advantageous nor necessary for the organisms' survival and adaptation to their current niches. However, the mechanisms underlying the evolutionary process that would allow these species to transition into and out of periods of adaptive stasis require further investigation.

The recovery of bacterial genomes from dominant freshwater populations has enabled the exploration of the evolutionary strategies and forces that govern bacterial lifestyles. While niche adaptation serves as the primary driving force behind the speciation process, our findings shed light on the prevalence of prolonged periods of adaptive stasis in abundant freshwater species characterised by small genome sizes[18,33]. During these phases, intrapopulation divergence in terms of genomic similarity is chiefly driven by mutation/recombination events and heightened positive selection acting on CP-encoding genes. The absence of functional redundancy and the increased elimination of diversity within the SPs of small genome species impede the emergence of evolutionary innovation. The inherent inflexibility within their SPs thus restricts the capacity of these organisms to explore novel genetic variations and effectively adapt to dynamic environmental conditions. Although we discovered the presence of adaptive stasis in several genome-reduced species (Supplementary Fig. 8 and Supplementary Data 7), further investigation is required to determine its prevalence among genome-reduced bacteria in diverse microbiomes.

The emergence of bacterial species is intricately linked to their capacity for ongoing evolution and adaptation within a dynamic ecological landscape. The SP plays a pivotal role in niche adaptation,

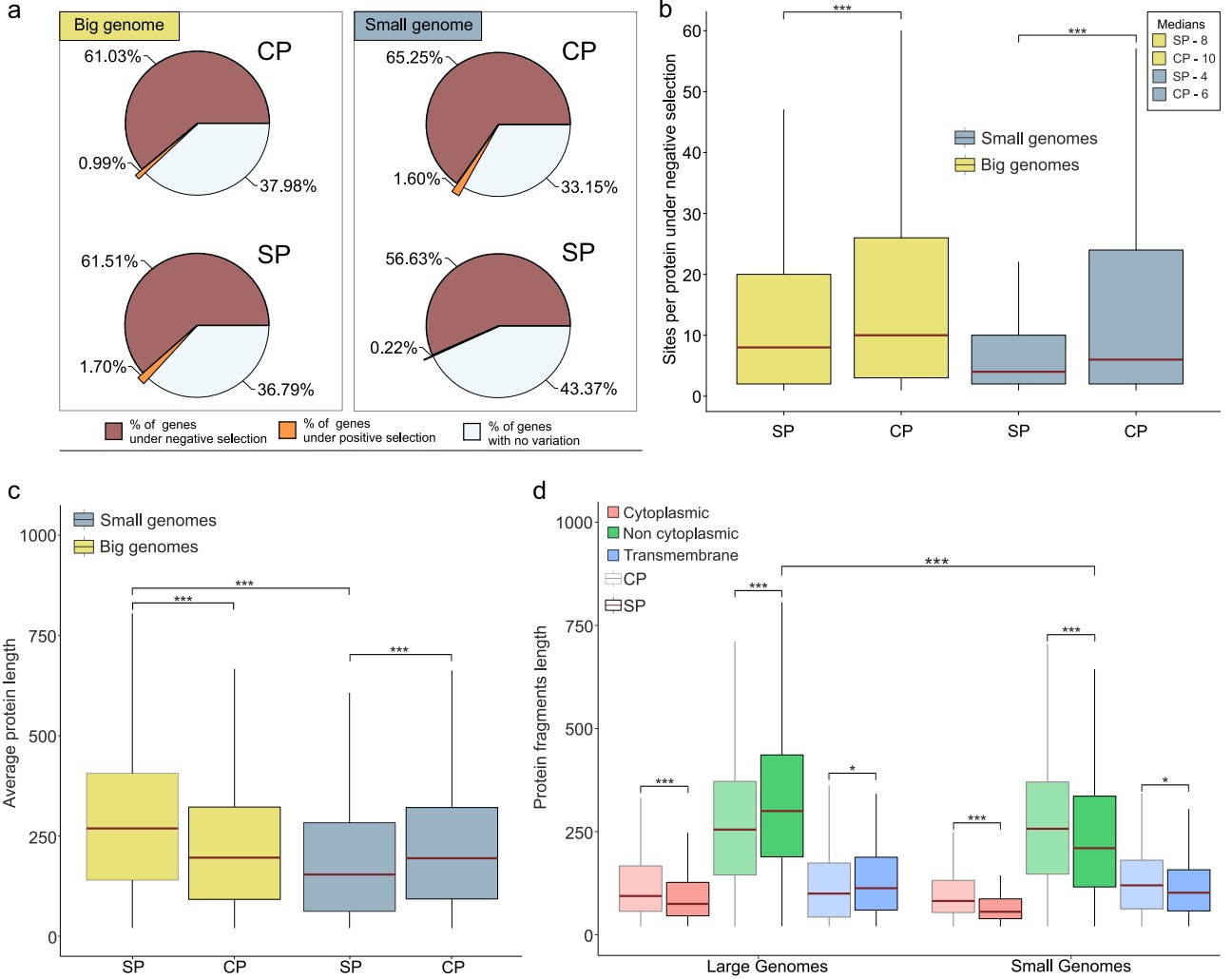

**Fig. 5 | Dynamics of selection forces. a**, **b** Genome-wide quantification of selection force for genes coding for SPs and CPs. **a** Percentages of genes under negative (red) and positive (orange) selection within the SPs and CPs. **b** Number of sites per gene under negative selection. The statistical difference within the predefined species categories (i.e., small and large species) (Large genomes SP $n = 1512$, Large genomes CP $n = 9390$; Small genomes SP $n = 512$; Small genomes CP $n = 6805$) was determined by Pairwise Wilcoxon rank-sum tests (Large genomes W = 7,774,646, $P$ value = 2.437e-09; Small genomes W = 2,090,672, $P$ value = 2.976e-14). **c** SPs and CPs length (aa). Average protein length was calculated for both SPs and CPs and compared within and between groups (Large genomes SP $n = 9147$, Large genomes CP $n = 32,288$; Small genomes SP $n = 12,602$; Small genomes CP $n = 1652$). Statistical significance within and between categories was determined through Pairwise Wilcoxon rank-sum tests ($P$ values < 2.2e-16 across all). **d** Protein subcellular localization fragments length for the SPs and CPs (Large genomes SP: Cytoplasmic $n = 916$, Non-cytoplasmic $n = 7313$, Transmembrane $n = 918$; CP: Cytoplasmic $n = 6185$, Non-cytoplasmic $n = 21,129$, Transmembrane $n = 4974$; Small genomes SP: Cytoplasmic $n = 319$, Non-cytoplasmic $n = 1011$, Transmembrane $n = 322$; CP: Cytoplasmic $n = 2220$, Non-cytoplasmic $n = 8573$, Transmembrane $n = 1809$). The overall difference was determined by Pairwise Wilcoxon rank-sum tests (Small genomes Cytoplasmic $P$ value = 6.353e-15, Small genomes Transmembrane $P$ value = 0.0035, Small genomes Non-cytoplasmic $P$ value = 7.805e-11; Large genomes Cytoplasmic $P$ value = 7.585e-10, Large genomes Transmembrane $P$ value = 0.0002306, Large genome Non-cytoplasmic $P$ value < 2.2e-16; Large and Small genome Non-cytoplasmic $P$ value < 2.2e-16). The central line across the boxplots identifies the median, marking the dataset's midpoint. The box itself demarcates the interquartile range, extending from the first quartile to the third quartile, encapsulating the central 50% of the data. The whiskers project from the box to the furthest data points not categorized as outliers and show the spread of the main body of the dataset. Raw data is provided as a Source Data file.

facilitating crucial functions such as nutrient recognition and uptake, inter-bacterial communication, surface attachment, and signal transduction. Given that adaptation predominantly occurs through the fixation of nonsynonymous mutations[34], the conspicuous absence of such variations in the SPs of genome-reduced freshwater species implies a stalling of the adaptive processes. Delving into the nuanced mechanisms governing SP diversity has provided profound insights into the eco-evolutionary processes shaping copiotrophic and oligotrophic lifestyle strategies. Rather than constraining these strategies to a rigid dichotomy, it is crucial to perceive them along a continuum, accommodating a spectrum of adaptations and behaviours. Despite these insights, it is vital to acknowledge the constraints associated with

capturing the diversity of uncultured bacterial species, as discussed in the manuscript's Limitations section.

Considering that bacterial species with small genome sizes frequently form substantial components of freshwater environments and fulfil vital ecological functions, comprehending their adaptability to changing climatic conditions is of paramount importance for ecosystem resilience and sustainable resource management.

## Limitations
While our sampling efforts spanned the same habitat types, freshwater lakes, and encompassed multiple time points, we acknowledge the inherent challenge of capturing the entirety of genomic diversity

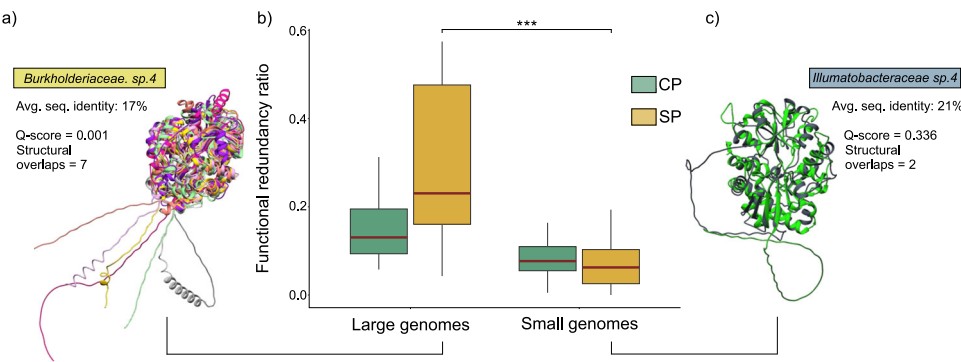

**Fig. 6 | Proteome functional redundancy. a, c** Structural alignments of the substrate binding protein (K02051) belonging to the NitT/TaUt family transport system. Each colour corresponds to one protein. These panels illustrate the levels of functional redundancy observed in the SPs of large (**a**) and small (**c**) genome species. The degree of structural similarity is quantified using the Q-score, while the number of overlaps indicates the aligned proteins (seven in (**a**), and two in (**c**)). Burkholderiaceae sp. 4 and Ilumatobacteraceae sp. 4 were selected as representatives of the large and small genomes species, respectively, due to their maximum number of NitT/TaUt substrate-binding proteins per species. **b** Analysis of functional redundancy ratios in CPs and SPs (Large CP $n = 192$; SP $n = 192$; Small CP $n = 150$; SP $n = 150$). Wilcoxon rank-sum tests were conducted to compare values within the SP category (P value < 2.2e-16). The central line across the boxplots identifies the median, marking the dataset's midpoint. The box itself demarcates the interquartile range, extending from the first quartile to the third quartile, encapsulating the central 50% of the data. The whiskers project from the box to the furthest data points not categorized as outliers and show the spread of the main body of the dataset. Raw data is provided as a Source Data file.

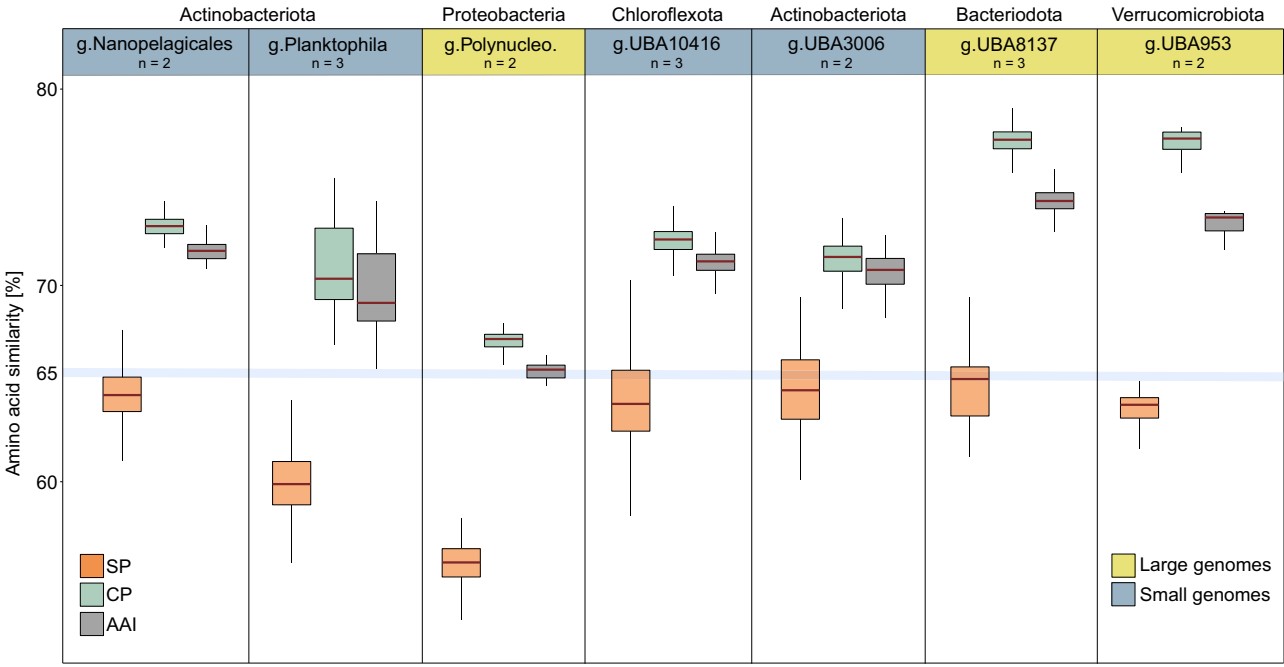

**Fig. 7 | Investigation of proteome similarity within genus-level classifications.** The upper panel displays the taxonomic labels of seven identified genera containing at least two species (out of the 30 analyzed species clusters), where species with small and large genomes are shown in blue and yellow, respectively. The Y axis shows the average amino acid similarity values for SPs (in orange), CPs (in green), and whole proteomes (in grey) within each genus (g.Nanopelagicales $n = 110$; g.Planktophila $n = 379$; g.Polynucleobacter $n = 171$; g.UBA10416 $n = 404$; g-UBA3006 $n = 99$; g.UBA8137 $n = 455$; g.UBA953 $n = 108$). The blue line represents the average amino acid similarity value of 65% used to define genus-level categories. The central line across the boxplots identifies the median, marking the dataset's midpoint. The box itself demarcates the interquartile range, extending from the first quartile to the third quartile, encapsulating the central 50% of the data. The whiskers project from the box to the furthest data points not categorized as outliers and show the spread of the main body of the dataset. Raw data is provided as a Source Data file.

within species. The impediments stem from methodological limitations that are exacerbated by the continuous evolution of bacterial populations over time. To address some of the mentioned challenges, our study employed a strategic approach that entailed repeated sampling campaigns in the selected environments. This iterative sampling method significantly bolstered our capacity to capture some of the species flexible genomes. To enhance our capability to investigate the flexible genome, we have meticulously curated our pdCEL database,

prioritising species clusters with a minimum of 9 representatives. However, it is important to acknowledge that the recovery of partial genomes (median completeness = 77.03%) may impose limitations on the extent of within-species diversity that was recovered. Furthermore, it is important to note that the metagenome-assembled genomes employed in this study serve as 'genomic pools' representing abundant populations with high sequence identity. However, they may not precisely capture the genomic constitution of specific clonal lineages, nor

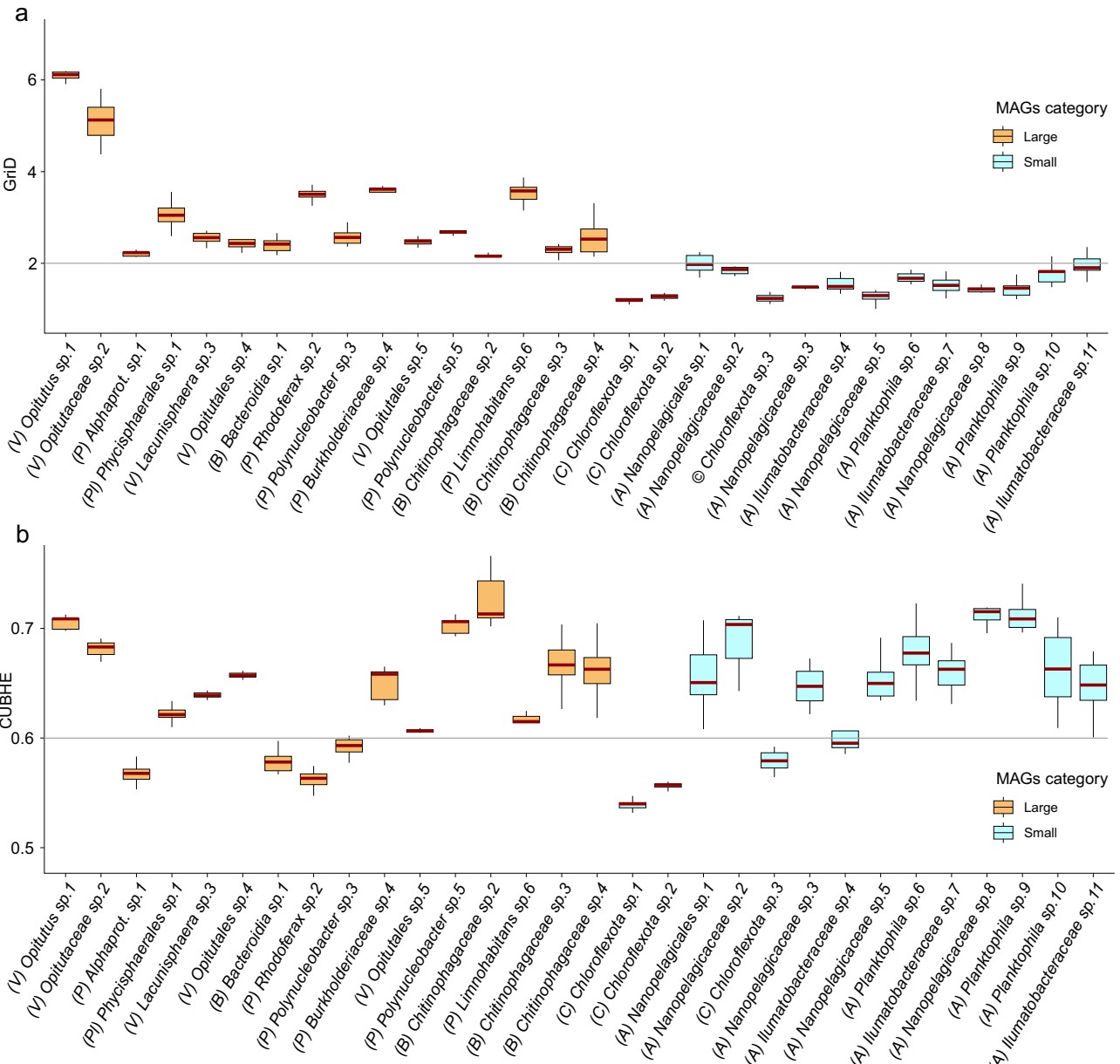

**Fig. 8 | Genome-based ecophysiological inferences. a** The upper panel displays Growth Rate Index (GRiD) values of the obtained 30 species clusters ((A) *Ilumato-bacteraceae sp.11 n* = 9; (A) *Ilumatobacteraceae sp.4 n* = 10; (A) Ilumatobacteraceae sp.7 *n* = 11; (A) *Nanopelagicaceae sp.2 n* = 10; (A) *Nanopelagicaceae sp.3 n* = 11; (A) *Nanopelagicaceae sp.5 n* = 12; (A) *Nanopelagicaceae sp.8 n* = 9; (A) *Nanopelagicales sp.1 n* = 9; (A) *Planktophila sp.10 n* = 9; (A) *Planktophila.6 n* = 14; (A) *Planktophila sp.9 n* = 11; (B) *Bacteroidia sp.1 n* = 17; (B) *Chitinophagaceae sp.2 n* = 13; (B) *Chit-inophagaceae sp.3 n* = 13; (B) *Chitinophagaceae sp.4 n* = 11; (C) *Chloroflexota sp.1 n* = 11; (C) *Chloroflexota sp.2 n* = 14; (C) *Chloroflexota sp.3 n* = 10; (P) *Alphaprot. sp.1 n* = 11; (P) *Burkholderiaceae sp.4 n* = 9; (P) *Limnohabitans sp.6 n* = 10; (P) *Poly-nucleobacter sp.3 n* = 19; (P) *Polynucleobacter sp.5 n* = 9; (P) *Rhodoferax sp.2 n* = 10; (Pl) *Phycisphaerales sp.1 n* = 17; (V) *Lacunisphaera sp.3 n* = 11; (V) *Opituaceae sp.2 n* = 12; (V) *Opitutales sp.4 n* = 9; (V) *Opitutales sp.5 n* = 12; (V) Opitutus sp.1 *n* = 9), where species with large and small genomes are represented with orange and light blue colors, respectively. The grey line serves as an indicator for GRiD values sur-passing 2. **b** The bottom panel shows codon usage bias of highly expressed genes (CUBHE). As for the upper panel, species containing large genomes are depicted in orange, while species containing small genomes are in light blue ((A)

*Ilumatobacteraceae sp.11 n* = 9; (A) *Ilumatobacteraceae sp.4 n* = 10; (A) Ilumato-bacteraceae sp.7 *n* = 11; (A) *Nanopelagicaceae sp.2 n* = 10; (A) *Nanopelagicaceae sp.3 n* = 11; (A) *Nanopelagicaceae sp.5 n* = 12; (A) *Nanopelagicaceae sp.8 n* = 9; (A) *Nano-pelagicales sp.1 n* = 9; (A) *Planktophila sp.10 n* = 9; (A) *Planktophila sp.6 n* = 14; (A) *Planktophila sp.9 n* = 11; (B) *Bacteroidia sp.1 n* = 17; (B) *Chitinophagaceae sp.2 n* = 13; (B) *Chitinophagaceae sp.3 n* = 13; (B) *Chitinophagaceae sp.4 n* = 11; (C) *Chloroflexota sp.1 n* = 11; (C) *Chloroflexota sp.2 n* = 14; (C) *Chloroflexota sp.3 n* = 10; (P) *Alphaprot. sp.1 n* = 11; (P) *Burkholderiaceae sp.4 n* = 9; (P) *Limnohabitans sp.6 n* = 10; (P) *Poly-nucleobacter sp.3 n* = 19; (P) *Polynucleobacter sp.5 n* = 9; (P) *Rhodoferax sp.2 n* = 10; (Pl) *Phycisphaerales sp.1 n* = 17; (V) *Lacunisphaera sp.3 n* = 11; (V) *Opituaceae sp.2 n* = 12; (V) *Opitutales sp.4 n* = 9; (V) *Opitutales sp.5 n* = 12; (V) Opitutus sp.1 *n* = 9). The grey threshold line is positioned at a CUBHE value of 0.6. The central line across the boxplots identifies the median, marking the dataset's midpoint. The box itself demarcates the interquartile range, extending from the first quartile to the third quartile, encapsulating the central 50% of the data. The whiskers project from the box to the furthest data points not categorized as outliers and show the spread of the main body of the dataset. Raw data is provided as a Source Data file.

fully represent the genetic diversity within populations that are highly similar yet of low abundance. An additional limitation to consider is that MAGs generated from short-read sequencing may potentially overlook variable genomic regions and mobile genetic elements, further complicating the comprehensive capture of microbial diversity.

Phobius, a bioinformatics tool utilised in this study, employs a combination of algorithms to predict signal peptides in bacterial proteins by analysing the N-terminus regions. While this approach offers significant value, especially in terms of ease of use and scalability, it has inherent limitations, as not all the cytoplasmic and signal peptide-bearing proteins may be correctly identified. In our benchmark analysis, Phobius correctly identified approximately 94.17% of the proteins used for testing. It is crucial to acknowledge, however, that the classification of proteomes and their sequences as secreted or cytoplasmic was based solely on computational predictions without confirmation from additional mass spectrometry-based experimental techniques.

Functional redundancy analysis involved the identification of proteins sharing identical functions. In this study, we opted for an annotation-based approach that is less sensitive to protein similarity levels or their phylogeny. The percentage of annotated proteins, as per the KEGG database, was 51.19% for SP and 69.36% for CP in large genome species, while in small genome species, it was 62.3% for SP and 76.26% for CP.

## Methods

### Sampling sites and DNA isolation

Samples from five freshwater lakes (that range in trophic status from oligotrophic to eutrophic; the Czech Republic and Switzerland) were used to recover genomic information from prokaryotes colonising diverse freshwater niches.

Římov Reservoir (470 m a.s.l., 48°50′N, 14°29′E, Czech Republic) is a meso-eutrophic, canyon-shaped dimictic water body with an area of 2.0 km$^2$ (length of 13.5 km, volume of $34.5 \times 106$ m$^3$, mean water retention time of 77 days, maximum depth of 43 m) that was built during 1974–1979 by damming a 13.5 km long section of the River Malše. The sampling was performed between June 2015 and August 2017, above the deepest point of the reservoir by using a Friedinger sampler. In all, 20 L of water were collected from 0.5 ($n = 10$) and 30 m ($n = 8$) depths and subjected to sequential peristaltic filtration through a series of 20, 5, and 0.2-µm-pore-size polycarbonate membrane filters (⌀ 142 mm) (Sterlitech Corporation, USA). The sample collection and filtration steps were similar for the rest of the lakes/pools unless otherwise stated. Jiřická pond (892 m a.s.l., 48°36.96′N 14°40.59′E, Czech Republic) is a dystrophic humic water body with an area of 0.035 km$^2$ (volume of $6.59 \times 103$ m$^3$, mean water retention time of 9 days, maximum depth of 3.7 m), located in the Novohradské mountains of Southern Bohemia. Fifteen epilimnia (0.5 m depth) water samples were collected between May 2016 and August 2017. Lake Zurich (406 m a.s.l., 47°18′N, 8°34′E, Switzerland) is an oligomesotrophic, perialpine monomictic water body, with an area of 67.3 km$^2$ (length of 40 km, volume of 3.3 km$^3$, mean water retention time of 1.4 years, maximum depth of 136 m). Thirteen samples were collected between 2013 and 2019 from the epilimnion (5 m depth, $n = 8$) and hypolimnion (80/120 m depth, $n = 5$) layers, and processed as described above. Lake Thun (558 m a.s.l., 46°41′N, 7°43′E, Switzerland) is an oligotrophic, alpine water body with an area of 48.3 km$^2$ (length of 17.5 km, volume of 6.5 km$^3$, mean water retention time of 1.8 years, maximum depth of 217 m). Two water samples were collected in June 2018 from 5 and 180 m depths. Lake Constance (395 m a.s.l., 47°32′N, 9°31′E, Swiss Confederation) is an oligotrophic perialpine lake with an area of 473 km$^2$ (length of 63 km, volume of 48 km$^3$, mean water retention time of 5 years, maximum depth of 252 m). Four samples were collected in July and October 2018 from 5 m and 200 m depths.

DNA was extracted from the 0.22-µm filters (0.2- to 5-µm fraction) using the ZR Soil Microbe DNA MiniPrep kit (Zymo Research, Irvine, CA, USA; cat # D6010) following the manufacturer's instructions. The total quantity of DNA was estimated using the Qubit dsDNA BR assay kit (Thermo Scientific, Waltham, MA, USA; cat # Q32850) on a Qubit 2.0 fluorometer (Life Technologies). DNA integrity was assessed by agarose gel (1%) electrophoresis and SYBR green I stain (Thermo Scientific, Waltham, MA, USA; cat # S7563. Shotgun sequencing was performed using the Novaseq 6000 sequencing platform ($2 \times 150$ bp) (Novogene, Hong Kong, China).

### Assembly, binning, and MAG designation

Raw Illumina metagenomic reads were pre-processed to remove low-quality bases/reads and adaptor sequences using the BBMap[35] v36.1 package. Briefly, PE reads were interleaved by reformat.sh[36] and quality trimmed by bbduk.sh[37] (qtri =rl trimq=18). Subsequently, bbduk.sh was used for adapter trimming and identification/removal of possible PhiX and p-Fosil2 contamination (k = 21 ref=vectorfile ordered cardinality). Additional inspections (i.e., de novo adapter identification with bbmerge.sh[38] were performed to ensure that the datasets met the quality threshold necessary for assembly. The pre-processed reads were assembled independently with MEGAHIT[39] v1.1.5 using the k-mer sizes: 39 49,69,89,109,129,149, and default settings. The pre-processed metagenomic datasets were mapped using bbwrap.sh[40] (kfilter=31 subfilter=15 maxindel=80) against the assembled contigs (longer than 3 Kbp) in a sample-dependent fashion, ensuring that each metagenomic dataset was specifically mapped against all assemblies. The resulting BAM files were utilised to generate contig abundance profiles with jgi_summarize_bam_contig_depths[41] (--percentIdentity 97). The contigs and their abundance files were subsequently used for hybrid binning with MetaBAT2[41] (based on tetranucleotide frequencies and coverage data; default settings). MetaBAT2 was selected based on its performance as the best-performing single binner in independent benchmarks[42] and its efficiency, being 10 to 50 times faster than other commonly used binners[43].

Post-binning curation was achieved by applying a taxonomy-based approach coupled with a GC cut-off. Briefly, the predicted proteomes (PRODIGAL[44] v2.6.3) of individual bins were queried (using mmseqs[45] search) against the curated prokaryotic GTDB database (R05-RS95)[46]. The obtained results were further converted into a BLAST-tab formatted file (using mmseqs convertalis) from which individual top hits (cut-offs: E-value 1e-3, identity 10%, coverage 10%, bitscore 50) were extracted, and their taxonomic labels used to annotate the queried proteomes. Taxonomic information was then used to classify each bin at the class level and discard individual constituent contigs for which taxonomic homogeneity was not achieved (more than 30% of the taxonomic labels belonged to a different class). Contigs without taxonomy information or those for which the GC content deviated by more than 15% from the bin median value, were discarded as well. Bin completeness, contamination, and strain heterogeneity were estimated by CheckM[47] v1.1.3 (using the lineage_wf workflow). Bins with estimated completeness above 40% and contamination below 5% were denominated as metagenome-assembled genomes (MAGs). All the obtained MAGs were taxonomically classified with GTDB-Tk[48] v1.4.0 (database release R05-RS95) using default settings.

To determine the estimated genome size, the MAG length was divided by its estimated completeness, and the resulting figure was multiplied by the difference between 100 and the MAG contamination value.

### Species denomination and intraspecific similarity comparisons

To establish species clusters, we used the previously obtained classification information to create groups that were taxonomically-homogenous. We further compared genome-wide nucleotide

identities within each pre-established group and identified MAG clusters that shared over 95% sequence similarity (with over 70% conserved DNA). These clusters were considered to belong to the same species, in line with accepted species delineation criteria for bacteria[20,49]. To calculate average nucleotide identities (ANI), we employed a previously described method[50]. This involved fragmenting the MAGs into 1 200 bp nucleotide fragments and performing reciprocal nucleotide BLAST searches (blastn) (for additional information see Goris and colab[50]). We set the maximum allowed gap size to 150 nucleotides and turned off low complexity sequence filtering (-F F) to identify reciprocal best hits. Species clusters that had ≥9 MAGs and ≥95% ANI ($n = 30$), were used for downstream analyses.

The identification of protein-coding genes was carried out using PRODIGAL[44] v2.6.3, employing its default settings. Phobius[51] v1.01 (-long option) was utilised to determine the presence of signal peptides in obtained proteomes. Proteins containing a signal peptide were assigned to the secreted proteome (SP), whereas the remaining proteins were assigned to the cytoplasmic one (CP). Further, the proteomes of every species group ($n = 30$) were partitioned into cytoplasmic and secreted subsets.

Phobius, a bioinformatics tool applied in this study, employs a combination of algorithms to predict signal peptides in bacterial proteins by recognizing and analysing the N-terminus regions. This methodology is commonly employed to predict protein localizations not only in well-known bacterial species[52,53] but also in uncultured microbial dark matter[54]. It is worth mentioning that Phobius is frequently integrated into prominent protein annotation databases and toolkits[55,56], underscoring its established reliability in predicting protein subcellular localisation. We benchmarked our approach for SP and CP identification, by analysing 120,300 cytoplasmic proteins, 10,189 signal peptide-bearing proteins, and 3156 periplasmic proteins from the reviewed UniProtKB/Swiss-Prot protein database. We took rigorous steps to ensure that the proteins selected had a clear association with bacteria in this curated, high-quality database. Phobius software, employing default parameters, was utilised for predicting signal peptides in all the aforementioned categories. By quantifying the number of sequences with incorrect predictions, the accuracy rate can be calculated using the following equation: Phobius accuracy = ($N$ of correct predictions)/($N$ of total predictions) × 100. The software demonstrated high accuracy, correctly identifying 97.66% of cytoplasmic proteins (CPs), 93.61% of signal peptide-bearing proteins (SPs), and 91.25% of periplasmic proteins (SPs) (Supplementary Data 9). These findings align with the accuracies reported in previous studies[57], reinforcing our confidence in the software's performance.

Average amino acid identity values were computed for the cytoplasmic and secreted proteomes of each species, employing the methodology detailed by Konstantinidis and Tiedje[58]. This approach involves performing reciprocal whole-proteome blastP comparisons, where a protein dataset (i.e., the SP and CP of a genome) is comprehensively compared against another within a species cluster. Thus, each proteome will be compared against all other proteomes present in the designated species, capturing pairwise similarities. Our implementation specifically utilised the blastP algorithm for bidirectional protein sequence comparisons. The percentage of proteins utilised for all-versus-all proteome comparisons is detailed for each species in Supplementary Information. It is noteworthy that the median percentage of proteins compared is 73.13% for SP and 75.46% for CP, aligning with the species pangenome concept, accounting for the anticipated presence of some genome-specific proteins.

## Genome annotations

Protein-coding genes were predicted with PRODIGAL[44] v2.6.3. Protein domains were annotated by querying the obtained proteomes (using the hmmscan-based 'pfam_scan.pl' script) against the HMM database present in Pfam[59] release 32. Additional domain architectures and protein annotations were performed by running InterProScan[55] (v5.24-63.0) with the databases CDD (v3.14), SMART (v7.1), and HAMAP (v201701.18), respectively. Protein annotation space was further enlarged by running hmmsearch (-evalue 1E-7 -prcov 70 -hmcov 70) against COGs[60] and TIGRFAM[61] HMM databases. BlastKOALA[62] was used to assign KO identifiers to orthologous genes. Inferences of complete secretion systems and flagella were conducted with the online KEGG mapping tools using summarised KO numbers, in a presence-absence fashion.

## Within-species diversity generation

To evaluate recombination and mutation rates within the secreted (SP) and cytoplasmic proteomes (CP) of the 30 species clusters, we employed a methodology previously delineated by Didelot and Wilson, 2015[63]. In summary, gene sequences from both SP and CP were extracted in DNA space, concatenated, and subsequently aligned using progressiveMauve[64] v2.3.1 with default settings. Core alignments longer than 500 bp were then extracted using the stripSubsetLCB script. The obtained core alignments were utilised to construct maximum-likelihood trees with PhyML[65] v3.3.3, utilising the options --datatype nt -p --bootstrap 100 --model HYK85 -f m -t e --alpha e --quiet --leave_duplicates. The resulting phylogeny, along with transition/transversion ratios, was employed to assess recombination/mutation rates with ClonalFrameML[63] v1.12 (-kappa transition/transversion ratio). To enhance the robustness of the analysis, 100 replicates per alignment were conducted using the --emsim 100 option (Supplementary Data 10). The method was validated using three Prochlorococcus and one Pelagibacter species, for which Metagenome-assembled genomes (MAGs), single-amplified genomes (SAGs), and isolates (in the case of Prochlorococcus) were accessible (Supplementary Data 11). Our findings demonstrated concordance in the data obtained from MAGs when compared to SAGs and isolates (available for one analysed Prochlorococcus species). These results are consistent with a recent study that utilised ClonalFrameML to evaluate recombination effects in MAGs recovered from ammonia-oxidizing archaea[66].

## Protein fragments length

Orthologous protein groups were determined using the OrthoFinder v.2.5.2[67] software (-I 3 -S blast). Subsequently, groups containing a minimum of eight sequences were chosen for further analysis to identify the subcellular localization of protein fragments (i.e., cytoplasmic, non-cytoplasmic, and transmembrane regions). The identification of specific protein regions was accomplished by utilising the output from Phobius software with a custom R script[68]. Following this, the bedtools v2.27.1 software[69] was employed to extract the identified specific regions of the proteins. These retrieved regions were then concatenated based on their subcellular localisation and used for length determination.

## Selection force

The identified orthologous protein groups (342 MAGs grouped in 30 species) were further scrutinized using HMMER[70] hmmscan v3.1b2 with an $E$-value threshold of 1E-7, and -prcov 70 and -hmcov 70 against a locally installed TIGRFAMs[61] v15.0 database. The resulting annotation data was then used to refine the orthologous groups. The recovered orthologous gene sequences in nucleotide space were further subjected to within-species codon alignments using prank[71] v.170427 (-codon -F). For downstream analyses, only species-level homologous gene groups containing three or more sequences were considered. The alignments obtained were employed to construct phylogenetic trees utilising IQ-TREE[72] v 2.1.3. (-st CODON). Site-specific selection pressure was inferred using the codon alignments and phylogenetic trees in a maximum-likelihood framework ($P$ value threshold 0.05) through FEL[73] v2.1 software implemented in HYPHY 2.5.32. Overall,

around 29,000 homologous genes within the 30 species clusters underwent an assessment of selection pressure.

## Functional redundancy ratios and protein structural alignments

The Functional Redundancy Ratio (FRR) was computed as a quantitative measure to evaluate the percentage of genes encoding redundant functionality within the previously identified SPs and CPs of selected bacterial species. The calculations involved several steps: (i) determining the total count of KEGG annotations per MAG, (ii) obtaining the unique counts of KEGG[74] annotations per MAG, (iii) calculating the FRR by dividing the number obtained in step (ii) by the total obtained on step (i) and iv) implementing a one-minus-ratio transformation to enhance the intuitiveness of the interpretation. A FER closer to 0 indicates a very low occurrence of duplicated functional annotations, while one close to 1 suggests very high functional redundancy. The percentage of utilised KEGG annotations for large-genome species was 51.19% for SP and 69.36% for CP, while for small-genome species, it was 62.3% for SP and 76.26% for CP[67].

A protein structural analysis was carried out on the NitT/TauT family transport system substrate-binding proteins (K02051) as they were found present in most studied species groups. Protein structures and complex prediction were carried out using ColabFold v1.5.2[75] with default settings. Predicted protein structure model accuracy was assessed using the percentage of the Inter-residue Distance Difference Test (pIDDT). The pIDDT measures the inter-residue distances in the predicted model deviating from the experimentally determined reference structure. The models with the highest pIDDT values were selected for subsequent analysis. Protein structural alignments and overlaps were conducted with the MatchMaker function within Chimera 1.17.1 software[76] with default settings. Overlap similarity quantification was performed with the Match -> Align function, where Q-score values were selected as a measurement of similarity. Briefly, the Q-score estimates the degree of structural similarity between protein structures by comparing the internal distances, specifically the positions of the Cα atoms. A higher Q-score value indicates greater similarity between the compared structures and ranges between 0 and 1[77].

## Bacterial growth rate estimates

To correlate species with general lifestyle strategies, we employed GRiD[27] v1.3 and the gRodon2[28] R package (in R v4.1.1) for estimating in situ growth and maximal growth rates, respectively.

GRiD, a versatile tool designed for precise estimation of microbial growth rates, excels in analysing both complete/draft genomes and metagenomic bins, especially under ultra-low sequencing coverage (> 0.2×). Its adaptability is underscored by its ability to function without requiring prior knowledge of microbial composition and coverage. GRiD operates by calculating contig coverage, organising them to simulate a circular genome, and refining growth values through statistical filters. The resulting GRiD values serve as indicators of microbial growth rates, where a higher ratio denotes faster growth. In this study, GRiD was executed in 'single' mode with default settings[27,78].

gRodon is a powerful package designed for predicting the maximal growth rate of prokaryotic organisms based on genomic data. Leveraging codon usage statistics, gRodon identifies the optimisation of highly expressed genes, serving as a robust indicator of selection for rapid growth. Prokka[79] v.1.13 (default options) was used to predict coding DNA sequences from the 342 MAGs within the 30 species clusters. Obtained untranslated coding sequences (CDS) within the.ffn files as well as the CDS names from the.gff files (extracted as follows: sed -n '/##FASTA/q;p' my_genome/my_genome.gff | awk '$3 = = "CDS"' | awk '{print $9}' | awk 'gsub(";.*","")' | awk 'gsub("ID = ","")' > CDS_names.txt) were used as input within the gRodon2 script. Codon pair bias (CPB) as well as the codon usage bias of highly expressed genes (CUBHE) were used as a unit of measure for growth potential.

## Pan-genome analysis

To assess the pan-genome of the 30 analysed species-clusters, the rapid large-scale prokaryote pan genome pipeline Roary[80] v.3.13.0 was used. More specifically, the MAGs in each species cluster were annotated using Prokka v.1.13 enabling the --metagenomes flag. The resulting annotations were supplied to Roary as a set of.gff (general feature format) files per species cluster. Roary generated a core alignment using the mafft[81] aligner with 10 iterations to assess the core (present in 99% - 100% of the strains) and the pan genes (present in 0% - 100% of the strains) within each species cluster using default options. For each species, two R plots were created based on the 'number_of_new_genes.Rtab' and the 'number_of_genes_in_pan_genome.Rtab' output files created by the 'roary-create_pan_genome_plots.R' script from the Roary tool, showing the number of new genes and the total number of genes per added MAG to the pangenome.

## Microdiversity analyses

Considering the expected microdiversity within most species, we intentionally chose one metagenome, namely RH-22June17, to investigate the species-level microdiversity of the metagenome-assembled genomes (MAGs) derived from this metagenome. This sample allowed us to investigate the microdiversity of 23 out of the 30 species clusters discussed in the manuscript. From the resulting 7 species clusters, no MAG could be recovered from this respective sample.

inStrain[78] is an advanced program crafted for profiling intrapopulation genetic diversity, commonly referred to as microdiversity, across complete genomes. This tool leverages metagenomic paired reads to conduct a thorough analysis of microbial populations, adopting a microdiversity-aware methodology. The standard use involves the generation of a BAM file by mapping metagenomic reads to a bacterial genome within the sample, with inStrain then employed for a comprehensive characterisation of the microdiversity present. We utilised the inStrain software to unravel population microdiversity and perform comparisons at both secreted proteomes (SPs) and cytoplasmic proteomes (CPs) levels, following established protocols[78]. In brief, quality-filtered Illumina shotgun reads were mapped against the metagenome-assembled genomes (MAGs) using the bbmap[35] software. Subsequently, a scaffold-to-bin file was generated using the 'parse_stb.py' script recommended by the dRep[82] software. Gene prediction was executed with PRODIGAL, utilising default settings (Supplementary Data 12).

To identify distinct populations within the same species, we subsampled one metagenome (i.e., RH-22June17) to 100 million sequences and exclusively utilised MAGs derived from same dataset (belonging to 23 species). Establishing a one-to-one correspondence between MAGs and contigs through concatenation, we employed blastn to compare the subsampled dataset (100 million sequences) against MAGs. Criteria included an alignment length of at least 100 nucleotides and an E value ≤ 1e-5, generating output files formatted to '-m 8'. The results of this approach were utilised to create metagenomic recruitment plots, taking advantage of the best-hit distributions across the MAGs.

## Benchmarking

We conducted an SP/CP benchmark analysis using an abundant environmental marine species, *Prochlorococcus_B marinus_B* (Supplementary Fig. S3). This species was chosen due to the availability of cultured representatives, single-cell amplified genomes (SAGs), and metagenome-assembled genomes (MAGs). By comparing the SPs/CPs ratio among genomes from cultures, SAGs, and MAGs, it becomes evident that the results are indeed comparable, with cultures and MAGs yielding highly similar outcomes. These results were further supported by the absence of statistical difference (ANOVA; df = 2; f-value = 0.3852; *P* value = 0.6811) in the SPs/CPs ratio among the three groups (i.e., culture, SAGs and MAGs). The small deviation observed in

the SAGs can likely be attributed to errors introduced during the single-cell amplification steps.

## Statistics and reproducibility

All statistics were performed within the R[83] v4.0.3 and RStudio[84] v1.3.1093, Apricot Nasturtium software. Data normality was assessed by the Shapiro–Wilk test, followed by residues distribution visualisation. Simple comparisons between two datasets were performed by student t-test for normal distributed data and Wilcoxon rank-sum test for non-normal datasets. Parametric analysis of variance (ANOVA) was performed for normally distributed datasets followed by multiple pairwise comparisons with Tukey's test. Non-parametric datasets analysis of variance was performed by using the Kruskal–Wallis test followed by pairwise comparisons with pairwise Wilcoxon rank-sum tests. For non-normal samples, the correlation index was assessed using the Pearson-rank correlation test. Variable interaction as well as multiple correlations were assessed by multiple regression models. Discussed tests were performed using the corresponding functions within the stats v.4.1.3 package. The stats package is part of R.

In an attempt to assess the effect of the lake diversity on the SPs/CPs ratio within each species cluster, we investigated the relationship between the SPs/CPs ratio and the different predictor variables (i.e., Coding density and lake diversity). Linear models were fitted to the data using the lm function within R. Three models (i.e., M1, M2 and M3) were formulated as follows, (i) simple linear regression with coding density as sole predictor (M1), (ii) simple linear regression with lake diversity as sole predictor (M2) and (iii) interaction model between coding density and lake diversity (M3). Models were evaluated using a dual approach, considering both the Akaike Information Criterion (AIC) and the significance predictor within the model.

No statistical method was used to predetermine sample size. No data were excluded from the analyses. The experiments were not randomized. The Investigators were not blinded to allocation during experiments and outcome assessment.

## Reporting summary

Further information on research design is available in the Nature Portfolio Reporting Summary linked to this article.

## Data availability

All sequence data generated during this study have been deposited in the EBI/NCBI (Bioprojects: PRJEB35770, PRJEB35640, PRJNA428721, PRJNA429145). The accession numbers for the 52 raw metagenomic datasets are listed in Supplementary Data 1. The 5519 MAG IDs, their accession numbers, Bioproject IDs, and Sample IDs, along with additional metadata, are provided in Supplementary Data 2. Data supporting the conclusions of this study was deposited in Figshare under the following link https://doi.org/10.6084/m9.figshare.23546067. All additional important data supporting the study's conclusions are included in this publication and its Supplementary Data 3–13. Source data are provided with this paper.

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

## Acknowledgements

We are grateful for the support from Maliheh Mehrshad, Tanja Shabarova, Michaela Salcher, Petr Znachor, Pavel Rychtecký, Petr Porcal, and the Institute of Hydrobiology of the Czech Academy of Sciences for their assistance with sampling in Jiřická Pond and Rimov Reservoir. Our thanks also go to Eugen Loher and the Canton of Bern's Laboratory for Water and Soil Protection (GBL) for facilitating Lake Thun sampling, as well as to Thomas Posch and the crew of the Kormoran research vessel (Institute for Lake Research of the State Agency for Environment Baden-Württemberg) for their help with Lake Constance sampling. Special appreciation is extended to Vinicius Kavagutti, Cecilia Chiriac, Rohit Ghai, Alizée Le Moigne, and Angel Rain-Franco for their insightful discussions and advice. A.-S.A. and L.S.M. were supported by the Ambizione grant PZ00P3_193240 (Swiss National Science Foundation). P.-A.B. was supported by research grant 20-12496X (Grant Agency of the Czech Republic). J.P. was supported by the research grant 10000877 (Swiss National Science Foundation).

## Author contributions

Conceptualization: A.-S.A., L.S.M., and J.P. Bioinformatics: A.-S.A., L.S.M., C.H., and P.-A.B. Statistics: L.S.M. Writing, original draft: A.-S.A. with input from L.S.M. Writing, review, and editing: L.S.M, C.H., P.-A.B., and J.P. Funding acquisition: A.-S.A.

## Competing interests

The authors declare no competing interests.
