## [Peer Review File · Nature Communications]

REVIEWER COMMENTS

Reviewer #1 (Remarks to the Author):

This work investigates the adaptive capacity of aquatic (freshwater mostly) bacteria, through the analyses of genes encoding the secreted proteome vs. those encoding the cytoplasmatic proteome in large and small genomes.

-The study makes some broad claims on the evolution of bacteria while being based on a dataset of freshwater metagenome-assembled genomes (MAGs) from five lakes. Most of the used MAGs represent partial (incomplete) genomes (see extended data Fig1). Furthermore, it is known that MAGs in most cases do not recover the flexible genome (as genes that are part of the flexible genome may have low abundances, and not end up in assemblies). So, it is unclear to me that this dataset is enough to make broad claims on bacterial adaptive diversification. I'm also wondering why publicly available complete genomes (from cultures), including those from other environments, such as marine and soils, were not used if the aim is to investigate bacterial diversification in general (not only lakes). Genomes obtained from cultures in GTDB are a good resource, and I see some limited efforts in this direction were made in Extended Figure 6, but just very few marine genomes, mostly from *Pelagibacter*, were included. In sum, I think this work needs to show whether the used MAG collection, which includes a substantial fraction of incomplete genomes, is representative of full genomes, to support the main conclusions. Furthermore, broad claims need to be backed up with results from a broad set of environments, in my opinion (otherwise, results would apply mostly to lakes). In short, could adding more habitats (soil, marine), lakes (from other latitudes), and full genomes from cultures change the main conclusions?

-Some of the claims are bold, and it is difficult for me to see how they connect to the actual results. For example, on page 7 it says, "During these phases, the intrapopulation divergence in terms of genomic similarity primarily occurs as a result of non-adaptive genetic drift affecting CP-encoding genes." Conclusions like this need more supporting information, in my opinion. What is the evidence of drift? Did the authors measure the effective population size (N_e) to support this conclusion? (i.e., smaller N_e compared to other bacteria would at least suggest a higher prevalence of drift compared to selection).

-The manuscript is difficult to follow, and some conclusions seem to come out of the blue (e.g., several lines on page 7). One of the reasons may be that authors tried to summarise their work as much as possible. But, in order to show the reader the relevance and implications of this work, as well as the lines of reasoning, more information is needed. Specifically, background, concepts,

results, and conclusions should be better articulated. The scope should be better defined, since, as mentioned before, it is not solid to derive broad evolutionary processes from a dataset originating only from a few lakes. Alternatively, the scope of the work should be bacteria in temperate (?) freshwaters.

Other comments:

Fig1b: Even though the correlation seems clear, values of SP/CP range between ca. 0.96 and 1.1. Is this signal enough to draw conclusions?

Page 3 and several other locations, "species with large/small MAGs" sounds incorrect. Wouldn't be "species with large/small genomes"?

Page 4, "higher turnover rates of their SPs"... needs to be better explained.

Figure 2. Check "e)" in the legend, it is not in the figure.

Page 6, reads, "thus, the SPs evolution arises as the main speciation driver in both small and large MAG species", this sounds bold, and more supporting information is needed, in particular from other environments. Otherwise, tone down.

Extended Fig1. This figure could be a main figure, so the reader can clearly see from which dataset conclusions are drawn. In particular, it is relevant to see that High-Quality MAGs were ca. 28% of the total MAGs. As commented above, when this figure is shown, it is important to highlight that all the information comes from 5 lakes and indicate how representative this dataset is of broad processes such as bacterial speciation across habitats (if that is the aim).

Extended Figure 5: check colors (e.g., I can not see the "orange" that is mentioned)

Reviewer #2 (Remarks to the Author):

In their manuscript entitled “Genome-reduced bacteria exhibit pervasive episodes of adaptive stasis”, the authors reconstructed 5,519 freshwater prokaryotic genomes (dpCEL database) from 51 metagenomes sampled across 5 European lakes. They selected 342 genomes from 30 species to compare the evolution of cytosolic and secreted proteomes (CPs and SPs, respectively) and used the SPs as a proxy for adaptive signatures to environmental factors. I found the principle of comparing evolutionary signatures between the SP and CP of particular originality and interest, which led to the identification of intriguing patterns and results (reduced diversity, functional redundancy and positive selection in the SP of genome-reduced bacteria, which the authors interpret as evidence for adaptive stasis). However, as is, I find the main claims are not sufficiently supported and would require (1) thoroughly discussing the methodological limitations of the approach used, (2) properly addressing possible alternative explanations for the patterns observed (instead of adaptive stasis) and (3) realigning the claims with the data by limiting extrapolations.

Limitations of inferring the secreted proteome from computational annotations

To determine the secreted proteome, the authors scanned the sequences of predicted proteins using Phobius, which is used to infer the presence of a signal peptide. The authors then attribute all protein sequences with a signal peptide to the SP, while they attribute the other predicted proteins (i.e., with no detected signal peptide) to the CP. Although this approach has a lot of value, in particular its ease of use and scalability, it also comes with drawbacks that the author ought to better discuss, considering how important the SP/CP dichotomy is for the conclusions of this work. In particular, one could ask: how does the rate of false positives/negatives of Phobius impact the attribution to SP/CP? Could the lower fraction of proteins belonging to the secreted proteome be an artifact due to lower annotation rates? How does the annotations used here compare to the subcellular localisation reported in the SwissProt database (whenever a match is available)? In addition, if any of the organisms in the pdCEL database were to have had its extracellular proteome characterized experimentally, it could serve as a perfect benchmark to validate the methods used here by the authors. Although I understand that some of these points may be beyond the scope of this manuscript, considering how fundamental the predicted localization is for this work, I would suggest including additional support for this methodological strategy as well as clear discussions of its limitations.

Ruling out potential sampling biases

The authors processed a set of >5,500 MAGs to select 30 species with enough representative genomes for their downstream analyses ($n \geq 9$ MAGs), for a final dataset of 342 MAGs. Although by selecting species with at least 9 genomes, the authors are trying to ensure that they sampled enough of the within-species diversity to ensure the validity of their downstream analysis, the structure of the dataset may require further sanity checks. For instance, the ~50 samples were taken from 5 different lakes. If the ≥ 9 genomes for a single species are coming from the same lake, the sampling of the diversity of that species would be biased compared to a species that has genomes

reconstructed across the 5 lakes. As such, the authors should ensure that the reduced diversity observed in species with smaller genomes does not correlate with a smaller number of lakes being sampled for said species. This could be as simple as adding a small bar plot within Figure 1a displaying this number for instance.

Average amino-acid identity as a diversity metric

To evaluate the diversity of the CPs and SPs across the studied species, the authors use the average amino-acid identity (AAI) within the SPs/CPs of proteins. This approach was designed to compare the relatedness between two prokaryotes based on whole genomes (<https://journals.asm.org/doi/10.1128/jb.187.18.6258-6264.2005>), however, it may not be the best metric to explore the diversity within a set of proteins. If my understanding is correct, the authors have used a reciprocal best-hit strategy (with a 70% coverage cutoff), as reported in the paper aforementioned, to evaluate the AAI distributions. However, such an approach raises several questions: what happens if a protein sequence does not align to any other in the set (i.e., has no homolog above the detection limit)? I assume the comparison would then be ignored, however, shouldn't it increase the diversity within the studied set of proteins? If so, to ensure that the approach used by the authors is appropriate, the proportion of proteins for which no reciprocal best hit was found across the studied species should be reported. Alternatively, grouping SPs/CPs into orthologous clusters (which the authors have done already for a downstream analysis) and reporting the number of clusters along with the intra-cluster AAI could provide a more robust picture of the diversity within SPs and CPs across species.

Discussion of alternative explanations to the adaptive stasis

The authors observe a reduced SP in bacterial species with small genomes, which associates with a comparatively lower within-species diversity in the proteins of the SP as opposed to the CP (based on average amino-acid identity). The authors then study the selective pressures on that diversity and interpret their results as a lack of adaptation in the SP of genome-reduced bacteria. However, there could be alternative scenarios that would need to be ruled out or at least discussed to support such a conclusion.

First, adaptation may not rely solely on mutations. For instance, prokaryotes can experience high rates of horizontal transfers leading to gene gains and losses that would enable rapid adaptation of the SP without relying on mutations as the generator of diversity, which would not be detected in the current analyses. Yet, this adaptive scenario is particularly relevant in this analysis, since variations in the pangenome of genome-reduced bacteria (e.g., as the marine cyanobacteria *Prochloroccus*, see here <https://www.nature.com/articles/nrmicro3378> for instance) is known to be particularly important for niche-adaptation.

Second, another potential confounding factor is that species with large and small genomes are adapted to different ecological niches. Indeed, larger genomes often associate with a particle-attached lifestyle or nutrient-rich waters (copiotrophic) while reduced-genomes often associate with a free-living, oligotrophic lifestyle (see <https://www.nature.com/articles/ismej201460> or

<https://www.nature.com/articles/s41564-017-0008-3> for instance). Although this would not invalidate the association between the smaller genomes and less diverse SP, the difference in ecological niches may arguably be more important to explain the SP/CP differences than the genome size. Although it may not be able to untangle these effects (large vs small genome or copiotrophic vs oligotrophic), I believe the overlap between the two sets of traits should at least be discussed when interpreting the results.

Finally, could the differences in SP size and diversity be linked to the presence/absence of a motility, attachment and/or chemotaxis apparatus? These apparatus are expected to take up a substantial portion of the secreted proteome and to be under important evolutionary constraints (as potential phage or predator receptor or antigen). This could be tested using bioinformatic prediction tools such as TXSScan for instance (<https://github.com/macsy-models/TXSScan>). Although the presence of these apparatus in larger genomes and their absence in smaller ones would not change the validity of the patterns observed, it would enrich the interpretation of the results and

Realign conclusions with supporting data/results or underlying hypotheses

In a number of places, the authors use strong language to phrase their conclusions, which may not align with the data or results they have to support them. This include, for instance:

- Page 2: "Here, we exploit the power of genome-resolved metagenomic analyses to capture the diversity of dominant aquatic bacterial populations in order to assess the adaptive state of bacterial species." However, assessing the adaptive state of a bacterial species would arguably go beyond what the authors are conducting in this study.

- Page 3: "Given that this level of proteome divergence (Fig. 1c) appears to be comparable across various species exhibiting diverse evolutionary histories, lifestyle strategies, and genome sizes, it is highly compelling to attribute it to the fundamental influence of baseline genetic drift." This statement would suggest that much of the within-species diversity is non-adaptive, which is very challenging to prove and goes against a lot of our understanding of microbial within-species diversity. Please see <https://academic.oup.com/mbe/article/35/6/1338/4976545> or <https://www.nature.com/articles/s41579-020-0368-1> for instance.

- Page 6: "This indicates that adaptive stasis as observed here is not an "evolutionary lobster trap" and that small MAGs species only transiently decouple evolution from environmental adaptation." The observation that the difference in SP and CP diversity observed within species does not hold true at the within-genera level is very interesting, and could actually motivate much of the work done by the authors (by reporting it at the beginning thus motivating the follow up analyses). However, as phrased, their interpretation of this difference of diversity as a "transient[...] decoupl[ing] between evolution and environmental adaptation" relies on several assumptions that should be at least discussed before reaching such a strong conclusion.

- Page 6: "Hence, the apparent evolutionary stasis can be attributed to their secreted proteomes reaching maximum efficiency." Again, a very strong statement, while alternative explanations to the lack of diversity are not discussed and there is no functional data to support the proteome efficiency beyond little positive selection detected in the protein sequences. As mentioned before, there are

alternative sources of adaptation, and non-synonymous mutations may actually be adaptive (as also mentioned by the authors).

- Page 7: “The inherent inflexibility within their SPs thus restricts the capacity of these organisms to explore novel genetic variations and effectively adapt to dynamic environmental conditions.” Same as above, this strong statement relies on the hypothesis that any adaptation to the environment is mediated by (non-synonymous) mutation of existing genes rather than acquisition of new genomic fragments for instance without discussing it.

In addition, multiple of these conclusions (including in the title) generalize the results based on 30 freshwater microbial species samples across 51 metagenomes to all genome-reduced bacteria. The authors briefly mention that “further investigation is required to determine its prevalence among genome-reduced bacteria in diverse microbiomes”, yet many of their statements do not reflect that. In practice, the authors should either realign their conclusions with the data they analyze, or support the scale of their statement with new data (e.g. analysis of the SP/CP across species from GTDB database for instance).

Minor/Miscellaneous points:

- The authors describe “51 shotgun sequencing datasets”. Clarify whether this refers to 51 independent sequencing datasets (each containing multiple samples) or 51 samples, i.e. 51 metagenomes.

- There are multiple places where the authors use the phrasing “small species” to refer to species with reduced genomes, however, it is the genome that is small not the species itself.

- Page 4, the authors write that “a reduction in the number of genes under negative selection indicates increased selection pressure”. However, wouldn’t that be the opposite? The more negative selection the more selective pressure?

- Page 4, the authors mention that synonymous mutations may be non-neutral. See this <https://academic.oup.com/mbe/article/35/6/1338/4976545> for additional discussion on the lack of neutral sites in bacterial genomes.

- In figure 2a, the authors report the % of genes under positive, negative selection and those that are invariable. Wouldn’t the invariable genes be under negative selection as well? Or at minima, there is no variation and thus no way to test for the selective regime? As such, interpreting an increase in invariant genes (and thus a decrease in negatively selected genes) as less negative selection seems misleading. Possibly, the ratio of positively selected genes vs negatively selected genes would be a better approach to interpret this compositional dataset. In which case the ratio would increase from CP to SP for large genomes (indicating positive selection in the SP) while it would largely decrease for small genomes (indicating strong negative selection).

- In figure 2b, the number of sites under negative selection per protein will be highly dependent on the protein length. Although the length distribution is reported in Figure 2c, it would be better practice to rapport a normalized value in figure 2b.

- The functional redundancy analysis is based on KO annotations. However, the percentage of genes with KO annotations may be low and variable across genomes/species. This may weaken the estimations computed here. A per species protein clustering approach (either using the orthologous clusters directly, or de novo less stringent clustering using e.g. mmseqs2) would allow to compute similar values by dividing the number of clusters by the number of proteins independently of annotation rates.
- Although I understand the attractiveness of dividing the dataset between species with large and small genomes, could it be more powerful to use the genome size as a continuous explanatory variable? One could then correlate the difference between SP and CP diversity or the ratio of negatively selected and positively selected sites by the genome size across the 30 species.
- The observation that the SPs diversity is high between genera but not within species for genome-reduced bacteria is very intriguing and interesting. Could this be linked species with smaller genomes having higher effective population sizes? Genome-reduced bacteria often have large population size (Pelagibacter, Prochlorococcus, acl, ...), and therefore a more efficient selection with less room for drift, potentially reducing the emergence of non-adaptive diversity while rapidly allowing the fixation the adaptive innovations. On the contrary copiotrophs with large genomes may undergo more bloom-like events that would favor the emergence of non-selective processes.
- In “Assembly, binning, and MAG designation” the authors state that the “metagenomic datasets were mapped [...] against the assembled contigs [...] in a sample-dependent fashion.”. Please clarify whether the reads from all metagenomic samples were mapped against the contigs from all samples, or if only the reads from a given sample were mapped to the matching assembly. In particular, the approach leveraging the abundance of contigs across multiple samples has proven important to improve the quality of the MAGs reconstructed (see <https://www.nature.com/articles/s41592-023-01934-8> for a most recent evaluation).
- Please add a data availability statement referencing the bioproject ids at which the metagenomes and MAGs can be accessed.

Reviewer #3 (Remarks to the Author):

The authors have produced a large set of metagenome-assembled genomes (MAGs) of bacteria and archaea from freshwater bodies in Central Europe and used it to investigate relative rates of adaptation among microbial taxa. Their analyses indicated elevated selective pressure on secreted proteins (SPs) in small MAGs, as compared to large MAGs. This was interpreted as a proof of an adaptive stasis in microbial taxa with small genomes, caused by the reduced functional redundancy in their genomes (no “backup plan”). This study addresses important questions in microbial evolution and presents a compelling hypothesis that deserves robust testing. However, I believe that the type of core data used in this study - >5,500 MAGs – is prone to major methodological biases and may be fundamentally unsuitable for the type of analyses performed here. While MAGs are very

instrumental in the recovery genomic information from yet uncultured taxa, they are highly prone to both the inclusion of genome regions from unrelated organisms (chimerism) and exclusion of hypervariable genome regions (pangenome under-sampling). Importantly, this is not reflected by genome completeness estimates by CheckM and similar tools, which simply count the presence of a small subset of core genes. For a few studies exploring these methodological challenges in great detail, see the following:

1 Sczyrba, A. et al. Critical Assessment of Metagenome Interpretation - A benchmark of metagenomics software. *Nature Methods* 14, 1063-1071 (2017).
<https://doi.org/10.1038/nmeth.4458>

2 Vollmers, J., Wiegand, S., Lenk, F. & Kaster, A. K. How clear is our current view on microbial dark matter? (Re-)assessing public MAG & SAG datasets with MDMcleaner. *Nucleic Acids Res* (2022).
<https://doi.org/10.1093/nar/gkac294>

3 Orakov, A. et al. GUNC: detection of chimerism and contamination in prokaryotic genomes. *Genome Biol* 22, 178 (2021). <https://doi.org/10.1186/s13059-021-02393-0>

Pangenome under-sampling is particularly relevant here, as it may introduce major biases leading to flawed conclusions. As an illustration, despite two decades of substantial efforts, marine microbiologists still struggle to produce MAGs of *Pelagibacter* and *Prochlorococcus*, the two most abundant genera in the ocean, both of which have small genomes. Meanwhile, it is well known that pangenomes of *Pelagibacter* and *Prochlorococcus* are enormous, with ample evidence for frequent horizontal gene transfer and recombination. Thus, the situation with *Pelagibacter* and *Prochlorococcus* is the complete opposite to the adaptive stasis proposed in the manuscript: they evolve rapidly and recombine their genomes frequently, challenging scientists' ability to represent their populations in MAGs.

In order to make their data more interpretable, authors of the manuscript should evaluate how accurately their MAGs represent the genomic composition of analyzed microbial communities. The following is essential:

1. How accurate are the metagenome assembly and binning methods used in the study? This can be addressed by validating the entire workflow with mock metagenomes, such as those used in the Sczyrba et al. publication cited above and publicly available.

2. What fraction of genes encoding secreted and cytoplasmic proteins by the entire community is represented by the obtained MAGs? This can be done by the recruitment of individual metagenome reads on MAGs.

3. How do metagenome assembly algorithms, which produce consensus sequences from a multitude of organisms, impact the ability to measure evolutionary pressure metrics, such as Dn/Ds ratio? This can be done by software designed to explore intra-population variation in MAGs, such as inStrain:

Olm, M. R. et al. inStrain profiles population microdiversity from metagenomic data and sensitively detects shared microbial strains. *Nat Biotechnol* 39, 727–736 (2021).

Dear Reviewers,

We appreciate the thoughtful comments provided by the referees on our manuscript with the identifier **NCOMMS-23-32470-T**. On behalf of my co-authors, I am pleased to address the raised issues and provide a detailed response to each comment. We believe that the insightful feedback and suggestions have significantly enhanced the overall quality of our work. The reviewer's statements are quoted in italics, followed by our responses in regular blue text.

REVIEWER COMMENTS

Reviewer #1 (Remarks to the Author):

This work investigates the adaptive capacity of aquatic (freshwater mostly) bacteria, through the analyses of genes encoding the secreted proteome vs. those encoding the cytoplasmatic proteome in large and small genomes.

-The study makes some broad claims on the evolution of bacteria while being based on a dataset of freshwater metagenome-assembled genomes (MAGs) from five lakes.

Response 1.1:

We appreciate the reviewer's insight, and upon careful consideration of the feedback, we have decided to refine the focus of our study. The revised title, **‘Freshwater genome-reduced bacteria exhibit pervasive episodes of adaptive stasis’**, is a more accurate representation of our work. This choice aligns with our decision to base our findings on a curated dataset of 5,500 metagenome-assembled genomes that we meticulously generated. Furthermore, we have revised the purpose of our study to emphasize our aim **‘to assess eco-evolutionary forces shaping the diversity of freshwater bacterial species’**. These adjustments better encapsulate the essence of our research and directly respond to the reviewer's comments. We believe these changes contribute to the overall coherence and precision of our manuscript.

-Most of the used MAGs represent partial (incomplete) genomes (see extended data Fig1). Furthermore, it is known that MAGs in most cases do not recover the flexible genome (as genes that are part of the flexible genome may have low abundances, and not end up in assemblies). So, it is unclear to me that this dataset is enough to make broad claims on bacterial adaptive diversification.

Response 1.2:

We appreciate the reviewer's meticulous examination of our manuscript and the invaluable insights provided. In response to the concern raised about the dataset's completeness to study genome evolution, we have incorporated the median completeness of our MAGs species clusters into the manuscript (median completeness=77.03%). Notably, these completeness levels align with results of previous studies, obtained through cutting-edge sequencing and bioinformatic approaches, particularly on environmental samples¹.

We acknowledge the reviewer's note on the potential incompleteness of MAGs in recovering parts of the flexible genome. However, we want to emphasize our commitment in conducting repeated sampling campaigns for the selected environments as a proactive measure to reduce this limitation from the beginning. By sampling the same population multiple times, we significantly enhanced the capacity to capture its flexible genome. Crucially, we took a meticulous approach in clustering our pdCEL database into cluster-level species, specifically opting for clusters with a minimum of 9 representatives. This careful selection is geared toward maximizing our ability to recover the flexible genome. Additionally, we have incorporated **pangenome analyses** for all recovered species clusters, revealing to which extent, the flexible genome was successfully recovered (Supplementary Information - Fig. S9). Furthermore, we chose a specific sample for an in-depth microdiversity analysis and noted the prevalent dominance of one population at the time of sampling, yielding MAGs containing segments of the flexible genome (Supplementary Information – Fig. S6).

To enhance the robustness of our analyses, we conducted a benchmarking exercise on our within-species diversity analysis and the quantification of diversity-generating mechanisms. This validation utilized genomic data sourced from species represented by metagenome-assembled genomes (MAGs), single-cell amplified genomes (SAGs), and isolates. The outcomes of these analyses convincingly showed that the inclusion of MAGs does not introduce any bias into the conducted analyses (Supplementary Information - Figs. S3-4).

Recognizing the challenge of sampling all intraspecies diversity due to continuous population evolution, our comparison of multiple populations (i.e., MAGs) from the same environment allows us to disentangle eco-evolutionary strategies within freshwater environments. We are confident that the refined analysis and the reoriented focus of the paper significantly elevate the overall quality of our work. To address methodological limitations hindering the sampling of the complete species diversity, we have introduced a new section in the Methods part titled `Limitations`.

References:

1. Paoli, L. *et al.* Biosynthetic potential of the global ocean microbiome. *Nature* 2022 607:7917 607, 111–118 (2022).
2. Hofmann, F. M. P. *et al.* AMBER: Assessment of Metagenome BinnERs. *Gigascience* 7, 1–8 (2018).

3. Meyer, F. *et al.* Critical Assessment of Metagenome Interpretation: the second round of challenges. *Nature Methods* 2022 19:4 19, 429–440 (2022).

I'm also wondering why publicly available complete genomes (from cultures), including those from other environments, such as marine and soils, were not used if the aim is to investigate bacterial diversification in general (not only lakes). Genomes obtained from cultures in GTDB are a good resource, and I see some limited efforts in this direction were made in Extended Figure 6, but just very few marine genomes, mostly from Pelagibacter, were included.

Response 1.3:

We appreciate the reviewer's insightful suggestion, and in response, we broadened our search to include a more extensive collection of genomes, MAGs, and SAGs from various environmental samples. However, despite the relative accessibility of genomes from cultured bacteria, obtaining genomic data for genome-reduced representatives proved exceptionally challenging. Additionally, the scarcity of species with more than nine genomes in existing databases posed a significant limitation. In our effort to address this concern, we conducted a thorough screening of GTDB and several studies reporting extensive genomic data sets. The compiled data is presented in Supplementary Table S6. Despite these efforts, the limitations in the data availability persisted, primarily due to the common practice of reporting a single representative genome for a species cluster. This practice significantly hampers the feasibility of conducting species-centric studies. However, we managed to recover and analyse genomic information from 57 (592 genomes) environmental species. Following the reviewer's suggestion, we opted to focus our results and discussion on the data generated, curated, and analysed. **These findings, representative of different environmental samples, serve to corroborate our conclusions, while it is important to note that they are not intended to provide a definitive account of the bacterial diversity in these environments.** We believe this refined approach enhances the robustness and reliability of our study.

In sum, I think this work needs to show whether the used MAG collection, which includes a substantial fraction of incomplete genomes, is representative of full genomes, to support the main conclusions.

Response 1.4:

We appreciate the reviewer's insightful suggestion, and to address this concern, we conducted a benchmark analysis using an abundant environmental marine species, **Prochlorococcus_B marinus_B** (Supplementary Information - Fig. S3). This species was chosen due to the availability of cultured representatives, single-cell amplified genomes (SAGs), and metagenome-assembled genomes (MAGs). By comparing the SPs/CPs ratio among genomes from cultures, SAGs, and MAGs, it becomes evident that the results are indeed comparable, with cultures and MAGs yielding highly similar outcomes. These results

were further supported by the absence of statistical difference (ANOVA; $df = 2$; f -value = 0.3852; p -value = 0.6811) in the SPs/CPs ratio among the three groups (i.e., culture, SAGs and MAGs). The small deviation observed in the SAGs can likely be attributed to errors introduced during the single-cell amplification steps. This comparative analysis substantiates the reliability of our MAG collection to draw meaningful conclusions.

Furthermore, broad claims need to be backed up with results from a broad set of environments, in my opinion (otherwise, results would apply mostly to lakes). In short, could adding more habitats (soil, marine), lakes (from other latitudes), and full genomes from cultures change the main conclusions?

Response1.5:

Following the reviewer's thoughtful suggestions, we have refrained from making overly generalized statements and opted to focus our discussion on the dataset we meticulously analysed. In response to the reviewer's query about the potential impact of including more habitats and genomes from cultures, we conducted an extensive search in public genomic repositories. This effort allowed us to recover species representatives from various environmental samples, ensuring a substantial number of genomes for analysis (see Response 1.3). The obtained results corroborated our findings, with some species showing more conserved SPs, while others exhibiting more diverse SPs (in comparison with their CPs) (see Supplementary Information - Fig. S8). However, it is essential to acknowledge the limitations arising from the restricted availability of genomic data for the majority of taxonomic groups in the analysed habitats. The primary constraint stems from the limited number of genomes that are currently available for a specific species.

-Some of the claims are bold, and it is difficult for me to see how they connect to the actual results. For example, on page 7 it says, "During these phases, the intrapopulation divergence in terms of genomic similarity primarily occurs as a result of non-adaptive genetic drift affecting CP-encoding genes." Conclusions like this need more supporting information, in my opinion. What is the evidence of drift? Did the authors measure the effective population size (N_e) to support this conclusion? (i.e., smaller N_e compared to other bacteria would at least suggest a higher prevalence of drift compared to selection).

Response1.6:

We understand the need for more specific evidence to support our conclusions. We acknowledge that measuring the effective population size is very challenging in our context. To enhance the transparency of our claims, we will rephrase the statement to better reflect the available evidence and avoid making assertions that cannot be directly supported by the parameters measured. Thus the sentence is rephrased to the following: `During these phases, intrapopulation divergence in terms of genomic similarity is chiefly driven by

mutation/recombination events and heightened positive selection acting on CP-encoding genes.`

-The manuscript is difficult to follow, and some conclusions seem to come out of the blue (e.g., several lines on page 7). One of the reasons may be that authors tried to summarise their work as much as possible. But, in order to show the reader the relevance and implications of this work, as well as the lines of reasoning, more information is needed. Specifically, background, concepts, results, and conclusions should be better articulated. The scope should be better defined, since, as mentioned before, it is not solid to derive broad evolutionary processes from a dataset originating only from a few lakes. Alternatively, the scope of the work should be bacteria in temperate (?) freshwaters.

Response1.7:

We appreciate the insightful feedback from the reviewer concerning the clarity and articulation of our manuscript. In response, we have realigned the scope of our work, as previously mentioned to closely match our dataset. The manuscript was enriched with more detailed background information to enhance the overall readability. These adjustments are intended to establish a more seamless flow between background, concepts, results, and conclusions. We are confident that these modifications effectively address the concerns raised.

Other comments:

Fig1b: Even though the correlation seems clear, values of SP/CP range between ca. 0.96 and 1.1. Is this signal enough to draw conclusions?

Response1.8:

The observed range of SP/CP values, spanning from approximately 0.96 to 1.1, may seem subtle initially, but it bears significant ecological and evolutionary implications. In theory, SP and CP values within species should ideally be 1, given that both categories accumulate mutations and are influenced by the same effects of recombination. Deviations from this expected value of 1 indicate distinct selective pressures acting on these gene categories. For example, a value of 0.96 signifies a 4% difference at the amino acid level. The contrasting patterns of SP conservation between large and small genome species further underscore its ecological relevance. We posit that the subtleties in SP/CP values reveal the underlying complexities in the selective forces shaping the genetic diversity within the studied species.

Page 3 and several other locations, "species with large/small MAGs" sounds incorrect. Wouldn't be "species with large/small genomes"?

Response1.9:

We appreciate the reviewer's keen observation, and we have incorporated the suggested change. The term "species with large/small MAGs" has been replaced with "species with large/small genomes" throughout the manuscript to accurately reflect the intended meaning.

Page 4, "higher turnover rates of their SPs"... needs to be better explained.

Response1.10:

We followed the reviewer's suggestion and enhanced the clarity of the specified sentence. The original 'While the exact cause of length reduction in SPs compared to CPs remains elusive, it is plausible that small MAG species have higher turnover rates of their SPs.' has been revised to 'While identifying the exact cause of length reduction in SPs compared to CPs remains elusive, it's plausible that small genome species undergo more frequent SPs turnover, involving the removal and degradation of existing proteins, along with a dynamic renewal process, where new proteins are synthesized to replace those removed.'

Figure 2. Check "e)" in the legend, it is not in the figure.

Response 1.11:

Thank you for pointing out the discrepancy in Figure 2 (now Figure 4). We have checked and resolved the inconsistency in the legend.

Page 6, reads, "thus, the SPs evolution arises as the main speciation driver in both small and large MAG species", this sounds bold, and more supporting information is needed, in particular from other environments. Otherwise, tone down.

Response1.12:

We have revised our statement to align more accurately with the analysed data. 'Therefore, the evolution of SPs emerges as a pivotal factor driving freshwater bacteria differentiation in both the analysed small and large genome species.'

Extended Fig1. This figure could be a main figure, so the reader can clearly see from which dataset conclusions are drawn. In particular, it is relevant to see that High-Quality MAGs were ca. 28% of the total MAGs. As commented above, when this figure is shown, it is important to highlight that all the information comes from 5 lakes and indicate how representative this dataset is of broad processes such as bacterial speciation across habitats (if that is the aim).

Response1.13:

We appreciate the reviewer's input on the importance of Extended Figure 1. We have promoted it to main figure (Figure 1). Moreover, we want to highlight that we carefully

considered the reviewer's insights, leading us to refine the study's objectives for better alignment with the analysed data (please refer to our Response 1.1). These adjustments ensure a more accurate representation of our research goals, directly addressing the reviewer's concerns. We are confident that these modifications significantly contribute to the overall clarity and precision of our manuscript.

Extended Figure 5: check colors (e.g., I can not see the "orange" that is mentioned).

Response1.14:

We thank the reviewer for the comment. Extended Figure 5 (promoted to main as Figure 7) has been enhanced to ensure better visibility of the orange colour.

Reviewer #2 (Remarks to the Author):

In their manuscript entitled "Genome-reduced bacteria exhibit pervasive episodes of adaptive stasis", the authors reconstructed 5,519 freshwater prokaryotic genomes (CEL database) from 51 metagenomes sampled across 5 European lakes. They selected 342 genomes from 30 species to compare the evolution of cytosolic and secreted proteomes (CPs and SPs, respectively) and used the SPs as a proxy for adaptive signatures to environmental factors. I found the principle of comparing evolutionary signatures between the SP and CP of particular originality and interest, which led to the identification of intriguing patterns and results (reduced diversity, functional redundancy and positive selection in the SP of genome-reduced bacteria, which the authors interpret as evidence for adaptive stasis). However, as is, I find the main claims are not sufficiently supported and would require (1) thoroughly discussing the methodological limitations of the approach used, (2) properly addressing possible alternative explanations for the patterns observed (instead of adaptive stasis) and (3) realigning the claims with the data by limiting extrapolations.

Response2.1:

We appreciate the thoughtful insights provided by the reviewer and their meticulous examination of our manuscript. The detailed comments raised will be comprehensively addressed in the subsequent responses to ensure clarity and coherence. Your feedback is invaluable in refining our work, and we are committed to addressing each point raised to enhance the overall quality and precision of our manuscript.

Limitations of inferring the secreted proteome from computational annotations

To determine the secreted proteome, the authors scanned the sequences of predicted proteins using Phobius, which is used to infer the presence of a signal peptide. The authors then attribute all protein sequences with a signal peptide to the SP, while they attribute the

other predicted proteins (i.e., with no detected signal peptide) to the CP. Although this approach has a lot of value, in particular its ease of use and scalability, it also comes with drawbacks that the author ought to better discuss, considering how important the SP/CP dichotomy is for the conclusions of this work. In particular, one could ask: how does the rate of false positives/negatives of Phobius impact the attribution to SP/CP? Could the lower fraction of proteins belonging to the secreted proteome be an artifact due to lower annotation rates? How does the annotations used here compare to the subcellular localisation reported in the SwissProt database (whenever a match is available)? In addition, if any of the organisms in the pdCEL database were to have had its extracellular proteome characterized experimentally, it could serve as a perfect benchmark to validate the methods used here by the authors. Although I understand that some of these points may be beyond the scope of this manuscript, considering how fundamental the predicted localization is for this work, I would suggest including additional support for this methodological strategy as well as clear discussions of its limitations.

Response 2.2:

Phobius, a bioinformatics tool applied in this study, employs a combination of algorithms for predicting signal peptides in bacterial proteins by recognizing and analysing the N-terminus regions. This methodology is commonly employed not only to predict protein localizations in well-known bacterial species^{1,2} but also in uncultured microbial dark matter³. It is worth mentioning that Phobius is frequently integrated into prominent protein annotation databases and toolkits^{4,5}, underscoring its established reliability in predicting protein subcellular localization.

We appreciate the guidance provided by the reviewer, and we have diligently incorporated the suggestions. To benchmark the identification of the secreted and cytoplasmic proteomes, we extracted 120,300 cytoplasmic proteins, 10,189 signal peptide-bearing proteins, and 3,156 periplasmic proteins from the UniProtKB/Swiss-Prot reviewed protein database. These proteins underwent rigorous scrutiny to ensure their attribution to bacteria within this high-quality, manually curated database. Phobius software, employing default parameters, was utilized to predict signal peptides of the proteins within the three aforementioned categories. The accuracy of Phobius was calculated by quantifying the number of sequences with correct signal peptide predictions using the following equation: **Phobius accuracy = (N of correct predictions) / (N of total predictions) x 100**. The software demonstrated its high accuracy while correctly identifying 97.66% of the cytoplasmic proteins (CPs), 93.61% of the signal peptide-bearing proteins (SPs), and 91.25% of the periplasmic proteins (SPs). These findings align with the accuracies reported in previous studies⁶, reinforcing our confidence in the performance of the Phobius software.

The results of Phobius are presented in the method description of the manuscript, while the potential limitations and pitfalls of the computational annotations are introduced in the **Limitations'** section.

Regrettably, none of the species from the pdCEL database were successfully isolated in culture, a common challenge due to the reluctance of many abundant environmental microbes towards state-of-the-art cultivation methods. Utilizing the Swiss-Prot database, we observe that, looking at the E. coli K12 strain, about 11.82% of its proteins (out of 3,019 proteins with cellular localization information) are experimentally identified as periplasmic- or outer membrane-associated. This aligns seamlessly with our findings, showing that a substantial fraction of the proteome is comprised by cytoplasmic proteins (CPs).

References:

1. Geisinger, E. *et al.* Antibiotic susceptibility signatures identify potential antimicrobial targets in the *Acinetobacter baumannii* cell envelope. *Nature Communications* 2020 11:1 11, 1–16 (2020).
2. Cong, Q., Anishchenko, I., Ovchinnikov, S. & Baker, D. Protein interaction networks revealed by proteome coevolution. *Science* (1979) 365, 185–189 (2019).
3. Neri, U. *et al.* Expansion of the global RNA virome reveals diverse clades of bacteriophages. *Cell* 185, 4023–4037.e18 (2022).
4. Jones, P. *et al.* InterProScan 5: genome-scale protein function classification. *Bioinformatics* 30, 1236–1240 (2014).
5. Mitchell, A. L. *et al.* InterPro in 2019: Improving coverage, classification and access to protein sequence annotations. *Nucleic Acids Res* 47, D351–D360 (2019).
6. Reynolds SM, Käll L, Riffle ME, Bilmes JA, Noble WS (2008) Transmembrane Topology and Signal Peptide Prediction Using Dynamic Bayesian Networks. *PLOS Computational Biology* 4(11): e1000213.

Ruling out potential sampling biases

The authors processed a set of >5,500 MAGs to select 30 species with enough representative genomes for their downstream analyses (n >= 9 MAGs), for a final dataset of 342 MAGs. Although by selecting species with at least 9 genomes, the authors are trying to ensure that they sampled enough of the within-species diversity to ensure the validity of their downstream analysis, the structure of the dataset may require further sanity checks. For instance, the ~50 samples were taken from 5 different lakes. If the >=9 genomes for a single species are coming from the same lake, the sampling of the diversity of that species would be biased compared to a species that has genomes reconstructed across the 5 lakes. As such, the authors should ensure that the reduced diversity observed in species with smaller

genomes does not correlate with a smaller number of lakes being sampled for said species. This could be as simple as adding a small bar plot within Figure 1a displaying this number for instance.

Response2.3

We thank the reviewer for the comment. To address this comment, we added the number of sampled lakes for each species cluster to Figure 3a. To test the relationship between the number of sampled lakes and the observed reduced diversity in small genome species, we constructed various statistical models. These models included multiple predictor variables and interactions. Specifically, we considered models incorporating SP/CP ratio, coding density, and lake diversity as well as their interactions. Our analysis, supported by the Akaike Information Criterion (AIC) to balance goodness of fit and model complexity, revealed that lake diversity had little to no effect on the SP/CP ratio, while coding density (surrogate of genome size) consistently indicated a significant relationship (supplementary Table S12). We appreciate the reviewer's attention to this aspect and trust that this clarification provides a more comprehensive view on our data.

Average amino-acid identity as a diversity metric

To evaluate the diversity of the CPs and SPs across the studied species, the authors use the average amino-acid identity (AAI) within the SPs/CPs of proteins. This approach was designed to compare the relatedness between two prokaryotes based on whole genomes (<https://journals.asm.org/doi/10.1128/jb.187.18.6258-6264.2005>), however, it may not be the best metric to explore the diversity within a set of proteins. If my understanding is correct, the authors have used a reciprocal best-hit strategy (with a 70% coverage cutoff), as reported in the paper aforementioned, to evaluate the AAI distributions. However, such an approach raises several questions: what happens if a protein sequence does not align to any other in the set (i.e., has no homolog above the detection limit)? I assume the comparison would then be ignored, however, shouldn't it increase the diversity within the studied set of proteins? If so, to ensure that the approach used by the authors is appropriate, the proportion of proteins for which no reciprocal best hit was found across the studied species should be reported. Alternatively, grouping SPs/CPs into orthologous clusters (which the authors have done already for a downstream analysis) and reporting the number of clusters along with the intra-cluster AAI could provide a more robust picture of the diversity within SPs and CPs across species.

Response2.4:

While we agree with the reviewer that the employed approach is suitable for evaluating the relatedness between two prokaryotes based on whole proteomes, it primarily involves conducting a thorough comparison of all proteins against each other, capturing pairwise

similarities. Our specific implementation utilized the blastP algorithm for bidirectional protein sequence comparisons.

In this context, we argue that an orthologous cluster-based approach demonstrates increased sensitivity to evolutionary phenomena such as protein duplications. As an illustration, let's consider a simplified scenario encompassing two genomes: the first containing peptide X with the sequence AAL, and the second one the peptide X as well as a duplication of X with a modification caused by a nonsynonymous mutation (AAL and AAA). When clustering these peptides and calculating their identity, the reported average is ~78% (Identity = Number of Identical Positions / Total Number of Positions = 14/18*100%). In this case, the identity between the peptides in the second genome (67%) contributes to decreasing the reported average. However, through all-versus-all comparisons of peptide X between the two genomes, we derive an average identity value of ~83% (Identity = Number of Identical Positions / Total Number of Positions = 10/12*100%). Notably, the peptides from the second genome are not compared against each other since they originate from the same genome. We contend that the latter method better captures the biological reality by performing proteome versus proteome comparisons in an all-versus-all fashion.

Despite recognizing the efficacy of orthologous cluster-based approaches (which we also employed in the manuscript), we prefer reporting the intraspecies amino acid identity as described. We believe that, in this case, it results in a more robust assessment of interpopulation proteome diversity.

The percentage of proteins utilized for all-versus-all proteome comparisons is added in detail for each species in supplementary Table S3. It is noteworthy that the median percentage of proteins compared is 73.13% for SP and 75.46% for CP, aligning with the species pangenome concept, accounting for the anticipated presence of some genome-specific proteins. Additional details about the method are provided in the dedicated section of the manuscript.

Discussion of alternative explanations to the adaptive stasis

The authors observe a reduced SP in bacterial species with small genomes, which associates with a comparatively lower within-species diversity in the proteins of the SP as opposed to the CP (based on average amino-acid identity). The authors then study the selective pressures on that diversity and interpret their results as a lack of adaptation in the SP of genome-reduced bacteria. However, there could be alternative scenarios that would need to be ruled out or at least discussed to support such a conclusion.

First, adaptation may not rely solely on mutations. For instance, prokaryotes can experience high rates of horizontal transfers leading to gene gains and losses that would enable rapid adaptation of the SP without relying on mutations as the generator of diversity, which would not be detected in the current analyses. Yet, this adaptive scenario is particularly relevant in this analysis, since variations in the pangenome of genome-reduced bacteria (e.g., as the

marine cyanobacteria *Prochlorococcus*, see here <https://www.nature.com/articles/nrmicro3378> for instance) is known to be particularly important for niche-adaptation.

Response 2.5:

We concur with the reviewer's insight that adaptation extends beyond mutation alone, recognizing the pivotal role of recombination in bacterial evolution. We initially omitted such analyses in light of prior studies indicating low recombination frequency in freshwater prokaryotes^{1,2}. However, upon reassessment, we opted to employ ClonalFrameML to assess the impact of recombination within the obtained species clades. Initially, we selected three *Prochlorococcus* species clusters and one *Pelagibacter* for methodological benchmarking (Supplementary Information - Fig. S4). Our findings revealed striking similarity in the data obtained from metagenome-assembled genomes (MAGs) compared to single-cell amplified genomes (SAGs) and isolates (available for one analysed *Prochlorococcus* species). These results align with a recent study implementing ClonalFrameML to assess recombination effects in MAGs recovered from ammonia-oxidizing archaea². Comprehensive details on the methodology, including benchmarking, are presented in the Methods section.

Recognizing the pivotal significance of these findings, we have opted to incorporate them into the manuscript. Consequently, Figure 4 has been introduced to showcase the outcomes of ClonalFrameML. The results intriguingly indicate that, although recombination occurs less frequently than mutation, its impact is more important than previously thought. Notably, we observed a remarkable contrast in the v values, representing the average phylogenetic distance of imports (i.e., the DNA fragments introduced through recombination) for the CPs of large-genome bacteria (See Fig. 4). This observation not only suggests that large-genome bacteria engage in recombination with more phylogenetically distant groups but also highlights that, despite its infrequency, recombination introduces sequence novelty. Remarkably, this novel sequence diversity is preferentially preserved within the SPs of large-genome bacteria. This newfound result aligns seamlessly with our analysis, showcasing increased diversity in the SPs of large bacterial species, and supports the insights derived from analysing evolutionary processes. **We posit that the investigation of evolutionary forces offers insights into the type of selection that acts on genes, irrespective of whether the diversity arises from mutation or recombination.**

While we acknowledge the excellent work on *Prochlorococcus*, it is vital to clarify that the study centres around the *Prochlorococcus* pangenome (genus-level) rather than a specific *Prochlorococcus* species. Despite the extensive nature of the *Prochlorococcus* pangenome and the involvement of flexible genes in niche adaptation, it is noteworthy that the genus comprises 332 species as of November 2023 (according to the GTDB database). The paper in question suggests that isolates occupying similar habitats often possess similar sets of flexible genes³. While our sampling efforts spanned the same habitat types, freshwater

lakes, and encompassed multiple time points, we acknowledge the inherent challenge of capturing the entirety of genomic diversity within populations. Even our aspiration to obtain more than nine MAGs for a species cluster is susceptible to potential loss of flexible genes. Therefore, it is crucial to recognize the inherent limitations in our approach. However, we contend that genes specific to a population, conferring a selective advantage, are more likely to be evolutionarily preserved. Our strategy of sampling the same habitat type multiple times enhances the likelihood of capturing the diversity of these populations. To transparently address these inherent constraints, we have included a dedicated section called Limitations in our manuscript.

References

1. Zaremba-Niedzwiedzka, K. *et al.* Single-cell genomics reveal low recombination frequencies in freshwater bacteria of the SAR11 clade. *Genome Biol* **14**, 1–14 (2013).
2. Ngugi, D. K. *et al.* Postglacial adaptations enabled colonization and quasi-clonal dispersal of ammonia-oxidizing archaea in modern European large lakes. *Sci Adv* **9**, eadc9392 (2023).
3. Biller, S. J., Berube, P. M., Lindell, D. & Chisholm, S. W. Prochlorococcus: the structure and function of collective diversity. *Nature Reviews Microbiology* **2014 13:1** **13**, 13–27 (2014).

Second, another potential confounding factor is that species with large and small genomes are adapted to different ecological niches. Indeed, larger genomes often associate with a particle-attached lifestyle or nutrient-rich waters (copiotrophic) while reduced-genomes often associate with a free-living, oligotrophic lifestyle (see <https://www.nature.com/articles/ismej201460> or <https://www.nature.com/articles/s41564-017-0008-3> for instance). Although this would not invalidate the association between the smaller genomes and less diverse SP, the difference in ecological niches may arguably be more important to explain the SP/CP differences than the genome size. Although it may not be able to untangle these effects (large vs small genome or copiotrophic vs oligotrophic), I believe the overlap between the two sets of traits should at least be discussed when interpreting the results.

Response 2.6:

We appreciate the reviewer's observation, and in response, we have expanded our ecological interpretation of the obtained results.

As our designated species clusters belong to uncultivated bacterial groups, we aimed to uncover aspects of their physiological potential by leveraging biological processes like replication and genomic characteristics, particularly codon usage bias^{1,2}. The methodology

involved the calculation of *in situ* replication rates using a "peak-to-trough ratio" approach and the inference of maximal growth rates by analysing the codon usage bias. Further technical details about these approaches are outlined in the Methods section. The results indicate that all large genome species exhibited active replication at the time of sampling, displaying high Growth rate index (GRiD) values (>2), suggesting fast-growth through multi-fork replication (Figure 8a). It is important to note that in a population where the majority of bacteria are replicating, the GRiD value would be equal to 2. In contrast, lower growth rates were predicted for small genome species, with few exceptions like Nanopelagicaceae sp.1, which displayed a median GRiD value of 1.97, challenging the observed pattern in the remaining small genome species. On the other hand, the codon usage bias of highly expressed gene (CUBHE) analysis (Figure 8b) indicated that most species theoretically have high maximal growth rates (CUBHE > 0.6), although these rates were likely not realized *in situ*, as showed by GRiD.

Upon juxtaposing genome sizes, *in situ* replication rates, and taxa with established ecophysiological traits^{3,4} (where feasible), it becomes apparent that larger genome species are likely affiliated with a copiotrophic lifestyle, while their small genome relatives exhibit characteristics indicative of an oligotrophic one. Rather than perceiving these two ecological strategies as a rigid dichotomy, it is pivotal to recognize them as a continuum that allows a spectrum of adaptations and behaviours. Broadly speaking, copiotrophic bacteria are marked by mobility, metabolic flexibility, and rapid growth responses—traits sharply contrasting those of oligotrophic species. The latter are recognized for their passive, free-living lifestyle and highly efficient organic compound uptake, especially at low nM concentrations. Copiotrophic entities, whether free-living or associated with lake snow particles, boast substantial cell and genome sizes, providing them with metabolic and regulatory flexibility⁴. Their heightened adaptability enables nuanced responses to swift nutrient changes, orchestrated through motility and chemotaxis, resulting in peak growth rates during favourable conditions. Conversely, genome-reduced oligotrophic bacteria, featuring smaller cell and genome sizes, have evolved to thrive in stable, nutrient-scarce, oligotrophic waters. Their growth strategy prioritizes efficiency over speed, maintaining a slow yet consistent pace⁴. Interpreting the variations in cytoplasmic and secreted proteomes within species through an eco-evolutionary lens reveals that the increase in positively selected genes and heightened sequence diversity in the SPs of large genome species likely propels a copiotrophic lifestyle. This enhancement expands the potential range of utilizable substrates. Conversely, the limited SP diversity in small genome species indicates enhanced substrate fidelity, as diversifying the substrate range may not confer an advantage, given the oligotrophic bacteria's reliance on a reduced set of metabolic pathways. These results and discussion are presented at page 5 of the manuscript.

References:

1. Emiola, A. & Oh, J. High throughput in situ metagenomic measurement of bacterial replication at ultra-low sequencing coverage. *Nature Communications* 2018 9:1 9, 1–8 (2018).
2. Weissman, J. L., Hou, S. & Fuhrman, J. A. Estimating maximal microbial growth rates from cultures, metagenomes, and single cells via codon usage patterns. *Proc Natl Acad Sci U S A* 118, e2016810118 (2021).
3. Giovannoni, S. J., Cameron Thrash, J. & Temperton, B. Implications of streamlining theory for microbial ecology. *The ISME Journal* 2014 8:8 8, 1553–1565 (2014).
4. Chiriac, M. C., Haber, M. & Salcher, M. M. Adaptive genetic traits in pelagic freshwater microbes. *Environ Microbiol* 25, 606–641 (2023).

Finally, could the differences in SP size and diversity be linked to the presence/absence of a motility, attachment and/or chemotaxis apparatus? These apparatus are expected to take up a substantial portion of the secreted proteome and to be under important evolutionary constraints (as potential phage or predator receptor or antigen). This could be tested using bioinformatic prediction tools such as TXSScan for instance (<https://github.com/macsy-models/TXSScan>). Although the presence of these apparatus in larger genomes and their absence in smaller ones would not change the validity of the patterns observed, it would enrich the interpretation of the results and

Response 2.7:

We appreciate the reviewer’s suggestion to explore the relationship between SP and the absence/presence of motility and secretion systems. To address this comment, we conducted an analysis using the KEGG database (see Methods) to validate the absence/presence of such mechanisms. Our findings indicated that larger MAGs exhibited type I, II, III and IV secretion systems with type II (n = 7) being the most prominent one. Additionally, key flagellar proteins were only present in a few of the large genome species (n = 7). On the other hand, small genome species lacked both secretion systems and flagella (Supplementary Information - Fig. S7). However, it is challenging to attribute SP diversity to the presence of secretion systems and flagella since these systems were found to be absent in some of the large genome species.

Realign conclusions with supporting data/results or underlying hypotheses

In a number of places, the authors use strong language to phrase their conclusions, which may not align with the data or results they have to support them. This include, for instance: - Page 2: “Here, we exploit the power of genome-resolved metagenomic analyses to capture the diversity of dominant aquatic bacterial populations in order to assess the adaptive state of bacterial species.” However, assessing the adaptive state of a bacterial species would arguably go beyond what the authors are conducting in this study.

Response 2.8:

To better encapsulate the essence of our research and align it with the reviewer's feedback, we have refined the purpose of our study. The revised statement reads now: 'Here, we exploit the power of genome-resolved metagenomic analyses to assess eco-evolutionary forces shaping the diversity of freshwater bacterial species.' This adjustment ensures a more accurate representation of the study's objectives and brings the conclusions into closer context with the supporting data and their underlying hypotheses.

- Page 3: *"Given that this level of proteome divergence (Fig. 1c) appears to be comparable across various species exhibiting diverse evolutionary histories, lifestyle strategies, and genome sizes, it is highly compelling to attribute it to the fundamental influence of baseline genetic drift."* This statement would suggest that much of the within-species diversity is non-adaptive, which is very challenging to prove and goes against a lot of our understanding of microbial within-species diversity. Please see <https://academic.oup.com/mbe/article/35/6/1338/4976545> or <https://www.nature.com/articles/s41579-020-0368-1> for instance.

Response 2.9:

We appreciate the reviewer's feedback and thank them for highlighting relevant literature. As a response, we have chosen to rephrase and expand upon the paragraph.

At page 1 we introduced the following paragraph:

' While the diversity within species emerges from ongoing processes of variation generation, natural selection, and genetic drift, variability is introduced into populations through mutation and gene flow. Genetic drift operates randomly, leading to the elimination of genetic variation within a population, while natural selection selectively preserves or eliminates variations based on their fitness advantages or disadvantages¹.'

At page 3 we introduced the following paragraph:

One possible interpretation of this observation is that the SPs, which differentiate the two categories, are involved in lifestyle strategies and undergo selection at the niche level, while the similarity in CPs across categories reflects the selection for function. Thus, it becomes apparent that while genetic drift acts uniformly across populations (as it simultaneously purges both CP and SP diversity), selection can operate at different levels and may be driven by different factors: niche adaptation in the case of SPs and the conservation of core metabolic functions in the case of CPs.

References:

1. Van Rossum, T., Ferretti, P., Maistrenko, O. M. & Bork, P. Diversity within species: interpreting strains in microbiomes. *Nature Reviews Microbiology* 2020 18:9 18, 491–506 (2020).

- Page 6: *"This indicates that adaptive stasis as observed here is not an "evolutionary lobster trap" and that small MAGs species only transiently decouple evolution from environmental adaptation." The observation that the difference in SP and CP diversity observed within species does not hold true at the within-genera level is very interesting, and could actually motivate much of the work done by the authors (by reporting it at the beginning thus motivating the follow up analyses). However, as phrased, their interpretation of this difference of diversity as a "transient[...] decoupl[ing] between evolution and environmental adaptation" relies on several assumptions that should be at least discussed before reaching such a strong conclusion.*

Response 2.10:

To address the reviewer's suggestion, we have rephrased the indicated statement. The new version reads: 'This indicates that adaptive stasis as observed here is not an "evolutionary lobster trap" and that small genome species only transiently halt the evolution of their secreted proteomes.'

- Page 6: *"Hence, the apparent evolutionary stasis can be attributed to their secreted proteomes reaching maximum efficiency." Again, a very strong statement, while alternative explanations to the lack of diversity are not discussed and there is no functional data to support the proteome efficiency beyond little positive selection detected in the protein sequences. As mentioned before, there are alternative sources of adaptation, and non-synonymous mutations may actually be adaptive (as also mentioned by the authors).*

Response 2.11:

Following the reviewer's suggestion, we incorporated the results of recombination analyses into the revised manuscript (Figure 4), where we further discussed the role of gene flow in generating variability in bacterial genomes. We acknowledge that there is no supporting data indicating proteome efficiency beyond its high conservation and susceptibility to strong purifying selection. As a result, we have rephrased the sentence to: 'The observed lack of significant evolutionary changes in the secreted proteomes of these organisms can be likely attributed to the notion that these proteomes have already reached an optimal state through the course of evolution, where further major alterations are neither advantageous nor necessary for the organisms' survival and adaptation to their current environments'

- Page 7: *"The inherent inflexibility within their SPs thus restricts the capacity of these organisms to explore novel genetic variations and effectively adapt to dynamic environmental conditions." Same as above, this strong statement relies on the hypothesis*

that any adaptation to the environment is mediated by (non-synonymous) mutation of existing genes rather than acquisition of new genomic fragments for instance without discussing it.

In addition, multiple of these conclusions (including in the title) generalize the results based on 30 freshwater microbial species samples across 51 metagenomes to all genome-reduced bacteria. The authors briefly mention that “further investigation is required to determine its prevalence among genome-reduced bacteria in diverse microbiomes”, yet many of their statements do not reflect that. In practice, the authors should either realign their conclusions with the data they analyze, or support the scale of their statement with new data (e.g. analysis of the SP/CP across species from GTDB database for instance).

Response 2.12:

We appreciate the concerns raised by the reviewer and have undertaken several measures to address them. In response to the inquiry about the impact of recombination on intraspecific diversity generation, we conducted targeted analyses, as elaborated in Response 2.5. Furthermore, we adjusted our approach by minimizing generalizations and restricting our discussion to the data we have generated.

To enrich our dataset, we expanded our search to encompass a broader collection of genomes, metagenome-assembled genomes (MAGs), and single-cell amplified genomes (SAGs) sourced from diverse environmental samples. Despite the relative accessibility of genomes from cultured bacteria, obtaining genome data for genome-reduced representatives posed exceptional challenges. Additionally, the scarcity of species with nine or more genomes in existing databases presented a significant limitation. To address this concern, we conducted a thorough screening of the Genomic Taxonomy Database (GTDB) and consulted various studies reporting extensive genomic datasets. We present the compiled information in Supplementary Table S6.

Despite these efforts, limitations in the available data persisted, primarily due to the common practice of reporting a single representative genome for a species cluster. This practice significantly hinders the feasibility of conducting species-centric studies. Nevertheless, we expanded our range of analyses by compiling a species-centric database containing 592 genomes across 57 species (median number of genomes = 9; Supplementary Information - Fig. 8). This new dataset includes metagenome-assembled genomes (MAGs), single-amplified genomes (SAGs), and cultured representatives, allowing robust cross-validation against our study. In line with the reviewer's suggestion, we chose to concentrate our results and discussion on the meticulously generated, curated, and analysed data. The gained insights, representative for various environmental samples, serve to support our conclusions, and it is important to note that they are not intended to offer a definitive account of bacterial diversity in these environments. We believe this refined approach enhances the robustness and reliability of our study.

Minor/Miscellaneous points:

- The authors describe "51 shotgun sequencing datasets". Clarify whether this refers to 51 independent sequencing datasets (each containing multiple samples) or 51 samples, i.e. 51 metagenomes.

Response 2.13

We appreciate the reviewer's diligence in seeking clarification. In response to the query regarding the description of "51 shotgun sequencing datasets," we have revised the text to enhance clarity. The improved passage now reads: "An extensive database of approximately 5,500 prokaryotic metagenome-assembled genomes (MAGs) was constructed by applying genome-resolved metagenomics techniques to 51 independent shotgun-sequenced samples (Fig. 1,2)."

- There are multiple places where the authors use the phrasing "small species" to refer to species with reduced genomes, however, it is the genome that is small not the species itself.

Response 2.14

We followed the reviewer's recommendation and replaced 'small species' with genome-reduced species throughout the manuscript.

- Page 4, the authors write that "a reduction in the number of genes under negative selection indicates increased selection pressure". However, wouldn't that be the opposite? The more negative selection the more selective pressure?

Response 2.15:

We appreciate the reviewer's insightful observation. While we agree that an increase in negative selection generally indicates higher selective pressure, in the context of our study, the statement 'a reduction in the number of genes under negative selection indicates increased selection pressure' specifically refers to the observed pattern in species with small genome sizes. In this case, the reduction in the number of genes under negative selection is associated with an increase in the number of invariable genes within the SPs. We intended to convey that, despite the typically similar mutational rates for genes coding SPs and CPs and their exposure to similar recombination effects due to their location on the same bacterial chromosome, the observed reduction in genes under negative selection, along with an increase in invariable ones, implies an elevated selection pressure. This indicates that even mutations not affecting amino acid identity within SPs carry a fitness cost. We have rephrased the corresponding sentence in the manuscript to enhance clarity and avoid any potential confusion:

'Given that genes coding for SPs and CPs typically evolve under similar mutational rates (as being located on the same bacterial chromosome) and are being therefore subjected to

similar recombination effects, a reduction in the number of genes under negative selection compensated by the increase in the invariable genes indicates enhanced selection pressure¹. This suggests that even mutations that do not impact the amino acid identity within the SPs incur a fitness cost.'

Reference:

1. Rocha, E. P. C. Neutral Theory, Microbial Practice: Challenges in Bacterial Population Genetics. *Mol Biol Evol* 35, 1338–1347 (2018).

- Page 4, the authors mention that synonymous mutations may be non-neutral. See this <https://academic.oup.com/mbe/article/35/6/1338/4976545> for additional discussion on the lack of neutral sites in bacterial genomes.

Response 2.16:

We appreciate the reviewer for highlighting this work. In response, we've integrated the referenced study into our data interpretation, as discussed in Response 2.15.

- In figure 2a, the authors report the % of genes under positive, negative selection and those that are invariable. Wouldn't the invariable genes be under negative selection as well? Or at minima, there is no variation and thus no way to test for the selective regime? As such, interpreting an increase in invariant genes (and thus a decrease in negatively selected genes) as less negative selection seems misleading. Possibly, the ratio of positively selected genes vs negatively selected genes would be a better approach to interpret this compositional dataset. In which case the ratio would increase from CP to SP for large genomes (indicating positive selection in the SP) while it would largely decrease for small genomes (indicating strong negative selection).

Response 2.17:

Invariant genes exhibit 100% sequence identity. A comparison of the proportion of invariant genes between the SPs and CPs of small-genome species reveals an increase in the former. With only a minimal percentage (0.2%) of SP genes under positive selection, this increase primarily stemmed from a reduction in the number of genes under negative selection. It is important to note that we do not interpret this as a decrease in negative selection pressure, but rather we see it as an augmentation of the influence of negative selection. The paragraph of the text was rephrased to enhance its clarity:

'Given that genes coding for SPs and CPs typically evolve under similar mutational rates (as being located on the same bacterial chromosome) and are being therefore subjected to similar recombination effects, a reduction in the number of genes under negative selection compensated by the increase in the invariable genes indicates enhanced selection pressure.'

We do not believe that creating ratios between the number of genes under positive and negative selection would offer a more accurate depiction in our context. This is because a decrease in the number of genes under negative selection (accompanied by an increase in invariable ones) might be misinterpreted as an increase in positive selection. To illustrate, let's consider two hypothetical genomes: A, with 20 genes under positive selection, 80 under negative, and 100 invariable; and B, with 5 genes under positive selection, 20 under negative, and 175 invariable. The ratio of positively selected genes to negatively selected genes is 0.25 for both cases, suggesting similar selection pressure. However, genome B is experiencing heightened selective pressure.

- In figure 2b, the number of sites under negative selection per protein will be highly dependent on the protein length. Although the length distribution is reported in Figure 2c, it would be better practice to report a normalized value in figure 2b.

Response 2.18:

We appreciate the reviewer's comment. While it is theoretically plausible that longer genes may contain more sites under negative selection due to a slightly higher probability of mutations in a longer DNA sequence, it's important to consider that coding DNA sequences are under selective pressure. In the context of natural selection, the gene functions as a unit of selection, emphasizing the importance of the encoded functionality^{1,2} rather than solely the gene's length. Therefore, our analysis focuses on the gene as a functional unit subjected to selective pressures rather than emphasizing the correlation with its length. Moreover, the metric of number of sites per gene is widely utilized in the literature, and we prefer to maintain consistency with this convention to facilitate comparisons with previous studies.

References:

1. Ågren JA. Selfish genetic elements and the gene's-eye view of evolution. *Curr Zool.* 2016 Dec;62(6):659-665. doi: 10.1093/cz/zow102. Epub 2016 Oct 23. PMID: 29491953; PMCID: PMC5804262.
2. Williams, George C, 'The gene as a unit of selection', *Natural Selection: Domains, Levels and Challenges* (New York, NY, 1992; online edn, Oxford Academic, 31 Oct. 2023).

- The functional redundancy analysis is based on KO annotations. However, the percentage of genes with KO annotations may be low and variable across genomes/species. This may weaken the estimations computed here. A per species protein clustering approach (either using the orthologous clusters directly, or de novo less stringent clustering using e.g. mmseqs2) would allow to compute similar values by dividing the number of clusters by the number of proteins independently of annotation rates.

Response 2.19:

We understand the concerns raised by the reviewer regarding the functional redundancy analysis based on KO annotations. In response to the query regarding the percentage of genes with KO annotations, we calculated the percentage of successful annotations based on KEGG Database for large (SP: 51.19%; CP: 69.36%) and small (SP:62.3%; CP: 76.26%) MAGs. We also followed the reviewer's suggestion to perform a protein clustering approach using mmseqs2 as well as well as OrthoFinder software.

Clustering proteins using a *de novo* similarity-based approach presents challenges, particularly when attempting to group them based on function. This difficulty arises from the existence of paralogous genes that may share overlapping functions and genes acquired through recombination events. For instance, when clustering our proteomes with an 80% identity threshold (anticipating that proteins from the same species generally exhibit $\geq 80\%$ identity), we observed the **cpo non-heme chloroperoxidase** (K00433; EC:1.11.1.10) protein appearing in two distinct clusters within the genome RE-4nov15-139. These two protein variants from the RE-4nov15-139 MAG display a 17% sequence identity. The functional and taxonomical annotations of these proteins were manually verified and cross checked with CDD and NCBI databases.

A similar pattern was noted for the **ABCB-BAC ATP-binding cassette subfamily B** (K06147) proteins, found in two clusters at 71% identity in the RE-4nov15-139 genome, as well as the **livF branched-chain amino acid transport system ATP-binding protein** (K01996), present in two copies across eight genomes with a 46% identity between copies. It's crucial to highlight that the annotations and taxonomy assignments of these proteins underwent meticulous manual verification.

In the instance of RE-4nov15-139, we need to cluster the proteome at a 17% identity threshold to encompass the cpo non-heme chloroperoxidase gene within the same cluster. Conversely, for the livF branched-chain amino acid transport system ATP-binding protein in RE-4nov15-139, clustering at 71% identity is necessary. These examples underscore the challenge of establishing a universal threshold that can capture all proteins with the same function in a cluster. Setting the threshold too permissive will result in including proteins with distinct functions in the same cluster, while setting it too restrictive will lead to proteins with the same function residing in separate clusters.

When employing the OrthoFinder approach at both the SP and CP levels across six species (comprising three large and three small-genome ones), we observed instances where specific genes that have same function were present in at least two orthogroups. This identification of orthogroups within the mentioned categories was accomplished using OrthoFinder software, as outlined in the methods section, with default settings.

Subsequently, the obtained orthogroups were annotated using BlastKOALA with default options. Noteworthy examples include:

Nanopelagicaceae sp.1:

At the SP level, the livK proteins, responsible for the branched-chain amino acid transport system substrate-binding protein, were distributed across two orthogroups. A similar pattern was observed for eight other proteins having identical functions. Despite attempting to relax the MCL index for more permissive clustering of orthogroups, the approach did not resolve the issue, and proteins with the same function continued to be associated with distinct orthogroups.

CP level: A total of 36 functions (i.e. identical annotations) were found to be distributed between two or more orthogroups.

Planktophila sp.10:

SP level: Nine functions were found to be split into two distinct orthogroups.

CP level: Forty-six functions were associated with at least two orthogroups each.

Illumatobacteraceae sp.11:

SP level: The fabG 3-oxoacyl-[acyl-carrier protein] reductase (K00059) protein and 6 others were identified in two distinct orthogroups.

CP level: 72 functions were associated with at least two orthogroups.

Hence, we posit that, while not flawless, the most effective approach to identify proteins with similar functions is through annotations. We present the percentage of annotated proteins in the SPs and CPs of small and large genome species, along with a discussion of the inherent challenges, in our Limitations chapter.

- Although I understand the attractiveness of dividing the dataset between species with large and small genomes, could it be more powerful to use the genome size as a continuous explanatory variable?

Response 2.20:

While we appreciate the reviewer's suggestion to use the genome size as a continuous explanatory variable, our decision to categorize the MAGs as belonging to large and small genome species serves specific purposes: i) it enhances the reader's comprehension throughout the study, and, more importantly ii) it allows a focused comparison between two distinct bacterial lifestyle strategies. Genome-reduced bacteria are optimised for

nutrient intake in low-nutrient environments and possess distinct genomic features such as high coding density, small intergenic-spacers and reduced metabolic capability¹. Contrastingly, their larger genome counterparts possess a rich metabolic repertoire and a variety of means to interact with their niche. This duality is highlighted when comparing the percentage of SP and CP between both categories (Figure 3a). Large genome species display a higher proportion of SP proteins compared to small genome species. This distinction becomes evident at the boundary between the categories illustrated by Chitinophagaceae sp. 4 and Chloroflexota sp.1 (Figure 3a), underlining the need for a distinct separation of large and small genome species.

To further elucidate the impact of this categorization on the SP/CP ratio, we performed a similar analysis as the one described in Response 2.3. Here, the genome size category was employed as an explanatory variable to explain the SP/CP ratio. Our analysis, guided by the Akaike Information Criterion (AIC) to balance goodness of fit and model complexity, revealed that being categorized as small or large has a significant effect on the SP/CP ratio observed (AIC= -167.85; Adjusted r-squared = 0.54, p-value= 3.206e-05). These results accentuate the need and impact of genome categorization in enhancing the robustness and clarity of our study.

References:

1.Giovannoni, S., Cameron Thrash, J. & Temperton, B. Implications of streamlining theory for microbial ecology. ISME J 8, 1553–1565 (2014). <https://doi.org/10.1038/ismej.2014.60>

One could then correlate the difference between SP and CP diversity or the ratio of negatively selected and positively selected sites by the genome size across the 30 species.

Response 2.21:

We appreciate the suggestion to explore the correlation between SP and CP diversity, as well as the ratio of negatively and positively selected sites compared to genome size. Upon examining the correlation between SP similarities and genome size, a robust negative correlation was evident ($r = -0.51$, p-value = 0.003), while a weaker correlation for CP similarities in relation to genome size was observed ($r = 0.30$, p-value = 0.009). Additionally, the SP/CP ratio showed a strong negative correlation ($r=-0.552$, p-value = 0.001) against the genome size.

To further explore the relationship between the number of negatively/positively selected sites and genome size, species specific medians were computed. No significant correlation was found for negatively selected sites within both the SP ($r = 0.16$, p-value = 0.39) and the CP ($r = 0.038$, p-value = 0.83) categories. Moreover, it was not possible to analyse the correlation between the sites under positive selection and the SP/CP levels due to the fact that many median sites under positive selection per species proved to be zero. We emphasize that, in order to capture the impact on genome size across the 30 species, we

opted to correlate the SP/CP ratio with coding density, as it serves as a surrogate of genome size. To validate coding density as a valid approximation for genome size, we assessed their correlation, confirming their close relationship (Supplementary Information - Fig. 2; $r = -0.63$, $p\text{-value} = 2.2e-16$). This method allows us to capture the effect of genome size without relying on the discussed redundant relationships. It is noteworthy that, unlike genome size (which was not completely recovered for all the species in our dataset), coding density is genome completeness independent. Thus, as emphasized in our previous answer, we would like to keep the designated genome size categories as they reflect contrasting eco-evolutionary strategies.

- The observation that the SPs diversity is high between genera but not within species for genome-reduced bacteria is very intriguing and interesting. Could this be linked species with smaller genomes having higher effective population sizes? Genome-reduced bacteria often have large population size (Pelagibacter, Prochlorococcus, acI, ...), and therefore a more efficient selection with less room for drift, potentially reducing the emergence of non-adaptive diversity while rapidly allowing the fixation the adaptive innovations. On the contrary copiotrophs with large genomes may undergo more bloom-like events that would favor the emergence of non-selective processes.

Response 2.22:

We appreciate the reviewer's suggestions. However, we have observed temporal fluctuation patterns within the designated species clusters. While, in general, the total abundance tends to be higher for genome-reduced species than for those harbouring larger genomes (Figure 3), the situation is more intricate than it seems. Some large genome species, such as Burkholderiaceae sp.4 and Lacunisphaera sp.3, maintain large abundances, while others exhibit abundance peaks (e.g., Opiritales sp.4) or maintain consist low abundances (e.g., Rhodoferax sp.2). Similarly, small genome species display fluctuation patterns as well. For instance, Nanopelagicaceae sp.8 experiences more than a 10-fold increase over the sampling time (Supplementary Table S2). Hence, considering the intricate and to some extent differential nature of abundance patterns, we prefer to refrain from associating the observed diversity patterns of SPs/CPs with species abundances.

- In "Assembly, binning, and MAG designation" the authors state that the "metagenomic datasets were mapped [...] against the assembled contigs [...] in a sample-dependent fashion.". Please clarify whether the reads from all metagenomic samples were mapped against the contigs from all samples, or if only the reads from a given sample were mapped to the matching assembly. In particular, the approach leveraging the abundance of contigs across multiple samples has proven important to improve the quality of the MAGs reconstructed (see <https://www.nature.com/articles/s41592-023-01934-8> for a most recent evaluation).

Response 2.23:

We appreciate the reviewer's reference to the mentioned paper. It is crucial to highlight that we are fully aware of the improved binning quality achieved through multiple abundance data points. For clarity, it is essential to emphasize that each metagenome was individually mapped against all the obtained assemblies. The revised sentence is as follows: '... in a sample-dependent fashion, ensuring that each metagenomic dataset was specifically mapped against all assemblies.'

Furthermore, additional correction steps were taken to ensure high binning quality by performing post-binning curation, as described on pages 16-17.

- Please add a data availability statement referencing the bioproject ids at which the metagenomes and MAGs can be accessed.

Response 2.24:

We value the reviewer's thoroughness and would like to address the raised concern. To maintain the double-blind review process, critical sections like Data Availability, Funding, Authors Contributions, etc., were incorporated into the submission cover letter. It is important to note that all data, including metagenomic and genomic information, was published prior to submission. The journal has been provided with the relevant accession IDs and details of the public repositories housing the data.

Reviewer #3 (Remarks to the Author):

The authors have produced a large set of metagenome-assembled genomes (MAGs) of bacteria and archaea from freshwater bodies in Central Europe and used it to investigate relative rates of adaptation among microbial taxa. Their analyses indicated elevated selective pressure on secreted proteins (SPs) in small MAGs, as compared to large MAGs. This was interpreted as a proof of an adaptive stasis in microbial taxa with small genomes, caused by the reduced functional redundancy in their genomes (no "backup plan"). This study addresses important questions in microbial evolution and presents a compelling hypothesis that deserves robust testing. However, I believe that the type of core data used in this study - >5,500 MAGs – is prone to major methodological biases and may be fundamentally unsuitable for the type of analyses performed here. While MAGs are very instrumental in the recovery genomic information from yet uncultured taxa, they are highly prone to both the inclusion of genome regions from unrelated organisms (chimerism) and exclusion of hypervariable genome regions (pangenome under-sampling). Importantly, this not reflected by genome completeness estimates by CheckM and similar tools, which simply count the presence of a small subset of core genes. For a few studies exploring these

methodological challenges in great detail, see the following:

1 Szczyrba, A. et al. *Critical Assessment of Metagenome Interpretation - A benchmark of metagenomics software*. *Nature Methods* 14, 1063-1071 (2017).

<https://doi.org:10.1038/nmeth.4458>

2 Vollmers, J., Wiegand, S., Lenk, F. & Kaster, A. K. *How clear is our current view on microbial dark matter? (Re-)assessing public MAG & SAG datasets with MDMcleaner*. *Nucleic Acids Res* (2022). <https://doi.org:10.1093/nar/gkac294>

3 Orakov, A. et al. *GUNC: detection of chimerism and contamination in prokaryotic genomes*. *Genome Biol* 22, 178 (2021). <https://doi.org:10.1186/s13059-021-02393-0>

Response 3.1:

We appreciate the thoughtful comment from the reviewer and their reference to pertinent work in the field. We acknowledge the intrinsic challenges linked to metagenome-assembled genomes (MAGs), particularly in the context of potential pangenome undersampling.

It is important to note that pangenome undersampling is a shared concern affecting not only MAGs but also single-amplified genomes and cultured isolates. The limitations stem from the inherent difficulty in capturing the entirety of species diversity, a challenge exacerbated by the continuous evolution of bacterial populations across time. To address some of the mentioned challenges, our study employed repeated sampling campaigns in the selected environments. This iterative sampling method significantly bolstered our capacity to capture some of the species' flexible genomes. To enhance our capability to investigate the flexible genome, we meticulously curated our pdCEL database, prioritizing species clusters with a minimum of 9 representatives. This rigorous selection process was devised to guarantee a thorough representation of species clusters and their flexible genomic elements. Additionally, we conducted a pangenome analysis and introduced a 'Limitations' section into the manuscript, transparently addressing some of the methodological constraints associated with species diversity analysis. Additional comments regarding the concerns about pangenome undersampling are addressed in Response 3.2.

In terms of computational methodologies, we highlight that MEGAHIT distinguished itself as one of the top-performing software in the CAMI challenge¹ indicated by the reviewer, showcasing superior contiguity and successful assembly of a substantial fraction of genomes across diverse coverages. Notably, it garnered the highest scores in recovering both unique strains (genomes with <95% ANI to any other in the dataset) and common strains (genomes with an ANI ≥95% to another genome in the dataset). Also, the MetaBAT binning software demonstrated high accuracy, and while MetaBAT2 (which was used) wasn't explicitly tested, we capitalized on its performance by integrating it with a comprehensive dataset of abundance data known to enhance the effectiveness and sensitivity of multi-coverage metagenomic binning software². Our selection of MetaBAT2 is based on its performance as

the best-performing single binner in independent benchmarks³ and its efficiency, being 10 to 50 times faster than other commonly used binners⁴. Additionally, we emphasize that a meticulous step of metagenomic binning curation was implemented to avoid misbinning of taxonomically unrelated contigs, as outlined in the Methods section.

When evaluating our analysis workflow, it is essential to state that the observed consistency between SP and CP diversity underscores the absence of major methodological flaws in our data generation process. The uniformity in our data generation protocols ensure that any displayed variation between these two protein categories indicate substantive biological differences rather than methodological artifacts. To further enhance the robustness of our methods, we conducted the following additional analyses:

A) Performed a benchmark for intraspecific protein diversity analysis utilizing the cultivated representatives, SAGs, and MAGs of some environmental species available (Supplementary Information - Fig. 3). The comparison of the SP/CP ratios among genomes originating from cultures, SAGs, and MAGs, revealed comparable results. Especially, culture-genomes and MAGs from the same species display a highly similar pattern. The slight deviation regarding the species-specific SP/CP ratio of MAGs compared to that of SAGs is likely attributable to errors that were introduced during the single-cell amplification steps. This comparative analysis supports the reliability of our MAG collection to draw meaningful conclusions about species diversity and evolution.

B) Executed a benchmark for DNA extraction, sequencing, and assembly (Supplementary Information - Table 1). Presented also in Response 3.4. See also Method section.

C) Analysed 592 genomes belonging to 57 environmental species (median number of genomes/species = 9) and identified similar diversity patterns in CPs and SPs diversity as observed in our generated datasets (Supplementary Table S6).

References:

1. Szczyrba, A. et al. Critical Assessment of Metagenome Interpretation—a benchmark of metagenomics software. *Nature Methods* 2017 14:11 14, 1063–1071 (2017).
2. Mattock, J. & Watson, M. A comparison of single-coverage and multi-coverage metagenomic binning reveals extensive hidden contamination. *Nature Methods* 2023 20:8 20, 1170–1173 (2023).
3. Hofmann, F. M. P. et al. AMBER: Assessment of Metagenome BinnERs. *Gigascience* 7, 1–8 (2018).
4. Meyer, F. et al. Critical Assessment of Metagenome Interpretation: the second round of challenges. *Nature Methods* 2022 19:4 19, 429–440 (2022).

Pangenome under-sampling is particularly relevant here, as it may introduce major biases leading to flawed conclusions. As an illustration, despite two decades of substantial efforts, marine microbiologists still struggle to produce MAGs of Pelagibacter and Prochlorococcus, the two most abundant genera in the ocean, both of which have small genomes. Meanwhile, it is well known that pangenomes of Pelagibacter and Prochlorococcus are enormous, with ample evidence for frequent horizontal gene transfer and recombination. Thus, the situation with Pelagibacter and Prochlorococcus is the complete opposite to the adaptive stasis proposed in the manuscript: they evolve rapidly and recombine their genomes frequently, challenging scientists' ability to represent their populations in MAGs.

Response 3.2:

We appreciate the reviewer's observation regarding the potential misinterpretation of undersampled pangenomes as adaptive stasis, particularly in small genome species like Prochlorococcus and Pelagibacter, which exhibit substantial variability.

However, it is crucial to note that our study focused on common freshwater bacterial species, which may not necessarily exhibit the same patterns. Furthermore, we want to underscore that the genera **Pelagibacter** and **Prochlorococcus** encompass **855** and **332 species**, respectively (according to the GTDB database as of November 2023), contributing to the extensive genus-level pangenome mentioned by the reviewer. However, we want to emphasize that our analysis specifically targeted intraspecific rather than intragenic diversity (as was the case for Pelagibacter and Prochlorococcus).

Recombination events, such as horizontal gene transfer, play a pivotal role in shaping the complexity and size of microbial pangenomes¹. In accordance with the Neutral Theory outlined in "Microbial Practice: Challenges in Bacterial Population Genetics"², these genetic variations can induce neutral, deleterious, or adaptive effects within genomes of bacterial species. Despite reports suggesting that interspecies variability can enhance a species' adaptive potential³, conflicting patterns have been reported by other studies^{4,5}. One plausible conclusion that can be drawn from previous work is that species exhibit different, species-dependent pangenome sizes. Also, the accessory genes within them exert contrasting effects on the species' adaptation to their environment.

Since we cannot fully claim the inherent completeness nor adaptability of the pangenomes of the analysed species, we can only showcase the variability observed through sampling numerous ecosystems across Europe, encompassing diverse seasonal conditions and physicochemical parameters. To specifically address the reviewer's comment, we analysed the pangenome of each species-cluster using Roary as mentioned in the methods section (Supplementary Information Fig. S9). We find that the number of genes belonging to the pangenome of each species increases with each MAG added, while the rate of gene addition per genome decreases. This indicates the incomplete recovery of the total amount of

accessory genes for each species but also shows that interspecies variability has been caught. Simultaneously, the slow saturation of newly added genes suggests that none of the presented species-clusters is greatly suffering from pangenome undersampling (Supplementary Information Fig. S9). While the variability persists across all species groups featuring 9 to 19 MAGs, the conserved structures within the sampled variability, notably within the secreted proteome of small MAG species, supports the assumption that such specialized species might indeed exploit stasis as a successful ecological strategy.

References:

1. Medini, D., et al., *The microbial pan-genome*. *Current Opinion in Genetics & Development*, 2005. 15(6): p. 589-594.
2. Rocha, E.P.C., *Neutral Theory, Microbial Practice: Challenges in Bacterial Population Genetics*. *Molecular Biology and Evolution*, 2018. 35(6): p. 1338-1347.
3. Sela, I., Y.I. Wolf, and E.V. Koonin, *Theory of prokaryotic genome evolution*. *Proc Natl Acad Sci U S A*, 2016. 113(41): p. 11399-11407.
4. Baumdicker, F., W.R. Hess, and P. Pfaffelhuber, *The Infinitely Many Genes Model for the Distributed Genome of Bacteria*. *Genome Biology and Evolution*, 2012. 4(4): p. 443-456.
5. Conrad, R.E., et al., *Toward quantifying the adaptive role of bacterial pangenomes during environmental perturbations*. *The ISME Journal*, 2022. 16(5): p. 1222-1234.

In order to make their data more interpretable, authors of the manuscript should evaluate how accurately their MAGs represent the genomic composition of analyzed microbial communities. The following is essential:

1. *How accurate are the metagenome assembly and binning methods used in the study? This can be addressed by validating the entire workflow with mock metagenomes, such as those used in the Sczyrba et al. publication cited above and publicly available.*

Response 3.3.1:

We thank the reviewer for pointing out that demonstrating the accuracy of the data processing pipeline adds weight to our analyses, especially when it comes to the genomic composition of environmentally-derived MAGs.

To validate the accuracy of the method, DNA was extracted from the **Microbial Community Standard ZymoBIOMICS D6300** (Zymo Research Inc., Irvine, CA, USA) and sequenced with Illumina, employing the same protocols for DNA extraction and library preparation as outlined in the Methods section. The data was curated and assembled using the identical procedures as described for the environmental samples to confirm the reliability of the

analysis. The mock community comprises 10 different species, including 8 bacteria (3G+, 5G-). After quality trimming, approximately 100 million paired-end reads (2x150bp) were retained (>Q18). Megahit (v.1.2.9) was employed with default options to assemble the quality-filtered data, utilizing various kmer sizes (39, 49, 69, 89, 109, 129, 149). Due to the unique nature of the mock community and the binner (MetaBAT2) relying on differential abundance data from sampled time series, replicating the exact binning strategy used in the manuscript was not feasible. Further details regarding the binning are provided in Response 3.1.

For binning the mock community, we implemented a taxonomy-based approach similar to the one employed for the post-binning curation described in the manuscript for the environmental data. In essence, each contig with a length exceeding 3,000 bp was utilized to predict proteomes (PRODIGAL33 v2.6.3), which were queried (using mmseqs2 searches) against the curated prokaryotic GTDB database (R05-RS95). The results were then converted into a BLAST-tab formatted file (using mmseqs convertalis), from which individual top hits were extracted. The taxonomic labels derived from these hits were used to annotate the queried proteomes, and the taxonomic information of the contigs was employed for binning.

The accuracy of DNA extraction, sequencing, and metagenomic assembly was evaluated by calculating the average nucleotide identity between the generated draft genomes and the provided reference genomes of the ZymoBIOMICS Microbial Community Standard D6300. **The draft genomes of the eight bacteria exhibited a median average nucleotide identity of 99.98%, with a median coverage value of 96.425% compared to the reference.** This highlights the precision and reliability of the applied method.

ZymoBIOMICS.STD	SequencingTechnology	Avg_ANI (%)	Avg_Cov (%)
Bacillus sp.	Illumina	100	97.91
Enterococcus sp.	Illumina	100	97.17
Escherichia sp.	Illumina	99.95	84.95
Lactobacillus sp.	Illumina	99.99	91.75
Listeria sp.	Illumina	99.99	97.25
Pseudomonas sp.	Illumina	99.89	96.97
Salmonella sp.	Illumina	99.92	87.04
Staphylococcus sp.	Illumina	99.97	95.88

2. What fraction of genes encoding secreted and cytoplasmic proteins by the entire community is represented by the obtained MAGs? This can be done by the recruitment of individual metagenome reads on MAGs.

Response 3.3.2:

We appreciate the reviewer's suggestions. To address the concern, we conducted normalized read mapping for each of the approximately 5,500 metagenome-assembled

genomes (MAGs). In Figure 3, we included an inset that illustrates the total abundance of the targeted species clusters (342 MAGs) in relation to the total abundance of all MAGs obtained. Individual abundances are now available in Supplementary Table S2.

3. How do metagenome assembly algorithms, which produce consensus sequences from a multitude of organisms, impact the ability to measure evolutionary pressure metrics, such as Dn/Ds ratio? This can be done by software designed to explore intra-population variation in MAGs, such as inStrain: Olm, M. R. et al. inStrain profiles population microdiversity from metagenomic data and sensitively detects shared microbial strains. Nat Biotechnol 39, 727–736 (2021).

Response 3.3.3:

We appreciate the reviewer's concerns regarding the potential influence of metagenome assembly algorithms on evolutionary pressure metrics such as the ratio between non-synonymous and synonymous mutations. In response to these concerns, we followed the reviewer's suggestions and employed the **inStrain** software¹ to evaluate the impact of existing microdiversity on our selection pressure metrics.

Considering the expected microdiversity within most species, we intentionally chose one metagenome, specifically RH-22June17, to investigate the species-level microdiversity of the metagenome-assembled genomes (MAGs) derived from this sample. This sample allowed us to investigate the microdiversity of 23 MAGs that are organized in 23 of the 30 species-clusters discussed in the manuscript. From the resulting 7 species-clusters, no MAG could be recovered from that respective sample. Our examination focused on the following key metrics, including:

- i) **Nucleotide Diversity:** Examining the overall genetic variation within the 23 selected species.
- ii) **Divergent Site Counts:** Representing the number of positions with single nucleotide variations or substitutions, providing insights about the genomic diversity.
- iii) **pN/pS Ratio:** a measure of whether the set of mutations in a gene are biased towards synonymous (S) or non-synonymous (N) mutations. The pN/pS ratio is calculated based on the mutations that are at least present on two alleles. $pN/pS > 1$ shows bias towards N mutations, indicating that the respective gene is under active positive selection and acquires mutations. $pN/pS < 1$ represents a bias towards S mutations, indicating that the gene is under stabilizing (negative) selection and does not accept mutations leading to structural changes in the amino acid sequence. $pN/pS = 1$ shows that N and S mutations randomly occur at the expected rate, potentially indicating that the respective gene is non-functional.

We consider these metrics as robust indicators for capturing microdiversity within our specified species.

Concerning nucleotide diversity, we observed no significant differences between SPs and CPs diversity (Pairwise Wilcoxon Rank Sum Tests, p-value = 0.36). Although a slightly higher diversity was generally noted for small genome species, these differences did not reach statistical significance (Supplementary Information – Fig. S5a). A similar pattern was observed in the case of the number of divergent sites per gene, with no statistical differences between SPs and CPs (Pairwise Wilcoxon Rank Sum Tests, p-value = 0.36) (Supplementary Information – Fig. S5b) or between genome-size categories (i.e., large and small). However, the pN/pS results presented a highly consistent pattern with the conducted evolutionary pressure analyses (Supplementary Information - Figure S5c). Most of the within-species diversity is under negative selection. Furthermore, the genes encoding the SPs of the small genome species exhibit very low levels of positive selection (Supplementary Information - Figure S5d). Therefore, we assert that extant microdiversity does not introduce major biases into our results; rather, it reinforces our confidence in the reliability of the employed methods.

We further extracted 100 million reads from the aforementioned metagenome (i.e., RH-22June17) and calculated their similarity using the blastn algorithm against the 23 metagenome-assembled genomes (MAGs), each representing a distinct bacterial species. Subsequently, we generated distribution and recruitment plots based on the compiled dataset (Supplementary Information - Figure S6). Our analysis indicates that while microdiversity is discernible, the majority of the reads are distributed within clonal complexes, denoted by more than 99% sequence similarity with the analysed MAGs. Considering our comprehensive benchmarking of DNA extraction, sequencing, assembly, and microdiversity analyses, we maintain that our methodology remains robust despite the presence of microdiversity. **Furthermore, we argue that if microdiversity indeed had a significant effect on our results, we would not have observed the evident distinctive patterns between large and small genome species, as well as between SPs and CPs.**

Although the investigation of microdiversity distribution across different species over time and space is an intriguing subject in its own right, it extends beyond the scope of the presented study.

Reference:

1. Olm, M.R., Crits-Christoph, A., Bouma-Gregson, K. *et al.* inStrain profiles population microdiversity from metagenomic data and sensitively detects shared microbial strains. *Nat Biotechnol* **39**, 727–736 (2021).

Reviewers' comments:

Reviewer #1 (Remarks to the Author):

My previous comments have been addressed satisfactorily. The resolution of some figures needs to be improved.

Reviewer #2 (Remarks to the Author):

I appreciate the authors' efforts to respond to my and other referees' comments and to update their manuscript accordingly. I believe the revisions have largely addressed my concerns and have improved the clarity and quality of the manuscript. Although I commend the authors for their work, I would also like to raise a few remaining points:

In response to my comments, the authors show that some functions are specific to the SP of species with large genomes (including secretion systems and flagellum). This raises the question of whether the pattern observed between the SPs of species with large and small genomes could be driven by a subset of function specific to the SP of large species. Conversely, does the pattern observed here hold true when focusing on functions that are shared by the SPs of both species with large and small genomes?

The authors assume that genes located on the same chromosome are under similar mutation rates (page 3, line 34). However, bacteria are known to have specific loci that are highly mutable ([https://doi.org/10.1016/S0960-9822\(00\)00005-1](https://doi.org/10.1016/S0960-9822(00)00005-1)) and there mounting evidence that the mutation rate is not constant across the chromosome but rather exhibits symmetry around the origin of replication (<https://journals.asm.org/doi/full/10.1128/mbio.01226-19>). The authors should better state the limitations of their hypothesis, or test whether CPs and SPs are equally distant from the origin of replication both in species with small and large genomes.

In figure 5a, the genes with no variations are labeled as "genes under no selection". I believe it should rather read "genes with no variation" based on the authors' response to my previous comments.

The authors have included estimates of growth rates, including on the basis of Codon Usage Biases (page 5, line 3). This is a welcome addition and I would invite the authors to consider citing original work on the codon usage imprint of growth: <https://doi.org/10.1371/journal.pgen.1000808>.

Reviewer #3 (Remarks to the Author):

3.1.

In their response, the authors have essentially ignored my concerns and made no adjustments to the manuscript.

The authors misunderstand the nature of incomplete genome recovery in various data types when using short read sequencing. In isolates, genome incompleteness is primarily caused by assembly failures in repeat regions, which usually comprise a minor part of microbial genomes. In SAGs, the main source of genome incompleteness is uneven DNA amplification, which is a random process with no known biases for or against certain genome regions. In MAGs, it has been well known that the less variable genome regions are more readily recovered, while the flexible part of the genome gets underrepresented. This is a major issue for the type of study that is presented in the manuscript, and it has to be properly addressed before making any conclusions about differences in the flexible gene composition among various microbial lineages. To be honest, I doubt that this is addressable, as the type of data appears to be at odds with the hypotheses tested. For a recent discussion about flexible gene underrepresentation in MAGs, see:

- Kerkvliet, J. J., Bossers, A., Kers, J. G., Meneses, R., Willems, R. & Schürch, A. C. Metagenomic assembly is the main bottleneck in the identification of mobile genetic elements. *PeerJ* 12 (2024). <https://doi.org/10.7717/peerj.16695>

There is no mentioning of this core data limitation in either the original, or the revised manuscript. The fact that same techniques were applied on all datasets does not solve the problem, as claimed by the authors, because different proportions of genome may be missing in different MAGs. The intercomparison of MAGs, isolates and SAGs, provided in Figure 3, appears extremely limited in scope and hard to interpret, given the fact that there is no associated method description or even reference to this figure from the manuscript text.

The rebuttal makes a statement “Executed a benchmark for DNA extraction, sequencing, and assembly (Supplementary Information - Table 1). However, this table is just a list of MAGs used in the study.

The rebuttal also states “Analysed 592 genomes belonging to 57 environmental species (median number of genomes/species = 9) and identified similar diversity patterns in CPs and SPs diversity as observed in our generated datasets (Supplementary Table S6).” However, Table S6 contains only a list of these genomes. Contrary to authors’ claims, I was not able to find any information on the “diversity patterns in CPs and SPs diversity” of these genomes in Table S6 or anywhere else in the manuscript.

In their rebuttal, the authors refer to results of a 2017 CAMI, which found MEGAHIT and MetaBAT outperforming some of the other algorithms for metagenome assembly and binning, as a proof of the robustness of their MAG generation procedure. However, the 2017 CAMI study found that the MEGAHIT-MetaBAT combo had <20% recall (genome recovery) and <60% precision (the inverse of contamination) when processing “high complexity” mock metagenomes (although the complexity of these materials was likely much lower than the natural freshwater microbiome). In order to properly interpret results presented in this manuscript, these methodological limitations should be given full, quantitative consideration.

3.2.

The rebuttal makes no logical sense. A saturation of gene recovery with each additional MAG is expected to be as prevalent or even more pronounced if such MAGs underrepresent the flexible pangenome.

3.3.1.

The authors did substantial additional work benchmarking their workflow, which is praiseworthy. However, they have not addressed the core question: can the type of data used in the manuscript accurately discriminate evolutionary patterns of specific proteins within specific species-level lineages? This is not easy, but one of the best approaches may be by testing whether these patterns can be accurately represented in MAGs produced from the “high complexity” mock metagenomes of 2017 CAMI or similar datasets, containing multiple strains of the same microbial species. Instead, the authors analyzed a low-complexity mock metagenome with only one representative per species and did not even try to evaluate the accuracy of their evolutionary metrics when applied on this dataset. In fact, results of this analysis do not appear to be included in the manuscript at all.

3.3.2.

I appreciate the authors’ effort to address my comment. Unfortunately, neither Figure 3, nor Table S2 contain quantitative, interpretable information. The inset in Figure 3 lacks legend, while bar charts in Table S2 lack scale.

3.3.3.

I find this rebuttal highly confusing. For example, the authors state that “no MAG could be recovered from that respective sample” right after saying that that same sample “allowed us to investigate the microdiversity of 23 MAGs that are organized in 23 of the 30 species- clusters discussed in the manuscript”. Even more confusing is the concluding statement: “Although the investigation of microdiversity distribution across different species over time and space is an intriguing subject in its own right, it extends beyond the scope of the presented study.” The manuscript specifically concerns genetic variation within microbial species, i.e., microdiversity, so ignoring it is an oxymoron. The figures referenced in this rebuttal contain no results relevant to my comment.

Dear Reviewers,

In response, we have prepared a detailed appeal document, alongside the rebuttal we initially submitted. This document is intended to transparently demonstrate how we have constructively addressed (point by point) each critique and comment in our rebuttal.

We believe that the concerns raised, while valuable, may stem from certain misunderstandings or a need for further clarification on our part, which we have endeavored to provide.

The reviewer's statements are quoted in italics, followed by our responses in regular blue text.

Reviewers' comments:

Reviewer #1 (Remarks to the Author):

Comment 1.1: My previous comments have been addressed satisfactorily. The resolution of some figures needs to be improved.

Response 1.1: We sincerely thank Reviewer #1 for their constructive feedback and are pleased to hear that our revisions have satisfactorily addressed their previous comments. All figures in question have been carefully reviewed and revised. We have increased the resolution of these figures to ensure that they are of high quality and clarity. This improvement will facilitate better visualization and understanding of the data presented.

Reviewer #2 (Remarks to the Author):

Comment 2.1: I appreciate the authors' efforts to respond to my and other referees' comments and to update their manuscript accordingly. I believe the revisions have largely addressed my concerns and have improved the clarity and quality of the manuscript.

Response 2.1: We are deeply grateful for your positive feedback on the revisions made to our manuscript. Your acknowledgment of the improved clarity and quality of our work is highly encouraging and validates the effort we have invested in addressing both your concerns and those of other referees.

Comment 2.2: Although I commend the authors for their work, I would also like to raise a few remaining points: In response to my comments, the authors show that some functions are specific to the SP of species with large genomes (including secretion systems and flagellum). This raises the question of whether the pattern observed between the SPs of species with large and small genomes could be driven by a subset of function specific to the SP of large species. Conversely, does the pattern observed here hold true when focusing on functions that are shared by the SPs of both species with large and small genomes?

Response 2.2:

We appreciate the reviewer's feedback. However, we wish to clarify that not all species with large genomes in our study had secretion systems and flagella, as shown in Supplementary Figure S7. **Additionally, in response to this new comment**, we utilized KEGG annotations to identify specific functions (i.e., KO numbers) that are shared between the SPs of species with

both large and small genomes. We selected a random subset of 6 species, comprising 3 with small genomes and 3 with large genomes, and compared their secreted proteomes (based on KEGG IDs shared between size categories), presenting the findings in panel C of Supplementary Figure S7. Given that no distinct pattern was observed, we opted to include this additional analysis as a supplementary figure.

Comment 2.3: *The authors assume that genes located on the same chromosome are under similar mutation rates (page 3, line 34). However, bacteria are known to have specific loci that are highly mutable ([https://doi.org/10.1016/S0960-9822\(00\)00005-1](https://doi.org/10.1016/S0960-9822(00)00005-1)) and there mounting evidence that the mutation rate is not constant across the chromosome but rather exhibits symmetry around the origin of replication (<https://journals.asm.org/doi/full/10.1128/mbio.01226-19>). The authors should better state the limitations of their hypothesis, or test whether CPs and SPs are equally distant from the origin of replication both in species with small and large genomes.*

Response 2.3:

We thank the reviewer for their additional comments and the provided references. To accommodate these, we have acknowledged the mentioned limitations and included the suggested reference on page 4, lines 1 and 2. The paragraph now reads as follows:

*“Given that genes coding for SPs and CPs typically evolve under similar mutational rates (as being located on the same bacterial chromosome) and are being therefore subjected to similar recombination effects, a reduction in the number of genes under negative selection compensated by the increase in the invariable genes indicates enhanced selection pressure¹..... **Nonetheless, it is crucial to acknowledge that in some bacterial species, the mutation rate exhibits symmetry around the origin of replication^{2,3}.**”*

Comment 2.4: *In figure 5a, the genes with no variations are labeled as “genes under no selection”. I believe it should rather read “genes with no variation” based on the authors’ response to my previous comments.*

Response 2.4:

We are grateful to the reviewer for highlighting this issue. In response to your insightful feedback, we have updated Figure 5a to accurately reflect “genes with no variation”.

Comment 2.5: *The authors have included estimates of growth rates, including on the basis of Codon Usage Biases (page 5, line 3). This is a welcome addition and I would invite the authors to consider citing original work on the codon usage imprint of growth: <https://doi.org/10.1371/journal.pgen.1000808>.*

Response 2.5:

Thank you for your positive feedback on our inclusion of growth rate estimates based on Codon Usage Biases. We're delighted you found this addition welcome. Following your suggestion, we've cited the seminal work on codon usage and growth rates on page 5, line 3.

Reviewer #3 (Remarks to the Author):

Comment 3.1: *In their response, the authors have essentially ignored my concerns and made no adjustments to the manuscript.*

Response 3.1:

We respectfully disagree with the reviewer. We have exercised the utmost diligence in addressing their concerns and have performed all the suggested analyses without contesting their necessity. To demonstrate how each comment was addressed and each analysis was performed, we will include at the end of this appeal the responses we provided for the reviewer's initial comments. As we have:

1) demonstrated the recovery of parts of the flexible genome within our MAGs collection (Supplementary Figure S9 and Supplementary Figure S6c-d);

2) shown that our methodology is robust across different genome natures (isolates, MAGs, or SAGs) and levels of completeness, as evidenced by the benchmarks presented in Supplementary Figures S3 and S4;

3) conducted comprehensive benchmarking of our DNA extraction, sequencing, assembly, and binning strategy, even though these methods are routinely used by us and others in the field (references 1–3);

4) included the requested abundance data in Supplementary Table S2 (column I, normalized coverage) and provided a visual summary in Figure 1;

5) assessed the impact of microdiversity on our analyses in Supplementary Figure S5, thereby strengthening our conclusions and methodology,

we believe that we have addressed the reviewer's requests with care and diligence.

References:

1. Coelho, L. P. et al. Towards the biogeography of prokaryotic genes. *Nature* 2021 601:7892 601, 252–256 (2021).
2. Paoli, L. et al. Biosynthetic potential of the global ocean microbiome. *Nature* 2022 607:7917 607, 111–118 (2022).
3. Rodríguez del Río, Á. et al. Functional and evolutionary significance of unknown genes from uncultivated taxa. *Nature* 2023 1–8 (2023) doi:10.1038/s41586-023-06955-z.

Comment 3.2: *The authors misunderstand the nature of incomplete genome recovery in various data types when using short read sequencing. In isolates, genome incompleteness is primarily caused by assembly failures in repeat regions, which usually comprise a minor part of microbial genomes. In SAGs, the main source of genome incompleteness is uneven DNA amplification, which is a random process with no known biases for or against certain genome regions. In MAGs, it has been well known that the less variable genome regions are more readily recovered, while the flexible part of the genome gets underrepresented. This is a major issue for the type of study that is presented in the manuscript, and it has to be properly addressed before making*

any conclusions about differences in the flexible gene composition among various microbial lineages. To be honest, I doubt that this is addressable, as the type of data appears to be at odds with the hypotheses tested. For a recent discussion about flexible gene underrepresentation in MAGs, see:

• Kerkvliet, J. J., Bossers, A., Kers, J. G., Meneses, R., Willems, R. & Schürch, A. C. Metagenomic assembly is the main bottleneck in the identification of mobile genetic elements. *PeerJ* 12 (2024). <https://doi.org:10.7717/peerj.16695>.

Response:

We respectfully disagree with the reviewer. Our team is fully aware of the methodological limitations inherent in generating MAGs, SAGs, and 'isolates genomes.' It appears there may be a misunderstanding of our work by the reviewer, who seems to divert the discussion away from its core subject towards the challenges of metagenomic assembly in capturing mobile genetic elements and highly variable regions. While we acknowledge the validity of this concern, we contend that our research, which involved the formation of genomic 'species clusters' and the examination of selection pressure across hundreds of thousands of cytoplasmic and secreted proteomes, is robust. Our protein-by-protein analysis has demonstrated insensitivity to the type of genome (i.e., MAG, SAG, or isolate; as shown in Supplementary Figures S3-S4) and to the habitat (as evidenced by Supplementary Figure S8 and the detailed data in Supplementary Dataset Tables S6 and S7).

More importantly, we argue that the use of MAGs and metagenomes to determine selection pressure in microbial lineages¹, gene families^{2,3}, or bacterial strains⁴ is an approach that is widely accepted and utilized in the field.

References

1. Ngugi, D. K. et al. Postglacial adaptations enabled colonization and quasi-clonal dispersal of ammonia-oxidizing archaea in modern European large lakes. *Sci Adv* 9, eadc9392 (2023).
2. Coelho, L. P. et al. Towards the biogeography of prokaryotic genes. *Nature* 601, 252–256 (2022).
3. Rodríguez del Río, Á. et al. Functional and evolutionary significance of unknown genes from uncultivated taxa. *Nature* 2023 1–8 (2023) doi:10.1038/s41586-023-06955-z.
4. Olm, M. R. et al. inStrain profiles population microdiversity from metagenomic data and sensitively detects shared microbial strains. *Nature Biotechnology* 2021 39:6 39, 727–736 (2021).

Comment 3.3: *There is no mentioning of this core data limitation in either the original, or the revised manuscript. The fact that same techniques were applied on all datasets does not solve the problem, as claimed by the authors, because different proportions of genome may be missing in different MAGs. The intercomparison of MAGs, isolates and SAGs, provided in Figure 3, appears extremely limited in scope and hard to interpret, given the fact that there is no associated method description or even reference to this figure from the manuscript text.*

Response 3.3:

We respectfully disagree with the reviewer. On page 14, we introduced a new section entitled '**Limitations**,' which discusses the challenges associated with partial genomes analyses (Lines 4-15).

The intercomparison of MAGs, isolates, and SAGs mentioned by the reviewer is presented in Supplementary Figure S3. We concur with the reviewer that this figure was not adequately referenced in the manuscript. This oversight occurred due to our misunderstanding about the rebuttal being posted online. A detailed description of this benchmarking process was provided in our response to Reviewer 1 (Response 1.4). We have now included the following text in our Methods section, on page 13, lines 24-34:

“We conducted an SP/CP benchmark analysis using an abundant environmental marine species, *Prochlorococcus_B marinus_B* (Supplementary Figure S3). This species was chosen due to the availability of cultured representatives, single-cell amplified genomes (SAGs), and metagenome-assembled genomes (MAGs). By comparing the SPs/CPs ratio among genomes from cultures, SAGs, and MAGs, it becomes evident that the results are indeed comparable, with cultures and MAGs yielding highly similar outcomes. These results were further supported by the absence of statistical difference (ANOVA; $df = 2$; $f\text{-value} = 0.3852$; $p\text{-value} = 0.6811$) in the SPs/CPs ratio among the three groups (i.e., culture, SAGs and MAGs). The small deviation observed in the SAGs can likely be attributed to errors introduced during the single-cell amplification steps.”

Comment 3.4: *The rebuttal makes a statement “Executed a benchmark for DNA extraction, sequencing, and assembly (Supplementary Information - Table 1). However, this table is just a list of MAGs used in the study.*

Response:

We respectfully disagree with the reviewer. We have provided detailed explanations of our benchmarking for DNA extraction, sequencing, and assembly in Response 3.3.1, which is referenced within this document as Response 3.3.1. Table 1, which pertains to this discussion, can be found in the supplementary materials as follows:

Supplementary Table 1

Mock community	Sequencing Technology	Average ANI*	Average Coverage
Bacillus	Illumina	100	97.91
Enterococcus	Illumina	100	97.17
Escherichia	Illumina	99.95	85.95
Lactobacillus	Illumina	99.99	91.75
Listeria	Illumina	99.99	97.25
Pseudomonas	Illumina	99.89	96.97
Salmonella	Illumina	99.92	87.04
Staphylococcus	Illumina	99.97	95.88

*ANI: average nucleotide identity.

Supplementary Table 1 clearly outlines which bacterial strain was analyzed, the sequencing technology utilized, and the average nucleotide identity and coverage achieved. Furthermore, we provided the reviewer with comprehensive explanations in our reply.

Comment 3.5: *The rebuttal also states “Analysed 592 genomes belonging to 57 environmental species (median number of genomes/species = 9) and identified similar diversity patterns in CPs and SPs diversity as observed in our generated datasets (Supplementary Table S6).” However, Table S6 contains only a list of these genomes. Contrary to authors’ claims, I was not able to find any information on the “diversity patterns in CPs and SPs diversity” of these genomes in Table S6 or anywhere else in the manuscript.*

Response 3.5:

We appreciate the reviewer's observation and agree with their comment. The oversight was unintentional and resulted from a minor omission on our part. We would like to point out that the relevant information is detailed in Supplementary Tables S6 and S7, as well as in Supplementary Figure S8. We are now confident that this ensures the results can be easily accessed and reviewed by all interested readers.

Comment 3.6: *In their rebuttal, the authors refer to results of a 2017 CAMI, which found MEGAHIT and MetaBAT outperforming some of the other algorithms for metagenome assembly and binning, as a proof of the robustness of their MAG generation procedure. However, the 2017 CAMI study found that the MEGAHIT-MetaBAT combo had <20% recall (genome recovery) and <60% precision (the inverse of contamination) when processing “high complexity” mock metagenomes (although the complexity of these materials was likely much lower than the natural freshwater microbiome). In order to properly interpret results presented in this manuscript, these methodological limitations should be given full, quantitative consideration.*

Response 3.6:

We thank the reviewer for their comment. As stated in our initial Response 3.1 we used the MEGAHIT-MetaBAT2 combo due to its performance and wide acceptance in the community^{1,2}. We further added the following statement in the current version of the manuscript: “MetaBAT2 was selected based on its performance as the best-performing single binner in independent benchmarks⁸ and its efficiency, being 10 to 50 times faster than other commonly used binners⁹.” (page, lines 14-16).

References

1. Ngugi, D. K. et al. Postglacial adaptations enabled colonization and quasi-clonal dispersal of ammonia-oxidizing archaea in modern European large lakes. *Sci Adv* 9, eadc9392 (2023).
2. Coelho, L. P. et al. Towards the biogeography of prokaryotic genes. *Nature* 601, 252–256 (2022).

Comment 3.7: *The rebuttal makes no logical sense. A saturation of gene recovery with each additional MAG is expected to be as prevalent or even more pronounced if such MAGs underrepresent the flexible pangenome.*

Response 3.7: We appreciate the reviewer's comment, though it appears there may have been a misunderstanding, or the context may not have been fully considered. We have conducted pangenome analyses and have elaborated on this in our rebuttal, specifically in Response 3.2.

Comment 3.8: *The authors did substantial additional work benchmarking their workflow, which is praiseworthy. However, they have not addressed the core question: can the type of data used in the manuscript accurately discriminate evolutionary patterns of specific proteins within specific species-level lineages? This is not easy, but one of the best approaches may be by testing whether these patterns can be accurately represented in MAGs produced from the “high complexity” mock metagenomes of 2017 CAMI or similar datasets, containing multiple strains of the same microbial species. Instead, the authors analyzed a low-complexity mock metagenome with only one representative per species and did not even try to evaluate the accuracy of their evolutionary metrics when applied on this dataset. In fact, results of this analysis do not appear to be included in the manuscript at all.*

Response:

We analyzed one of the most widely utilized types of mock communities. Given that the median genomic identity among the genomes obtained from these mock communities was 99.98%, with no observable differences, we did not include additional evolutionary analyses in the main text. Nonetheless, we have undertaken further analyses on a diverse set of genomes. These are detailed in Supplementary Figure 8 and Supplementary Data S6 and S7, as previously mentioned. Should there be a need for more detailed genomic comparison data, we are prepared to supply this information.

Comment 3.9: *I appreciate the authors’ effort to address my comment. Unfortunately, neither Figure 3, nor Table S2 contain quantitative, interpretable information. The inset in Figure 3 lacks legend, while bar charts in Table S2 lack scale.*

Response 3.9:

We respectfully disagree with the reviewer. Figure 3 includes an inset that displays the quantitative abundance of species with small versus large genomes, relative to the total abundance of the dataset. Supplementary Table S2 provides normalized coverage values **for each species cluster, in a genome-by-genome fashion**. Normalizing abundance is a common practice for comparisons since metagenomic datasets typically vary in the number of reads they contain. This approach is standard in the field.

Comment 3.10: I find this rebuttal highly confusing. For example, the authors state that “no MAG could be recovered from that respective sample” right after saying that that same sample “allowed us to investigate the microdiversity of 23 MAGs that are organized in 23 of the 30 species- clusters discussed in the manuscript”. Even more confusing is the concluding statement: “Although the investigation of microdiversity distribution across different species over time and space is an intriguing subject in its own right, it extends beyond the scope of the presented study.” The manuscript specifically concerns genetic variation within microbial species, i.e., microdiversity, so ignoring it is an oxymoron. The figures referenced in this rebuttal contain no results relevant to my comment.

Response:

We regret that the reviewer found our rebuttal confusing. We suspect that there might have been a misunderstanding of the core message of our paper. We have analyzed microdiversity using the software suggested by the reviewer and presented the results in Supplementary Figure S5. Additional microdiversity analyses were also performed and included in Supplementary Figure S6. Since these analyses did not bring any new evidence or deviate from our previous observations, they were included as supplementary information.

We would like to emphasize that our methodology is well-established and supported by significant references in the field¹⁻⁵. We have made every effort to address all comments raised by the reviewer and have conducted all the analyses requested.

References:

1. Ngugi, D. K. et al. Postglacial adaptations enabled colonization and quasi-clonal dispersal of ammonia-oxidizing archaea in modern European large lakes. *Sci Adv* 9, eadc9392 (2023).
2. Coelho, L. P. et al. Towards the biogeography of prokaryotic genes. *Nature* 601, 252–256 (2022).
3. Rodríguez del Río, Á. et al. Functional and evolutionary significance of unknown genes from uncultivated taxa. *Nature* 2023 1–8 (2023) doi:10.1038/s41586-023-06955-z.
4. Olm, M. R. et al. inStrain profiles population microdiversity from metagenomic data and sensitively detects shared microbial strains. *Nature Biotechnology* 2021 39:6 39, 727–736 (2021).
5. Bulzu, P. A. et al. Casting light on Asgardarchaeota metabolism in a sunlit microoxic niche. *Nature Microbiology* 2019 4:7 4, 1129–1137 (2019).

The initial rebuttal provided for Review 3

Reviewer #3 (Remarks to the Author):

The authors have produced a large set of metagenome-assembled genomes (MAGs) of bacteria and archaea from freshwater bodies in Central Europe and used it to investigate relative rates of adaptation among microbial taxa. Their analyses indicated elevated selective pressure on secreted proteins (SPs) in small MAGs, as compared to large MAGs. This was interpreted as a proof of an adaptive stasis in microbial taxa with small genomes, caused by the reduced functional redundancy in their genomes (no “backup plan”). This study addresses important questions in microbial evolution and presents a compelling hypothesis that deserves robust testing. However, I believe that the type of core data used in this study - >5,500 MAGs – is prone to major methodological biases and may be fundamentally unsuitable for the type of analyses performed here. While MAGs are very

instrumental in the recovery genomic information from yet uncultured taxa, they are highly prone to both the inclusion of genome regions from unrelated organisms (chimerism) and exclusion of hypervariable genome regions (pangenome under-sampling). Importantly, this not reflected by genome completeness estimates by CheckM and similar tools, which simply count the presence of a small subset of core genes. For a few studies exploring these methodological challenges in great detail, see the following:

1 Sczyrba, A. et al. *Critical Assessment of Metagenome Interpretation - A benchmark of metagenomics software. Nature Methods* 14, 1063-1071 (2017).

<https://doi.org:10.1038/nmeth.4458>

2 Vollmers, J., Wiegand, S., Lenk, F. & Kaster, A. K. *How clear is our current view on microbial dark matter? (Re-)assessing public MAG & SAG datasets with MDMcleaner. Nucleic Acids Res* (2022). <https://doi.org:10.1093/nar/gkac294>

3 Orakov, A. et al. *GUNC: detection of chimerism and contamination in prokaryotic genomes. Genome Biol* 22, 178 (2021). <https://doi.org:10.1186/s13059-021-02393-0>

Response 3.1:

We appreciate the thoughtful comment from the reviewer and their reference to pertinent work in the field. We acknowledge the intrinsic challenges linked to metagenome-assembled genomes (MAGs), particularly in the context of potential pangenome undersampling.

It is important to note that pangenome undersampling is a shared concern affecting not only MAGs but also single-amplified genomes and cultured isolates. The limitations stem from the inherent difficulty in capturing the entirety of species diversity, a challenge exacerbated by the continuous evolution of bacterial populations across time. To address some of the mentioned challenges, our study employed repeated sampling campaigns in the selected environments. This iterative sampling method significantly bolstered our capacity to capture some of the species' flexible genomes. To enhance our capability to investigate the flexible genome, we meticulously curated our pdCEL database, prioritizing species clusters with a minimum of 9 representatives. This rigorous selection process was devised to guarantee a thorough representation of species clusters and their flexible genomic elements. Additionally, we conducted a pangenome analysis and introduced a 'Limitations' section into the manuscript, transparently addressing some of the methodological constraints associated with species diversity analysis. Additional comments regarding the concerns about pangenome undersampling are addressed in Response 3.2.

In terms of computational methodologies, we highlight that MEGAHIT distinguished itself as one of the top-performing software in the CAMI challenge¹ indicated by the reviewer, showcasing superior contiguity and successful assembly of a substantial fraction of genomes across diverse coverages. Notably, it garnered the highest scores in recovering both unique strains (genomes with <95% ANI to any other in the dataset) and common strains (genomes

with an ANI $\geq 95\%$ to another genome in the dataset). Also, the MetaBAT binning software demonstrated high accuracy, and while MetaBAT2 (which was used) wasn't explicitly tested, we capitalized on its performance by integrating it with a comprehensive dataset of abundance data known to enhance the effectiveness and sensitivity of multi-coverage metagenomic binning software². Our selection of MetaBAT2 is based on its performance as the best-performing single binner in independent benchmarks³ and its efficiency, being 10 to 50 times faster than other commonly used binners⁴. Additionally, we emphasize that a meticulous step of metagenomic binning curation was implemented to avoid misbinning of taxonomically unrelated contigs, as outlined in the Methods section.

When evaluating our analysis workflow, it is essential to state that the observed consistency between SP and CP diversity underscores the absence of major methodological flaws in our data generation process. The uniformity in our data generation protocols ensure that any displayed variation between these two protein categories indicate substantive biological differences rather than methodological artifacts. To further enhance the robustness of our methods, we conducted the following additional analyses:

A) Performed a benchmark for intraspecific protein diversity analysis utilizing the cultivated representatives, SAGs, and MAGs of some environmental species available (Supplementary Information - Fig. 3). The comparison of the SP/CP ratios among genomes originating from cultures, SAGs, and MAGs, revealed comparable results. Especially, culture-genomes and MAGs from the same species display a highly similar pattern. The slight deviation regarding the species-specific SP/CP ratio of MAGs compared to that of SAGs is likely attributable to errors that were introduced during the single-cell amplification steps. This comparative analysis supports the reliability of our MAG collection to draw meaningful conclusions about species diversity and evolution.

B) Executed a benchmark for DNA extraction, sequencing, and assembly (Supplementary Information - Table 1). Presented also in Response 3.4. See also Method section.

C) Analysed 592 genomes belonging to 57 environmental species (median number of genomes/species = 9) and identified similar diversity patterns in CPs and SPs diversity as observed in our generated datasets (Supplementary Table S6).

References:

1. Sczyrba, A. et al. Critical Assessment of Metagenome Interpretation—a benchmark of metagenomics software. *Nature Methods* 2017 14:11 14, 1063–1071 (2017).
2. Mattock, J. & Watson, M. A comparison of single-coverage and multi-coverage metagenomic binning reveals extensive hidden contamination. *Nature Methods* 2023 20:8 20, 1170–1173 (2023).
3. Hofmann, F. M. P. et al. AMBER: Assessment of Metagenome BinnERs. *Gigascience* 7, 1–8 (2018).

4. Meyer, F. et al. Critical Assessment of Metagenome Interpretation: the second round of challenges. *Nature Methods* 2022 19:4 19, 429–440 (2022).

Pangenome under-sampling is particularly relevant here, as it may introduce major biases leading to flawed conclusions. As an illustration, despite two decades of substantial efforts, marine microbiologists still struggle to produce MAGs of Pelagibacter and Prochlorococcus, the two most abundant genera in the ocean, both of which have small genomes. Meanwhile, it is well known that pangenomes of Pelagibacter and Prochlorococcus are enormous, with ample evidence for frequent horizontal gene transfer and recombination. Thus, the situation with Pelagibacter and Prochlorococcus is the complete opposite to the adaptive stasis proposed in the manuscript: they evolve rapidly and recombine their genomes frequently, challenging scientists' ability to represent their populations in MAGs.

Response 3.2:

We appreciate the reviewer's observation regarding the potential misinterpretation of undersampled pangenomes as adaptive stasis, particularly in small genome species like Prochlorococcus and Pelagibacter, which exhibit substantial variability.

However, it is crucial to note that our study focused on common freshwater bacterial species, which may not necessarily exhibit the same patterns. Furthermore, we want to underscore that the genera **Pelagibacter** and **Prochlorococcus** encompass **855** and **332 species**, respectively (according to the GTDB database as of November 2023), contributing to the extensive genus-level pangenome mentioned by the reviewer. However, we want to emphasize that our analysis specifically targeted intraspecific rather than intragenic diversity (as was the case for Pelagibacter and Prochlorococcus).

Recombination events, such as horizontal gene transfer, play a pivotal role in shaping the complexity and size of microbial pangenomes¹. In accordance with the Neutral Theory outlined in "Microbial Practice: Challenges in Bacterial Population Genetics"², these genetic variations can induce neutral, deleterious, or adaptive effects within genomes of bacterial species. Despite reports suggesting that interspecies variability can enhance a species' adaptive potential³, conflicting patterns have been reported by other studies^{4,5}. One plausible conclusion that can be drawn from previous work is that species exhibit different, species-dependent pangenome sizes. Also, the accessory genes within them exert contrasting effects on the species' adaptation to their environment.

Since we cannot fully claim the inherent completeness nor adaptability of the pangenomes of the analysed species, we can only showcase the variability observed through sampling numerous ecosystems across Europe, encompassing diverse seasonal conditions and physicochemical parameters. To specifically address the reviewer's comment, we analysed the pangenome of each species-cluster using Roary as mentioned in the methods section (Supplementary Information Fig. S9). We find that the number of genes belonging to the

pangenome of each species increases with each MAG added, while the rate of gene addition per genome decreases. This indicates the incomplete recovery of the total amount of accessory genes for each species but also shows that interspecies variability has been caught. Simultaneously, the slow saturation of newly added genes suggests that none of the presented species-clusters is greatly suffering from pangenome undersampling (Supplementary Information Fig. S9). While the variability persists across all species groups featuring 9 to 19 MAGs, the conserved structures within the sampled variability, notably within the secreted proteome of small MAG species, supports the assumption that such specialized species might indeed exploit stasis as a successful ecological strategy.

References:

1. Medini, D., et al., *The microbial pan-genome*. Current Opinion in Genetics & Development, 2005. 15(6): p. 589-594.
2. Rocha, E.P.C., *Neutral Theory, Microbial Practice: Challenges in Bacterial Population Genetics*. Molecular Biology and Evolution, 2018. 35(6): p. 1338-1347.
3. Sela, I., Y.I. Wolf, and E.V. Koonin, *Theory of prokaryotic genome evolution*. Proc Natl Acad Sci U S A, 2016. 113(41): p. 11399-11407.
4. Baumdicker, F., W.R. Hess, and P. Pfaffelhuber, *The Infinitely Many Genes Model for the Distributed Genome of Bacteria*. Genome Biology and Evolution, 2012. 4(4): p. 443-456.
5. Conrad, R.E., et al., *Toward quantifying the adaptive role of bacterial pangenomes during environmental perturbations*. The ISME Journal, 2022. 16(5): p. 1222-1234.

In order to make their data more interpretable, authors of the manuscript should evaluate how accurately their MAGs represent the genomic composition of analyzed microbial communities. The following is essential:

1. *How accurate are the metagenome assembly and binning methods used in the study? This can be addressed by validating the entire workflow with mock metagenomes, such as those used in the Sczyrba et al. publication cited above and publicly available.*

Response 3.3.1:

We thank the reviewer for pointing out that demonstrating the accuracy of the data processing pipeline adds weight to our analyses, especially when it comes to the genomic composition of environmentally-derived MAGs.

To validate the accuracy of the method, DNA was extracted from the **Microbial Community Standard ZymoBIOMICS D6300** (Zymo Research Inc., Irvine, CA, USA) and sequenced with Illumina, employing the same protocols for DNA extraction and library preparation as

outlined in the Methods section. The data was curated and assembled using the identical procedures as described for the environmental samples to confirm the reliability of the analysis. The mock community comprises 10 different species, including 8 bacteria (3G+, 5G-). After quality trimming, approximately 100 million paired-end reads (2x150bp) were retained (>Q18). Megahit (v.1.2.9) was employed with default options to assemble the quality-filtered data, utilizing various kmer sizes (39, 49, 69, 89, 109, 129, 149). Due to the unique nature of the mock community and the binner (MetaBAT2) relying on differential abundance data from sampled time series, replicating the exact binning strategy used in the manuscript was not feasible. Further details regarding the binning are provided in Response 3.1.

For binning the mock community, we implemented a taxonomy-based approach similar to the one employed for the post-binning curation described in the manuscript for the environmental data. In essence, each contig with a length exceeding 3,000 bp was utilized to predict proteomes (PRODIGAL33 v2.6.3), which were queried (using mmseqs2 searches) against the curated prokaryotic GTDB database (R05-RS95). The results were then converted into a BLAST-tab formatted file (using mmseqs convertalis), from which individual top hits were extracted. The taxonomic labels derived from these hits were used to annotate the queried proteomes, and the taxonomic information of the contigs was employed for binning.

The accuracy of DNA extraction, sequencing, and metagenomic assembly was evaluated by calculating the average nucleotide identity between the generated draft genomes and the provided reference genomes of the ZymoBIOMICS Microbial Community Standard D6300. **The draft genomes of the eight bacteria exhibited a median average nucleotide identity of 99.98%, with a median coverage value of 96.425% compared to the reference.** This highlights the precision and reliability of the applied method.

ZymoBIOMICS.STD	SequencingTechnology	Avg_ANI (%)	Avg_Cov (%)
Bacillus sp.	Illumina	100	97.91
Enterococcus sp.	Illumina	100	97.17
Escherichia sp.	Illumina	99.95	84.95
Lactobacillus sp.	Illumina	99.99	91.75
Listeria sp.	Illumina	99.99	97.25
Pseudomonas sp.	Illumina	99.89	96.97
Salmonella sp.	Illumina	99.92	87.04
Staphylococcus sp.	Illumina	99.97	95.88

2. What fraction of genes encoding secreted and cytoplasmic proteins by the entire community is represented by the obtained MAGs? This can be done by the recruitment of individual metagenome reads on MAGs.

Response 3.3.2:

We appreciate the reviewer's suggestions. To address the concern, we conducted normalized read mapping for each of the approximately 5,500 metagenome-assembled genomes (MAGs). In Figure 3, we included an inset that illustrates the total abundance of the targeted species clusters (342 MAGs) in relation to the total abundance of all MAGs obtained. Individual abundances are now available in Supplementary Table S2.

3. How do metagenome assembly algorithms, which produce consensus sequences from a multitude of organisms, impact the ability to measure evolutionary pressure metrics, such as Dn/Ds ratio? This can be done by software designed to explore intra-population variation in MAGs, such as inStrain: Olm, M. R. et al. inStrain profiles population microdiversity from metagenomic data and sensitively detects shared microbial strains. Nat Biotechnol 39, 727–736 (2021).

Response 3.3.3:

We appreciate the reviewer's concerns regarding the potential influence of metagenome assembly algorithms on evolutionary pressure metrics such as the ratio between non-synonymous and synonymous mutations. In response to these concerns, we followed the reviewer's suggestions and employed the **inStrain** software¹ to evaluate the impact of existing microdiversity on our selection pressure metrics.

Considering the expected microdiversity within most species, we intentionally chose one metagenome, specifically RH-22June17, to investigate the species-level microdiversity of the metagenome-assembled genomes (MAGs) derived from this sample. This sample allowed us to investigate the microdiversity of 23 MAGs that are organized in 23 of the 30 species-clusters discussed in the manuscript. From the resulting 7 species-clusters, no MAG could be recovered from that respective sample. Our examination focused on the following key metrics, including:

- i) **Nucleotide Diversity:** Examining the overall genetic variation within the 23 selected species.
- ii) **Divergent Site Counts:** Representing the number of positions with single nucleotide variations or substitutions, providing insights about the genomic diversity.
- iii) **pN/pS Ratio:** a measure of whether the set of mutations in a gene are biased towards synonymous (S) or non-synonymous (N) mutations. The pN/pS ratio is calculated based on the mutations that are at least present on two alleles. $pN/pS > 1$ shows bias towards N mutations, indicating that the respective gene is under active positive selection and acquires mutations. $pN/pS < 1$ represents a bias towards S mutations, indicating that the gene is under stabilizing (negative) selection and does not accept mutations leading to structural changes in the amino acid sequence. $pN/pS = 1$ shows that N and S mutations randomly occur at the expected rate, potentially indicating that the respective gene is non-functional.

We consider these metrics as robust indicators for capturing microdiversity within our specified species.

Concerning nucleotide diversity, we observed no significant differences between SPs and CPs diversity (Pairwise Wilcoxon Rank Sum Tests, p-value = 0.36). Although a slightly higher diversity was generally noted for small genome species, these differences did not reach statistical significance (Supplementary Information – Fig. S5a). A similar pattern was observed in the case of the number of divergent sites per gene, with no statistical differences between SPs and CPs (Pairwise Wilcoxon Rank Sum Tests, p-value = 0.36) (Supplementary Information – Fig. S5b) or between genome-size categories (i.e., large and small). However, the pN/pS results presented a highly consistent pattern with the conducted evolutionary pressure analyses (Supplementary Information - Figure S5c). Most of the within-species diversity is under negative selection. Furthermore, the genes encoding the SPs of the small genome species exhibit very low levels of positive selection (Supplementary Information - Figure S5d). Therefore, we assert that extant microdiversity does not introduce major biases into our results; rather, it reinforces our confidence in the reliability of the employed methods.

We further extracted 100 million reads from the aforementioned metagenome (i.e., RH-22June17) and calculated their similarity using the blastn algorithm against the 23 metagenome-assembled genomes (MAGs), each representing a distinct bacterial species. Subsequently, we generated distribution and recruitment plots based on the compiled dataset (Supplementary Information - Figure S6). Our analysis indicates that while microdiversity is discernible, the majority of the reads are distributed within clonal complexes, denoted by more than 99% sequence similarity with the analysed MAGs. Considering our comprehensive benchmarking of DNA extraction, sequencing, assembly, and microdiversity analyses, we maintain that our methodology remains robust despite the presence of microdiversity. **Furthermore, we argue that if microdiversity indeed had a significant effect on our results, we would not have observed the evident distinctive patterns between large and small genome species, as well as between SPs and CPs.**

Although the investigation of microdiversity distribution across different species over time and space is an intriguing subject in its own right, it extends beyond the scope of the presented study.

Reference:

1. Olm, M.R., Crits-Christoph, A., Bouma-Gregson, K. *et al.* inStrain profiles population microdiversity from metagenomic data and sensitively detects shared microbial strains. *Nat Biotechnol* **39**, 727–736 (2021).”

REVIEWERS' COMMENTS

Reviewer #2 (Remarks to the Author):

I thank the authors for their additional clarifications and adjustments and I now believe they have appropriately addressed my concerns.

Reviewer #3 (Remarks to the Author):

Authors' responses to my comments are superficial and fail to address my key concern. MAGs are very instrumental in many research areas, but, like any data type, have limitations and can be used appropriately only when relevant conditions are met. Scientific questions that the authors are trying to address are focused on microdiversity within microbial populations. MAGs are consensus sequences of a multitude of individual organisms and therefore, by definition, lack information about microdiversity within populations. In order to use MAGs, the authors have to provide evidence that they have a way to circumvent this fundamental limitation of the data type that they are using. Unfortunately, this has not been done.

Dear Editor,

We appreciate the thoughtful comments provided by the referees on our manuscript with the identifier NCOMMS-23-32470-B-Z. On behalf of my co-authors, I am pleased to address the raised issues and provide a detailed response to each comment. The reviewer's statements are quoted in italics, followed by our responses in regular blue text.

REVIEWERS' COMMENTS

Reviewer #2 (Remarks to the Author):

I thank the authors for their additional clarifications and adjustments and I now believe they have appropriately addressed my concerns.

Response: We sincerely thank Reviewer #2 for their constructive feedback and are pleased to hear that our revisions have satisfactorily addressed their previous concerns.

Reviewer #3 (Remarks to the Author):

Authors' responses to my comments are superficial and fail to address my key concern. MAGs are very instrumental in many research areas, but, like any data type, have limitations and can be used appropriately only when relevant conditions are met. Scientific questions that the authors are trying to address are focused on microdiversity within microbial populations. MAGs are consensus sequences of a multitude of individual organisms and therefore, by definition, lack information about microdiversity within populations. In order to use MAGs, the authors have to provide evidence that they have a way to circumvent this fundamental limitation of the data type that they are using. Unfortunately, this has not been done.

Response:

We deeply appreciate the reviewer's detailed insights and the substantial effort invested in evaluating our manuscript. We concur with the reviewer's observation regarding the nature of metagenome-assembled genomes as consensus sequences that may not fully capture microdiversity within populations. Recognizing this potential bias, we have proactively addressed it by performing analyses using the inStrain software, as kindly suggested. This approach has allowed us to assess the impact of microdiversity on our findings, which is now thoroughly documented in Supplementary Figure S5. The implications of these analyses are further corroborated by the distribution and recruitment plots presented in Supplementary Figure S6, as detailed in our initial response (Response 3.3.3).

We appreciate the opportunity to clarify a crucial aspect of our study that appears to have been misinterpreted. The primary aim of our research is not to dissect microdiversity

within microbial populations. Rather, our focus is on examining the genomes of prevalent environmental species to elucidate the impact of evolutionary pressures on genes coding for secreted proteins, particularly in relation to specific bacterial lifestyles. This distinction is fundamental to our approach, as our analysis predominantly compares two categories of genes within the same genome sets.

Furthermore, it is important to address the concern regarding microdiversity. We argue that the bulk of microdiversity, primarily represented by synonymous mutations, is unlikely to have a meaningful impact on the proteomes we have analyzed. Our findings, highlighted in Figure 5 and Supplementary Figure S5, support this perspective by demonstrating that microdiversity does not significantly alter the core conclusions of our study regarding the evolutionary pressures on secreted versus cytoplasmic proteins.

We hope this response clarifies our position and the steps we've taken to rigorously address the reviewer's concerns.